# Glioma synapses recruit mechanisms of adaptive plasticity

Kathryn R. Taylor[1], Tara Barron[1], Alexa Hui[1], Avishay Spitzer[2], Belgin Yalçin[1], Alexis E. Ivec[1], Anna C. Geraghty[1], Griffin G. Hartmann[1], Marlene Arzt[1], Shawn M. Gillespie[1], Yoon Seok Kim[1], Samin Maleki Jahan[1], Helena Zhang[1], Kiarash Shamardani[1], Minhui Su[1], Lijun Ni[1], Peter P. Du[1], Pamelyn J. Woo[1], Arianna Silva-Torres[1], Humsa S. Venkatesh[1], Rebecca Mancusi[1], Anitha Ponnuswami[1], Sara Mulinyawe[1], Michael B. Keough[1], Isabelle Chau[1], Razina Aziz-Bose[1], Itay Tirosh[2], Mario L. Suvà[3,4] & Michelle Monje[1,5,6,7,8 ✉]

The role of the nervous system in the regulation of cancer is increasingly appreciated. In gliomas, neuronal activity drives tumour progression through paracrine signalling factors such as neuroligin-3 and brain-derived neurotrophic factor[1-3] (BDNF), and also through electrophysiologically functional neuron-to-glioma synapses mediated by AMPA (α-amino-3-hydroxy-5-methyl-4-isoxazole propionic acid) receptors[4,5]. The consequent glioma cell membrane depolarization drives tumour proliferation[4,6]. In the healthy brain, activity-regulated secretion of BDNF promotes adaptive plasticity of synaptic connectivity[7,8] and strength[9-15]. Here we show that malignant synapses exhibit similar plasticity regulated by BDNF. Signalling through the receptor tropomyosin-related kinase B[16] (TrkB) to CAMKII, BDNF promotes AMPA receptor trafficking to the glioma cell membrane, resulting in increased amplitude of glutamate-evoked currents in the malignant cells. Linking plasticity of glioma synaptic strength to tumour growth, graded optogenetic control of glioma membrane potential demonstrates that greater depolarizing current amplitude promotes increased glioma proliferation. This potentiation of malignant synaptic strength shares mechanistic features with synaptic plasticity[17-22] that contributes to memory and learning in the healthy brain[23-26]. BDNF–TrkB signalling also regulates the number of neuron-to-glioma synapses. Abrogation of activity-regulated BDNF secretion from the brain microenvironment or loss of glioma TrkB expression robustly inhibits tumour progression. Blocking TrkB genetically or pharmacologically abrogates these effects of BDNF on glioma synapses and substantially prolongs survival in xenograft models of paediatric glioblastoma and diffuse intrinsic pontine glioma. Together, these findings indicate that BDNF–TrkB signalling promotes malignant synaptic plasticity and augments tumour progression.

Gliomas, including glioblastoma and diffuse midline gliomas (DMG), are the most common and lethal primary brain cancers in children and adults[27]. Progression of glioma is robustly regulated by interactions with neurons[1-5], including tumour initiation[3,28], growth[1-5,28] and invasion[5,29]. Neuron–glioma interactions include both paracrine factor signalling[1,3,28] and electrochemical signalling through AMPA receptor (AMPAR)-mediated neuron-to-glioma synapses[4,5]. Synaptic integration of high-grade gliomas into neural circuits is fundamental to cancer progression in preclinical model systems[4,5,29] and in human patients[30]. We hypothesized that gliomas may recruit mechanisms of adaptive neuroplasticity to elaborate and reinforce these powerful growth-promoting neuron–glioma interactions, and that neuronal activity-regulated BDNF signalling to the TrkB

receptor in glioma cells may have a crucial role in such malignant plasticity.

## BDNF–TrkB signalling drives glioma growth

Paediatric gliomas express high levels of the BDNF receptor TrkB (encoded by *NTRK2*) in malignant cells (Extended Data Fig. 1a,b). Unlike adult glioblastoma[31], paediatric high-grade gliomas such as DMGs of the brainstem, also called diffuse intrinsic pontine glioma (DIPG), do not express *BDNF* (Extended Data Fig. 1a). This suggests a microenvironmental source of BDNF ligand, consistent with previous evidence[1]. We therefore tested the role of neuronal activity-regulated BDNF secretion into the tumour microenvironment using a genetically

[1]Department of Neurology and Neurological Sciences, Stanford University, Stanford, CA, USA. [2]Department of Molecular Cell Biology, Weizmann Institute of Science, Rehovot, Israel. [3]Department of Pathology and Center for Cancer Research, Massachusetts General Hospital and Harvard Medical School, Boston, MA, USA. [4]Broad Institute of MIT and Harvard, Boston, MA, USA. [5]Department of Pediatrics, Stanford University, Stanford, CA, USA. [6]Department of Pathology, Stanford University, Stanford, CA, USA. [7]Department of Neurosurgery, Stanford University, Stanford, CA, USA. [8]Howard Hughes Medical Institute, Stanford California, Stanford, CA, USA. ✉e-mail: mmonje@stanford.edu

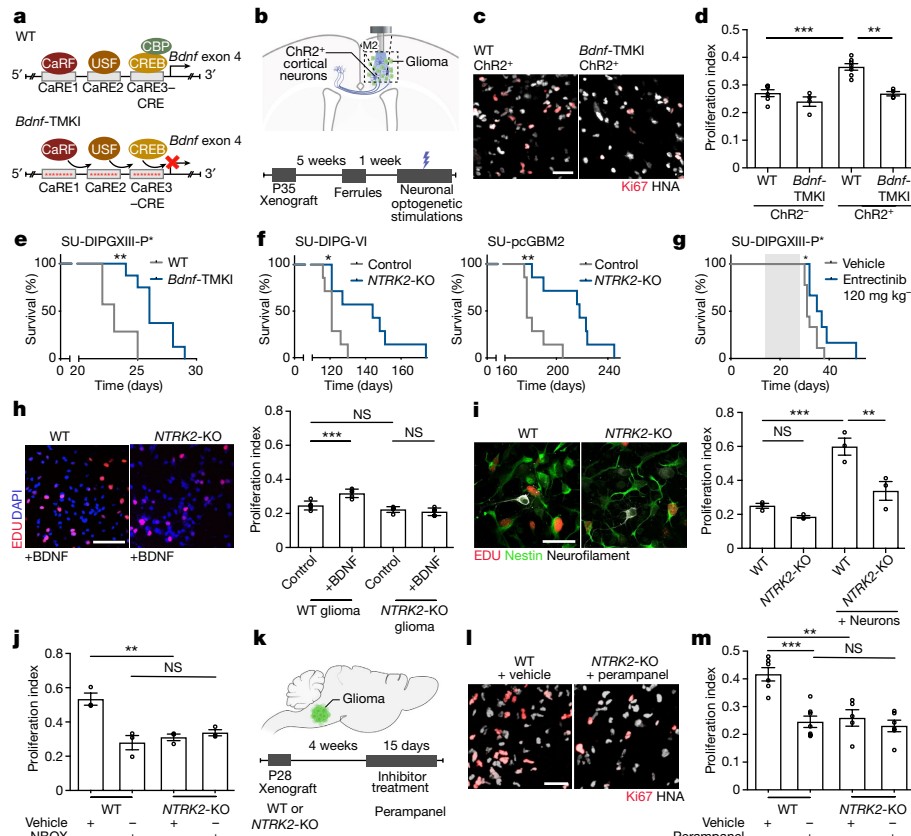

**Fig. 1 | Activity-regulated BDNF promotes glioma progression. a**, The *Bdnf*-TMKI model. CaRE, calcium regulatory element binding site; CRE, cAMP response element; WT, wild type. **b**, Optogenetic paradigm. M2, mouse premotor frontal cortex; P, postnatal day. **c**, Representative images of glioma (SU-DIPG-VI) xenografted into wild-type and *Bdnf*-TMKI cortex following blue-light stimulation of ChR2⁺ cortical neurons. HNA (grey) marks glioma cells, Ki67 (red) marks proliferating cells. Scale bar, 50 μm. **d**, Proliferation index (Ki67⁺ cells/HNA⁺ glioma cells) of xenografted SU-DIPG-VI glioma in wild-type or *Bdnf*-TMKI mice stimulated optogenetically (ChR2⁺ cortical neurons) or mock-stimulated (ChR2⁻ neurons). $n = 6$ (wild-type ChR2⁻), 4 (*Bdnf*-TMKI ChR2⁻), 7 (wild-type ChR2⁺) and 4 (*Bdnf*-TMKI ChR2⁺) mice. **e**, Survival curves of wild-type and *Bdnf*-TMKI mice bearing SU-DIPG-XIII-P* xenografts. $n = 7$ (wild type) and 8 (*Bdnf*-TMKI mice). **f**, Survival curves of mice bearing wild-type and *NTRK2*-KO orthotopic xenografts (SU-DIPG-VI and SU-pcGBM; $n = 7$ mice per group). **g**, Survival curves of SU-DIPG-XIII-P* xenografted mice treated with entrectinib versus vehicle-treated controls. Grey shading

indicates drug treatment. **h**, Representative images (left) and proliferation index (right; EdU⁺ cells/DAPI cells) of wild-type and *NTRK2*-KO glioma cultures (SU-DIPG-VI) with or without BDNF treatment ($n = 5$ coverslips per group). Scale bar, 100 μm. **i**, Representative images (left) and proliferation index (right; EdU⁺ cells/Nestin⁺ glioma cells) of wild-type and *NTRK2*-KO glioma (SU-DIPG-VI) cultured alone or with neurons ($n = 3$ coverslips per group). Scale bar, 50 μm. **j**, Proliferation index of SU-DIPG-VI wild-type and *NTRK2*-KO glioma co-culture with neurons (as in representative image in **i**), with or without NBQX ($n = 3$ coverslips per group; repeated in Extended Data Fig. 5a,b). **k–m**, Experimental scheme (**k**), Representative images (**l**) and quantification of proliferation rate (Ki67⁺ cells/HNA⁺ glioma cells) of wild-type and *NTRK2*-KO glioma xenografts (SU-DIPG-VI) treated with perampanel or vehicle control (**m**). $n = 6$ (wild type + vehicle), 7 (wild type + perampanel), 5 (*NTRK2*-KO + vehicle) and 6 (*NTRK2*-KO + perampanel) mice. Scale bar, 50 μm. Data are mean ± s.e.m. One-way ANOVA with Tukey's post hoc analysis (**d**,**h–j**,**m**); two-tailed log rank analysis (**e–g**). *$P < 0.05$, **$P < 0.01$, ***$P < 0.001$, ****$P < 0.0001$; NS, not significant.

engineered mouse model that is deficient in activity-induced expression of BDNF[32] (*Bdnf*-TMKI). This mouse model expresses baseline levels of BDNF ligand, but does not exhibit activity-regulated increases in BDNF expression and secretion owing to a loss of the CREB-binding site in the *Bdnf* promoter[32]. We expressed the excitatory, blue-light-gated opsin channelrhodopsin-2 in deep layer cortical projection neurons (Thy1::ChR2) in the *Bdnf-TMKI* mouse (Fig. 1a) to enable optogenetic stimulation of cortical projection (glutamatergic) neuronal activity. Patient-derived paediatric glioma (DIPG) cells were xenografted into the frontal cortex and subcortical white matter, and following a 5-week period of engraftment, cortical projection neuronal activity was optogenetically stimulated using our established protocol[1] (10-min session per day, 20 Hz blue-light stimulation with 30-s on/90-s off cycles) for 1 week. As expected[1], we observed an increase in glioma proliferation following optogenetic stimulation of cortical projection neuronal activity in *Bdnf* wild-type mice. The effects of cortical projection neuronal activity on glioma proliferation were markedly

attenuated in *Bdnf-TMKI* mice lacking activity-regulated BDNF expression and secretion (Fig. 1b–d and Extended Data Fig. 1c,d).

Given this contribution of activity-regulated BDNF to the proliferative influence of neuronal activity in the short term, we next probed the effect of activity-regulated BDNF on the survival of *Bdnf* wild-type and *Bdnf*-TMKI mice bearing patient-derived orthotopic paediatric glioma xenografts. We found that the loss of neuronal activity-regulated BDNF expression and secretion exerts a survival advantage in *Bdnf*-TMKI mice bearing patient-derived DIPG xenografts in the brainstem (Fig. 1e and Extended Data Fig. 1e), concordant with the hypothesis that activity-regulated BDNF signalling robustly influences glioma progression in the context of the brain microenvironment.

## Therapeutic targeting of TrkB

Genetic expression patterns of the neurotrophin receptors in DMG tumours (Extended Data Fig. 1a), suggests that BDNF acts on glioma

cells through the TrkB (encoded by *NTRK2*) receptor and that BDNF is a key neurotrophin to which paediatric glioma cells respond. Concordantly, the neurotrophins NGF and NT-3, which signal through TrkA and TrkC receptors, respectively, did not affect glioma cell proliferation in vitro. NT-4, a neurotrophin that also signals through TrkB, promotes glioma proliferation similarly to BDNF (Extended Data Fig. 1f). We therefore tested the effects of genetic or pharmacological TrkB blockade on growth of paediatric gliomas. We used CRISPR technology to delete *NTRK2* from human, patient-derived glioma cells (referred to as *NTRK2*-knockout (KO)). The knockout used a direct deletion in exon 1 of *NTRK2*, resulting in an approximately 80% decrease in TrkB protein levels (Extended Data Fig. 1g,h). Mice were xenografted orthotopically with patient-derived cells in which *NTRK2* was wild type (Cas9 control) or had been CRISPR-deleted (*NTRK2*-KO). Mice bearing orthotopic xenografts of *NTRK2*-KO DIPG in the brainstem or *NTRK2*-KO paediatric cortical glioblastoma in the frontal cortex exhibited a marked increase in overall survival compared with littermate controls xenografted with *NTRK2* wild-type cells (Fig. 1f and Extended Data Fig. 1i,j). Proliferation of *NTRK2*-KO glioma cells was similar in wild-type mice and in mice lacking activity-regulated BDNF (*Bdnf*-TMKI) following optogenetic stimulation of cortical projection neuronal activity, indicating that the loss of activity-regulated BDNF does not exert effects that are independent of glioma TrkB signalling (Extended Data Fig. 1k).

We next performed preclinical efficacy studies of pan-Trk inhibitors. Trk inhibitors have recently been developed for treatment of *NTRK*-fusion malignancies, including for *NTRK*-fusion infant gliomas[33–35]. Here, we tested the preclinical efficacy of these inhibitors in *NTRK* non-fusion gliomas such as DIPG. We first assessed the ability of entrectinib to cross the blood–brain barrier and found that systemic entrectinib (120 mg kg$^{-1}$, oral administration) reduced pharmacodynamic markers of TrkB signalling, including TrkB phosphorylation and downstream ERK phosphorylation in brain tissue (Extended Data Fig. 2a–c). Treatment of an aggressive patient-derived paediatric glioma (DIPG) orthotopic xenograft model with entrectinib increased overall survival compared to vehicle-treated controls (Fig. 1g and Extended Data Fig. 2d). Although entrectinib decreased the proliferation rate of xenografted *NTRK2* wild-type DIPG cells in vivo, it did not further decrease the proliferation rate of *NTRK2*-KO glioma xenografts (Extended Data Fig. 2e,f), demonstrating that the mechanism of action of entrectinib in DIPG is mediated through TrkB.

## BDNF regulates neuron–glioma interactions

We previously found that BDNF is one of multiple paracrine factors that can increase glioma proliferation in response to neuronal activity[1,3], albeit not as robustly as other neuron–glioma signalling mechanisms[1]. To confirm the relative contribution of activity-regulated BDNF ligand to the mitogenic effect of activity-regulated secreted factors, we optogenetically stimulated cortical explants from *Bdnf*-TMKI or *Bdnf* wild-type mice, collected conditioned medium, and tested the effects of conditioned medium on glioma cell proliferation in vitro using our well-validated experimental paradigm[1]. Exposure of patient-derived glioma cultures to conditioned medium from optogenetically stimulated *Bdnf* wild-type cortical explants increased tumour cell proliferation rate, as we have previously shown[1] (Extended Data Fig. 3a,b). Conditioned medium collected from optogenetically stimulated *Bdnf*-TMKI cortical explants elicited a mildly reduced proliferative response of glioma cells in monoculture compared with conditioned medium from wild-type cortical explants, indicating a small direct mitogenic effect of activity-regulated BDNF ligand secretion (Extended Data Fig. 3b), as expected[1].

Testing the effects of BDNF alone on glioma proliferation in vitro, we found that the addition of recombinant BDNF (100 nM) increases paediatric glioma (DIPG) cell proliferation from a rate of around 20% to around 30%. This effect is completely abrogated—as expected—with

CRISPR knockout of *NTRK2* and by pharmacological inhibition with entrectinib or larotrectinib (Fig. 1h and Extended Data Fig. 3c). A similarly modest increase in proliferation was observed in a range of patient-derived glioma monocultures exposed to BDNF, including thalamic DMG and paediatric cortical glioblastoma (Extended Data Fig. 3d).

Co-culture with neurons elicits a robust increase in glioma cell proliferation rate from around 20% to around 60%, underscoring the powerful effects of neurons on glioma proliferation that include neuroligin-3 (NLGN3) signalling and neuron-to-glioma synaptic mechanisms[1,3–5]. We sought to investigate the relative contribution of BDNF–TrkB signalling in neuron–glioma interactions using neuronal co-culture with *NTRK2* wild-type or *NTRK2*-KO glioma cells. In the absence of neurons, TrkB loss alone does not reduce paediatric glioma cell proliferation (Fig. 1i), consistent with the lack of BDNF ligand expression in paediatric glioma cells (Extended Data Fig. 1a). However, TrkB loss in glioma cells co-cultured with neurons resulted in a marked reduction in neuron-induced proliferation, decreasing the glioma cell proliferation rate from around 60% to around 30%. This reduction is disproportionate to the loss accounted for by BDNF mitogenic signalling alone, as described above (Fig. 1h). The magnitude of the change in glioma proliferation elicited by TrkB loss in response to BDNF ligand alone compared with that in the context of neuron co-culture (Fig. 1h,i) suggests that BDNF may have a more complex role in neuron–glioma interactions than simply as an activity-regulated growth factor.

To explore possible roles for BDNF–TrkB signalling in glioma pathophysiology, we examined gene-expression relationships between TrkB and other gene programmes at the single-cell level using available single-cell transcriptomic data from human H3K27M-mutated DMG primary biopsy tissue[36]. *NTRK2* is expressed in the majority of glioma cells at varying levels across the defined cellular subpopulations that comprise DMGs, including oligodendrocyte precursor cell-like tumour cells (OPC-like), astrocyte-like tumour cells (AC-like) and oligodendrocyte-like tumour cells (OC-like) (Extended Data Fig. 4a,b). As previously demonstrated[4], synaptic gene expression is enriched in the oligodendroglial compartments of the tumour (oligodendrocyte-like and oligodendrocyte precursor cell-like cellular subpopulations), whereas tumour microtube-associated gene expression is enriched in the astrocyte-like compartment (Extended Data Fig. 4c). Expression correlation analyses identified different patterns of genes in each cellular compartment that correlate with *NTRK2* expression (Extended Data Fig. 4d). Examples of genes that are strongly correlated with *NTRK2* in the astrocyte-like compartment include *GJA1*, *TTHY1*, *GRIK1* and *KCNN3*; *TTHY1* and *GJA1* are known to have crucial roles in tumour microtube formation and connectivity in adult high-grade gliomas[37,38]. In the OC-like compartment, *NTRK2* expression correlates with *NRXN2*, *NLGN3*, *CSPG4*, *PDGFRA*, *FGFR1*, *CNTN1*, *SLIT2*, *IGF1R* and *CACNG5*, and in the OPC-like compartment it correlates with *NRXN2*, *NRXN1*, *NLGN4X*, *SYT11*, *CREB5*, *SRGAP2C*, *CSPG4*, *ASCL1*, *PI3KR3*, *CDK6*, *EGFR* and *EPHB1* (Extended Data Fig. 4d). Gene Ontology analyses of these differentially correlated genes in each cellular sub-compartment revealed correlation of *NTRK2* with processes of synaptic communication and neural circuit assembly (Extended Data Fig. 4e–g). In the OPC-like compartment, *NTRK2* expression correlated with postsynaptic organization, axon guidance, neuronal projection guidance, neuronal migration, ERK signalling cascades and the AKT signalling cascade, consistent with the hypothesized role of TrkB in neuron-to-glioma synapses, consequent effects of AMPAR-mediated synaptic signalling on tumour migration[29] and expected signalling consequences of TrkB activation. In the OC-like compartment, the gene sets correlated with *NTRK2* expression involve synaptic organization, modulation of synaptic transmission, synaptic plasticity, and learning and memory. In the astrocyte-like compartment, which tends to engage in extensive tumour microtube connectivity[37], *NTRK2* expression correlated with genes involved in axon guidance and neuronal projection

morphogenesis. Together, these single-cell transcriptomic analyses support potential roles for TrkB signalling in neuron-to-glioma synaptic biology as well as glioma-to-glioma network formation, with TrkB correlated with distinct processes in astrocyte-like and oligodendroglial-like cellular subpopulations.

## Relationship between TrkB and AMPAR signalling

Glutamatergic neuron-to-glioma synapses are mediated by calcium-permeable AMPARs in both paediatric and adult gliomas, and robustly regulate glioma progression[4,5,29]. In the healthy brain, BDNF–TrkB signalling regulates glutamatergic synaptic transmission through several mechanisms[9–15,39]. To explore the hypothesis that the growth-promoting effects of activity-regulated BDNF–TrkB signalling in glioma involves modulation of synaptic biology, we explored whether the effects of glioma TrkB signalling are related to or independent of AMPAR signalling. We found that pharmacologically blocking AMPARs or genetically blocking TrkB through *NTRK2* knockout decreased tumour cell proliferation in vivo or in neuron–glioma co-culture (Fig. 1j–m and Extended Data Fig. 5a,b). However, we found no additive effect of blocking AMPARs and TrkB, suggesting a relationship between the mechanisms.

## Glioma glutamatergic current strength

We hypothesized that BDNF–TrkB signalling may function in glioma to strengthen neuron-to-glioma synapses. In healthy neurons, activity-regulated plasticity[23,24,40] of synaptic strength—the evoked amplitude of the postsynaptic current—dynamically modulates neural circuit function[41], and these synaptic changes are thought to underlie learning and memory[42]. One form of plasticity of synaptic strength involves increased AMPAR trafficking to the postsynaptic membrane[43,44]. Glutamatergic neurotransmission through NMDAR (N-methyl-D-aspartate receptor) and consequent calcium signalling can increase AMPAR trafficking to the postsynaptic membrane[17–22,45], but glioma cells do not strongly express NMDAR genes[4]. Another activity-regulated mechanism that can promote AMPAR trafficking to the membrane is BDNF–TrkB signalling and consequent stimulation of the CAMKII calcium signalling pathway[9–15]. Concordantly, inhibition of CAMKII reduces the proliferation-inducing effects of neuron–glioma co-culture (Extended Data Fig. 5c,d). We therefore tested the hypothesis that BDNF–TrkB signalling could induce plasticity of the malignant synapse—that is, it could increase the amplitude of glioma excitatory postsynaptic currents (EPSCs).

We performed whole-cell patch clamp electrophysiology of glioma cells xenografted to the hippocampus in an acute slice preparation[4] (Fig. 2a and Extended Data Fig. 6a). In response to transient and local glutamate application, and in the presence of tetrodotoxin to block indirect effects from neuronal action potentials, voltage-clamp recordings of xenografted glioma cells demonstrated inward currents that were blocked by the AMPAR blocker NBQX, and not by the NMDAR inhibitor AP-5 or the glutamate transporter inhibitor DL-threo-β-benzyloxyaspartate (TBOA) (Extended Data Fig. 6b,c), as expected[4,5]. As a control for mechanical effects of glutamate application by local puff, we found no inward currents following application of medium only (Extended Data Fig. 6d). Perfusion of BDNF over the slice preparation increased the amplitude of the glutamate-evoked currents (Fig. 2b,c). Confirming that glioma cell TrkB activation mediates the change in glutamate-evoked current amplitude, *NTRK2* knockout prevented the BDNF-induced increase in glutamate-evoked inward current amplitude, compared with *NTRK2* wild-type glioma xenografted cells (Fig. 2b,c). Consistent with the hypothesis that BDNF functions to modulate synaptic strength through the CAMKII calcium signalling pathway, the CAMKII inhibitor KN-93 blocks the effect of BDNF on glioma glutamatergic current amplitude, whereas the inactive

analogue KN-92 does not (Fig. 2d,e). For comparison, and as a control to assess the specificity of BDNF among other known neuronal activity-regulated paracrine factors[1–3], soluble NLGN3 was similarly tested and exerted no acute effect on glutamatergic current amplitude in glioma cells (Extended Data Fig. 6e,f). Together, these findings demonstrate that the BDNF–TrkB signalling pathway modulates the strength of glutamate-evoked currents in glioma cells.

Glutamatergic signalling in gliomas can be synaptic and extrasynaptic. To test the effects of BDNF on neuron-to-glioma synaptic currents, we stimulated the axonal afferents (Schaffer collaterals) into the CA1 region of the hippocampus in which glioma cells were engrafted (Fig. 2f). Axonal stimulation produced EPSCs in around 10% of xenografted glioma cells examined (5 out of 43 cells), consistent with the proportion of tumour cells that exhibit neuron-to-glioma synapses in similar experimental paradigms[4,5]. Glioma EPSCs increased in amplitude following exposure to BDNF (Fig. 2g,h). Together, these data illustrate that BDNF increases the strength of glutamate-evoked currents, including at AMPAR-mediated neuron-to-glioma synapses.

To explore the intracellular consequences of BDNF-induced current amplification we performed in situ calcium imaging of xenografted glioma cells that express the genetically encoded calcium indicator GCaMP6s (Fig. 2i and Extended Data Fig. 6g). As expected[4,5], local glutamate application induced calcium transients in glioma cells (Fig. 2j–l and Extended Data Fig. 6h–n). The intensity and duration of glutamate-evoked calcium transients were increased by BDNF exposure in two distinct patient-derived models of paediatric glioma (Fig. 2i–l and Extended Data Fig. 6j–n). Glioma calcium transients evoked by glutamate were blocked by the AMPAR blocker NBQX (Extended Data Fig. 6m,n), consistent with the known role for AMPARs mediating glutamatergic signalling in glioma[4,5] (Extended Data Fig. 6b,c).

## BDNF regulates AMPAR trafficking in glioma

In healthy neurons, BDNF–TrkB signalling increases the trafficking of AMPARs to the postsynaptic membrane via the CAMKII calcium signalling pathway[12,46] (Fig. 3a). Given the findings above that BDNF increased AMPAR-mediated currents and calcium transients in glioma, we next investigated the effect of BDNF on AMPAR trafficking to the cell membrane in glioma cells. Glioma cells express the GluA2, GluA3 and GluA4 AMPAR subunits[4,5]. As described below, levels of each of these subunits were increased at the cell membrane with BDNF exposure (Fig. 3). To examine the effects of BDNF on GluA4 and GluA3 subunits, glioma cell surface proteins were captured using biotinylation with avidin pull-down and probed for levels of AMPAR subunits. Consistent with the hypothesis that BDNF increases AMPAR trafficking to the glioma cell membrane, BDNF exposure increased glioma cell surface levels of the AMPAR subunits GluA3 and GluA4 compared with vehicle-treated control glioma cells (Fig. 3b–e). Examining the time course of increased AMPAR subunit levels at the cell membrane, GluA4 levels were increased by 5 min, peaked at 15 min, and remained elevated at 30 min following onset of BDNF exposure (Fig. 3b,c). As a control for other neuronal activity-regulated paracrine factors, we found that soluble NLGN3 did not evoke a change in GluA4 membrane levels (Fig. 3f,g).

To examine GluA2 subunit trafficking, we leveraged pHluorin technology for live imaging of AMPAR trafficking within glioma cells co-cultured with neurons. We expressed the GluA2 AMPAR subunit tagged to a pH-sensitive GFP[47] (pHluorin) in glioma cells and then performed high-resolution confocal live imaging of these AMPARs in patient-derived glioma cultures. Glioma cells express calcium-permeable AMPARs[4,5], so we generated a pHluorin-tagged calcium-permeable isoform (GluA2(Q)). We also expressed PSD95 tagged to RFP in glioma cells to confirm localization of GluA2 to the glioma postsynaptic site. Super-ecliptic pHluorins (SEPs) fluoresce when the N-terminus of the subunit moves from the acidic pH in the

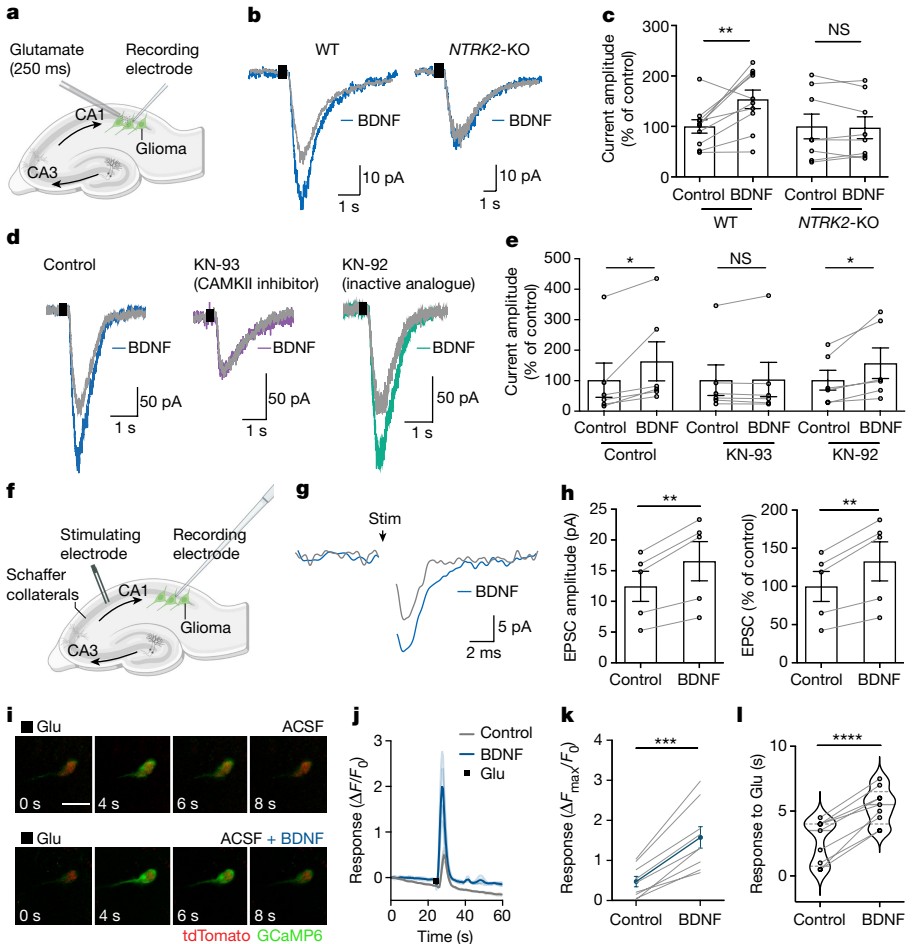

**Fig. 2 | BDNF–TrkB signalling increases the amplitude of glutamatergic currents in glioma cells. a**, Electrophysiological model; GFP+ glioma xenografted in hippocampal CA1; local glutamate puff. **b**, Representative traces of glutamate-evoked (black rectangle) inward currents in xenografted SU-DIPG-VI glioma before (grey) and after (blue) 30 min of BDNF perfusion in wild-type and *NTRK2*-KO glioma. **c**, Quantification of current amplitude in **b** ($n = 10$ cells, 6 mice (wild type) and 8 cells, 6 mice (*NTRK2*-KO)). **d**, Representative traces of xenografted SU-DIPG-VI in response to glutamate puff (black rectangle) before (grey traces) and after (blue, purple, and green traces) 30 min of BDNF perfusion. Left, control trace. Middle, with 2-h incubation with the CAMKII inhibitor KN-93. Right, with 2-h incubation with KN-92, an inactive analogue of KN-93. **e**, Quantification of current amplitude in **d** ($n = 6$ cells per group from 5 mice (control), 3 mice (KN-93-treated) and 3 mice (KN-92-treated)). **f**, Electrophysiological model as in **a**, with Schaffer collateral afferent stimulation. **g**, Representative averaged voltage-clamp traces of evoked glioma excitatory postsynaptic current (EPSC) in response to axonal stimulation (black arrow) before (grey) and after (blue) BDNF application. **h**, Quantification of EPSC amplitude in **g** ($n = 5$ out of 43 glioma cells exhibiting EPSCs from 4 mice). **i**, Representative two-photon in situ imaging (8-s time series) of glioma cell calcium transients evoked by local glutamate puff before (top) and after (bottom) perfusion with BDNF (100 ng ml$^{-1}$, 30 min). Green denotes glioma GCaMP6s fluorescence and red denotes tdTomato nuclear tag. Scale bar, 10 μm. **j**, GCaMP6s intensity traces of SU-DIPG-XIII-FL glioma cells response to glutamate puff with or without BDNF. $n = 4$ individual cells per group. Light grey shows traces for individual vehicle-treated cells and dark grey shows the average; light blue shows traces for individual BDNF-treated cells and dark blue shows the average. **k**, Responses of SU-DIPG-XIII-FL GCaMP6s cells to glutamate puff with or without BDNF ($n = 9$ cells, 3 mice). **l**, Duration of calcium transient in response to glutamate puff in SU-DIPG-XIII-FL GCaMP6s cells before and after BDNF exposure ($n = 9$ cells, 3 mice). Data are mean ± s.e.m. Two-tailed paired Student's *t*-test (**c**,**e**,**h**,**k**,**l**).

trafficking vesicle to the neutral pH on the outside of the cell membrane (Fig. 3h). To validate the pHluorin strategy, we confirmed that exposure of glioma cells expressing GluA2(Q)–SEP and PSD95–RFP to acidic medium (pH 5.5) quenched the signal, demonstrating that the majority of the AMPAR fluorescent signal at the postsynaptic puncta is from plasma membrane-bound GluA2 (Fig. 3i,j). Time-course imaging of individual puncta demonstrated that BDNF exposure elicits an increase in the postsynaptic levels of GluA2 on the glioma cells (Fig. 3k–m), on a time-scale consistent with the increased GluA3 and GluA4 trafficking (Fig. 3b–e) and the change in glutamate-evoked currents (Fig. 2). Together, these findings indicate that BDNF–TrkB signalling increases trafficking of AMPAR subunits to the cell membrane, accounting for the increased glutamate-evoked currents described above.

We next explored the signalling mechanisms of BDNF in patient-derived paediatric glioma cells. Several signalling pathways are known to be activated upon BDNF binding to the TrkB receptor, and the expression of different TrkB splice variants can alter the function of the receptor[48]. Using available single-cell transcriptomic data[4,36], we compared the expression of the *NTRK2* splice variants and found that although DMGs do express full-length TrkB, DMG cells exhibit higher expression of truncated TrkB, as has been previously described in adult glioblastoma[49] (Extended Data Fig. 7). Consistent with the transcriptomic analyses (Extended Data Fig. 4) and the role for CAMKII in BDNF-evoked changes in glutamatergic current strength (Fig. 2d,e) discussed above, western blot analysis of glioma cells demonstrates that BDNF exposure activates three main signalling cascades:

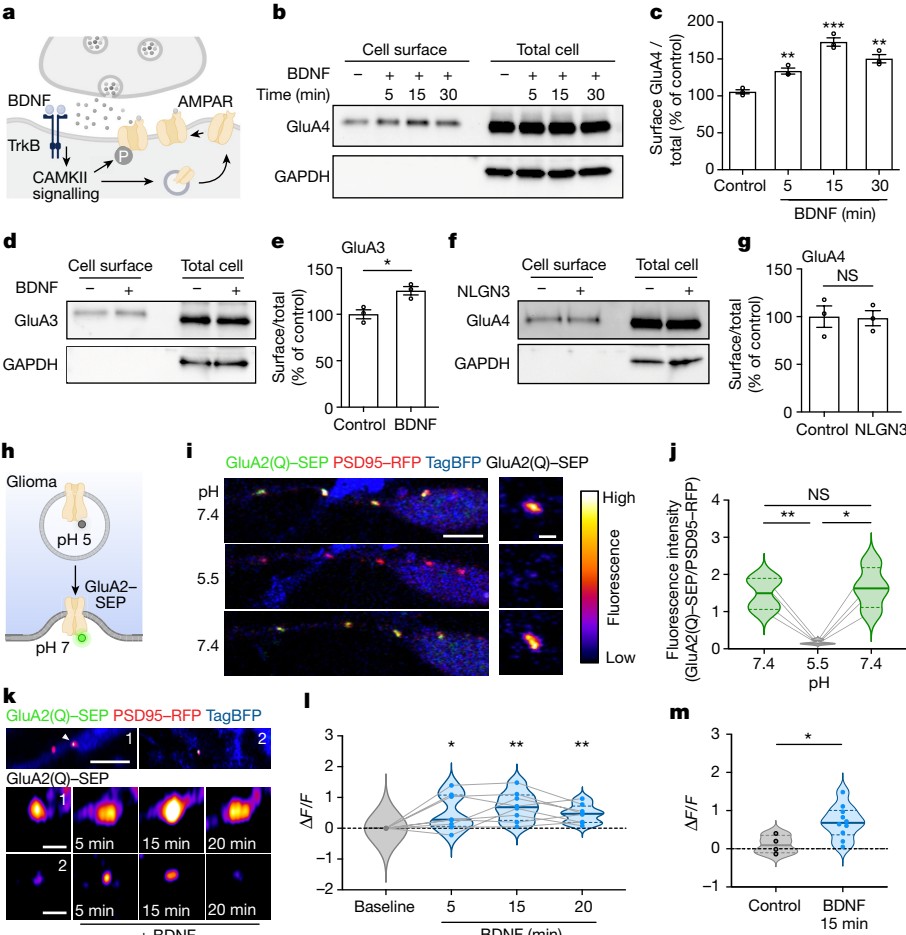

**Fig. 3 | BDNF regulates trafficking of AMPAR to the glioma postsynaptic membrane. a**, Schematic depicting AMPAR trafficking downstream of BDNF–TrkB–CAMKII signalling[46]. **b**, Western blot analysis of cell surface and total cell protein levels of GluA4 in SU-DIPG-VI glioma with or without BDNF treatment for 5, 15 and 30 min. **c**, Quantification of cell surface GluA4 in **b** ($n = 3$ independent biological replicates). **d**, Western blot analysis of cell surface and total cell protein levels of GluA3 in SU-DIPG-VI glioma with or without BDNF treatment for 30 min. **e**, Quantification of cell surface GluA3 in **d** ($n = 3$ independent biological replicates). **f**, Western blot analysis of cell surface and total cell protein levels of GluA4 in SU-DIPG-VI cells treated with NLGN3 for 30 min. **g**, Quantification of cell surface GluA4 data in **f** ($n = 3$ independent biological replicates). **h**, Schematic showing GluA2–SEP experiments. **i**,**j**, Validation of pHluorin approach. **i**, Left, representative images of a glioma cell process expressing GluA2(Q)–SEP, PSD95–RFP and whole-cell TagBFP in co-culture

with neurons. Right, representative GluA2(Q)–SEP puncta. Scale bars, 5 μm (left) and 1 μm (right). Cells were exposed to pH 7.4 followed by pH 5.5 and then pH 7.4. **j**, Quantification of fluorescence intensity of GluA2(Q)–SEP puncta before, during and after acidic exposure ($n = 4$ puncta from a representative cell). **k**, Top, representative images of two processes from glioma cells expressing GluA2(Q)–SEP, PSD95–RFP and TAG-BFP2 in co-culture with neurons (scale bar, 5 μm). Middle and bottom, representative images of GluA2(Q)–SEP puncta at 0, 5, 15 and 20 min of BDNF incubation (scale bar = 1 μm). **l**, Fluorescence intensity of co-localized GluA2(Q)–SEP:PSD95–RFP puncta over time with BDNF treatment ($n = 8$ puncta, 6 cells). **m**, Fluorescence intensity of co-localized GluA2(Q)–SEP:PSD95–RFP puncta after 15 min versus basal fluorescence in control (vehicle, $n = 4$ puncta, 2 cells) or BDNF-treated cells ($n = 8$ puncta, 6 cells). Data are mean ± s.e.m. Two-tailed unpaired Student's $t$-test (**c**,**e**,**g**,**m**); two-tailed paired Student's $t$-test (**j**); two-tailed one-sample $t$-test (**l**).

MAPK–ERK, PI3K–AKT and CAMKII calcium signalling (Extended Data Fig. 8a–d). In neurons, BDNF increases AMPAR trafficking via CAMKII calcium signalling[12], which has crucial roles in neuronal synaptic plasticity[20,50]. MAPK and PI3K have also been shown to have a role in AMPAR-mediated synaptic transmission[51]. Post-translational modifications induced by BDNF–TrkB signalling—such as phosphorylation—regulate the trafficking of AMPAR subunits[52]. We found that BDNF exposure increases phosphorylation of the subunit GluA4 at Ser862 in glioma cells (Extended Data Fig. 8e–h), a site that is known to facilitate the delivery of the subunit to the postsynaptic density in neurons[53]. Treatment with the pan-Trk inhibitor entrectinib abrogated this BDNF-induced increase in GluA4 phosphorylation in glioma cells (Extended Data Fig. 8g,h).

In contrast to these protein phosphorylation and trafficking effects of BDNF, paediatric glioma cells exhibited few gene-expression changes in response to BDNF exposure, with the exception of *VGF* (Extended

Data Fig. 9), a gene that is known to be regulated by BDNF in adult glioblastoma[31].

## Plasticity of synaptic connectivity

A subset of glioma cells engage in synapses[4,5] and accordingly, we observed a subset of xenografted glioma cells that exhibit an inward current in response to glutamate using patch clamp electrophysiology (Fig. 4a,b). *NTRK2*-KO tumours exhibited fewer cells that responded to glutamate (Fig. 4a,b). We hypothesized that *NTRK2* loss in glioma cells may alter the degree of malignant synaptic connectivity.

To investigate whether BDNF–TrkB signalling regulates the number of neuron-to-glioma synaptic connections, we performed immuno-electron microscopy in the brains of mice bearing wild-type and *NTRK2*-KO patient-derived glioma xenografts expressing GFP.

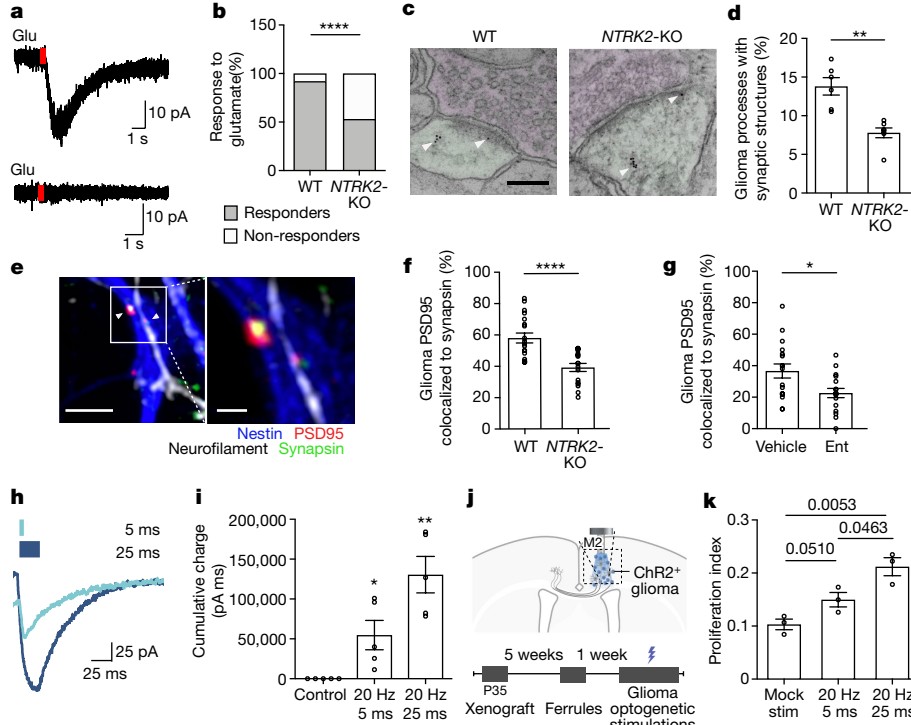

**Fig. 4 | Plasticity of neuron–glioma connectivity and functional effects of increasing synaptic strength.** **a**, Representative traces of glutamate-evoked (red rectangle) currents in xenografted glioma (SU-DIPG-VI). Top, glutamate responder cell. Bottom, non-responder cell. **b**, Quantification of glutamate responders and non-responders ($n = 25$ *NTRK2* wild-type glioma cells from 13 mice and 15 *NTRK2*-KO glioma cells from 7 mice). **c**, Immuno-electron microscopy of GFP⁺ wild-type and *NTRK2*-KO SU-DIPG-VI cells xenografted into mouse hippocampus. Arrowheads denote immunogold labelling of GFP (glioma). Presynaptic neurons are shaded magenta and glioma cells are shaded green. Scale bar, 2 μm. **d**, Quantification of neuron-to-glioma synapses in **c** ($n = 6$ wild-type and 7 *NTRK2*-KO glioma xenografted mice). **e**, Representative images of neuron–glioma co-culture with PSD95–RFP-expressing *NTRK2* wild-type and *NTRK2*-KO glioma cells. Neurofilament (axon, white), nestin (glioma, blue), synapsin (presynaptic puncta, green) and PSD95–RFP (glioma postsynaptic puncta, red) are labelled. Scale bars: 4 μm (left), 1 μm (right). **f**, Colocalization of postsynaptic glioma-derived PSD95–RFP with neuronal presynaptic synapsin in neuron-glioma co-cultures of *NTRK2* wild-type ($n = 19$ cells, 12 coverslips from 3 independent experiments) or *NTRK2*-KO glioma (SU-DIPG-VI, $n = 17$ cells, 12 coverslips from 3 independent experiments). **g**, Colocalization of postsynaptic glioma-derived PSD95–RFP with neuronal presynaptic synapsin in neuron–glioma co-cultures (SU-DIPG-VI) treated with vehicle or entrectinib (Ent) ($n = 18$ cells per group, 6 coverslips per group from 3 independent experiments). **h**, Electrophysiological trace of ChR2⁺ glioma cells (SU-DIPG-XIII-FL) in response to 5-ms (light blue) or 25-ms (dark blue) light-pulse width optogenetic stimulation. **i**, Quantification of total accumulated charge upon 2 s of optogenetic stimulation with 5-ms or 25-ms light-pulse width as shown in **h**, compared with no blue light ($n = 5$ glioma cells per group). **j**, Optogenetic model for stimulation of xenografted ChR2⁺ glioma. **k**, Proliferation index (Ki67⁺ cells/HNA cells) of xenografted ChR2⁺ glioma cells (SU-DIPG-XIII-FL) after mock or blue-light stimulation at 5-ms or 25-ms light-pulse width ($n = 3$ mice per group). Data are mean ± s.e.m. Two-sided Fisher's exact test (**b**); two-tailed unpaired Student's *t*-test (**d,f,g,k**); two-tailed one-sample *t*-test (**l**).

Using immuno-electron microscopy with immunogold labelling of GFP⁺ cells to unambiguously identify the malignant cells, we identified fewer neuron-to-glioma synaptic structures in the *NTRK2*-KO tumours compared with wild-type tumours (Fig. 4c,d and Extended Data Fig. 10a). To further test whether BDNF–TrkB signalling regulates the number of neuron-to-glioma synaptic connections, we co-cultured glioma cells—with or without *NTRK2* expression and expressing RFP-tagged PSD95—with neurons and quantified neuron-to-glioma synaptic puncta. Co-culture of *NTRK2*-KO glioma cells exhibited fewer synaptic structures with neurons evident as co-localized neuronal presynaptic puncta (synapsin) with glioma postsynaptic puncta (PSD95–RFP), compared with *NTRK2* wild-type glioma cells (Fig. 4e,f). Replicating this experiment using an orthogonal approach, we knocked down *NTRK2* expression using short hairpin RNA (shRNA) rather than CRISPR-mediated deletion, demonstrating a similar reduction in synaptic puncta in TrkB-deficient glioma cells (Extended Data Fig. 10b–d). Reduction in neuron-to-glioma synaptic structures was also seen with addition of entrectinib to neuron–glioma co-cultures (Fig. 4g). Together, these data demonstrate that BDNF–TrkB signalling modulates neuron-to-glioma synaptic connectivity.

## Effect of glioma synaptic plasticity

AMPAR-mediated synapses promote glioma growth and progression[4,5,29]. These growth-promoting effects are mediated in part by synaptic signalling-induced membrane depolarization, which alone is sufficient to drive glioma proliferation[4]. We next tested whether the magnitude of the depolarizing current differentially promotes cancer cell proliferation, which could potentially explain the role of synaptic plasticity in glioma growth. Using an optogenetic strategy to mimic varying degrees of glioma synaptic strength, we can control the amplitude of glioma membrane depolarization by applying a differing duration of the blue-light pulse (light-pulse width) to patient-derived glioma cells expressing the blue-light-sensitive cation channel ChR2. Applying a light-pulse width of 5 ms or 25 ms results in glioma cell membrane depolarization amplitude that increases with duration of the light pulse (Fig. 4h,i and Extended Data Fig. 10e,f). In vivo depolarization of ChR2-expressing glioma xenografts at a consistent frequency and light power with varying light-pulse widths to mimic varying synaptic strengths demonstrates increasing glioma proliferation rate with increasing magnitude of glioma membrane depolarization (Fig. 4j,k and Extended Data Fig. 10g). Together, these data link malignant synaptic plasticity to glioma pathophysiology.

## Discussion

Neurons form synapses with glioma cells via calcium-permeable AMPARs[4,5,29] and the consequent membrane depolarization promotes glioma progression through voltage-sensitive mechanisms that remain to be fully unravelled[4,6,29]. Tumour cells form a network with each other through long processes called tumour microtubes connected by gap junctions[37,38,54], and neuron–glioma electrochemical communication propagates through the glioma network via this gap-junctional coupling[4,5] such that a single neuron-to-glioma synapse may affect numerous glioma cells. Here we found that malignant synapses exhibit plasticity of both strength and number, and that greater depolarizing current amplitude results in greater effects on glioma growth. Increased AMPAR trafficking to the glioma cell membrane mediates this plasticity of synaptic strength, recapitulating a mechanism of synaptic plasticity that is operational in healthy neurons and contributes to learning and memory[23-26,42]. In neurons, AMPAR subunit composition influences receptor structure and electrophysiological properties[55,56], and it remains to be determined whether varying the subunit composition contributes to variations in glioma AMPAR-mediated currents. Whether other mechanisms of synaptic plasticity[41] occur in glioma remain to be determined in future work.

Neuronal activity promotes glioma progression through paracrine[1-3,28] and synaptic[4,5,29] signalling mechanisms. Our findings here illustrate that neuronal activity-regulated paracrine factors both directly promote glioma growth[1,3,31,49] and further reinforce neuron–glioma interactions. Two key activity-regulated paracrine factors, NLGN3 and BDNF, each promote neuron-to-glioma synaptic interactions in distinct ways: NLGN3 promotes the expression of genes encoding the AMPAR subunits GluA2 (*GRIA2*) and GluA4 (*GRIA4*) as well as TrkB[2] (*NTRK2*), whereas BDNF–TrkB signalling promotes trafficking of translated AMPAR subunits to the postsynaptic membrane to modulate the strength (amplitude) of postsynaptic currents. Both NLGN3[4] and BDNF promote neuron-to-glioma synapse formation. This potential for plasticity of malignant synaptic strength and connectivity raises several questions about the evolution of the neuron-to-glioma network over the disease course. It may be possible that certain experiences and activity patterns contribute to the neuroanatomical location of disease progression and that this activity-dependent reinforcement of neuron–glioma interactions and increased synaptic integration contributes to treatment resistance later in the disease course. Recent work examining gene-expression signatures in primary and recurrent glioma samples from the same individual patients suggests that synaptic signatures do indeed increase with disease progression[57]; the findings here indicate that such increased tumour synaptic biology can reflect neuronal activity-dependent hijacking of adaptive plasticity mechanisms. Limiting malignant network elaboration by targeting malignant synaptogenesis and plasticity may be crucial for disease control, a concept supported by the therapeutic potential of disrupting BDNF–TrkB signalling, as demonstrated here. These findings provide a rationale for expanding the potential therapeutic use of TRK inhibitors—which are already showing clinical promise in *NTRK*-fusion malignancies[33-35]—to also include non-*NTRK*-fusion gliomas.

Gliomas hijack processes of neural plasticity and integrate into neural networks in complex and dynamic ways, leveraging mechanisms that normally regulate neural circuit establishment during development and ongoing neural plasticity that contributes to cognition in the healthy brain. Understanding and targeting neural circuit mechanisms in glioma may be critical for the effective treatment of these deadly brain cancers.

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

# Methods

## Patient-derived glioma cell models

Patient-derived glioma cultures were derived as described previously[1,58] with informed consent under a protocol approved by the Stanford University Institutional Review Board (IRB). Patient-derived glioma models used include diffuse intrinsic pontine glioma (DIPG): SU-DIPG-VI, SU-DIPG-XIII-FL (derived from tumour in frontal lobe), SU-DIPG-XIII-P (tumour cultured from the pons of the same patient), SU-DIPG-21, SU-DIPG-25; thalamic DMG: QCTB-R059 and paediatric hemispheric glioblastoma: SU-pcGBM2. All cultures are monitored by short tandem repeat (STR) fingerprinting for authenticity throughout the culture period and mycoplasma testing was routinely performed. Characteristics of the glioma models used (patient information, molecular characterization, and other characteristics) have been previously reported[1,59,60].

Glioma cultures are grown as neurospheres (unless otherwise stated) in serum-free medium consisting of DMEM (Invitrogen), Neurobasal(-A) (Invitrogen), B27(-A) (Invitrogen), heparin (2 ng ml$^{-1}$), human-bFGF (20 ng ml$^{-1}$) (Shenandoah Biotech), human-bEGF (20 ng ml$^{-1}$) (Shenandoah Biotech), human-PDGF-AA (10 ng ml$^{-1}$) (Shenandoah Biotech), human-PDGF-BB (10 ng ml$^{-1}$) (Shenandoah Biotech). The spheres were dissociated using TrypLE (Gibco) for seeding of in vitro experiments.

## Neuron–glioma co-culture

For synaptic puncta assays, neurons were isolated from CD1 mice (The Jackson Laboratory) at P0-P1 using he Neural Tissue Dissociation Kit−Postnatal Neurons (Miltenyi), and followed by the Neuron Isolation Kit, Mouse (Miltenyi) per manufacturer's instructions. After isolation 200,000 neurons were plated onto circular glass coverslips (Electron Microscopy Services) pre-coated with poly-L-lysine (Sigma) and mouse laminin (Thermo Fisher Scientific) as described previously[4]. Neurons were cultured in BrainPhys neuronal medium containing B27 (Invitrogen), BDNF (10 ng ml$^{-1}$, Shenandoah Biotech), GDNF (5 ng ml$^{-1}$, Shenandoah Biotech), TRO19622 (5 μM; Tocris) and β-mercaptoethanol (Gibco). The medium was replenished on days in vitro (DIV) 1 and 3. On DIV 5, fresh medium was added containing 50,000 glioma cells expressing PSD95–RFP for synaptic puncta experiments and incubated for 72 h. The PSD95-RFP-expressing glioma culture (SU-DIPG-VI) was previously described[4] and used to knockdown NTRK2 by shRNA. The SU-DIPG-VI wild-type and NTRK2-KO glioma cultures were transduced with the PSD95-RFP-PURO construct (see 'Cloning constructs'). For EdU proliferation assays, 70,000 wild-type or NTRK2-KO glioma cells were plated and incubated for 48 h, before treatment with EdU (10 μM) with or without the AMPAR blocker NBQX (10 μM, Tocris) and incubated for a further 24 h. Following incubation, the cultures were fixed with 4% paraformaldehyde (PFA) for 20 min at room temperature and stained for immunofluorescence analysis. For EdU analysis, cells were stained using the Click-iT EdU Cell Proliferation kit (Thermo Fisher Scientific, C10337), before staining with primary antibodies mouse anti-nestin (1:500; Abcam, ab6320) and chicken anti-neurofilament (M + H, 1:1000; Aves Labs, NFM and NFH), or mouse anti-human nuclei clone 235-1 (1:250; Millipore, MAB1281) and rabbit anti-microtubule-associated protein 2 (MAP2; 1:500, EMD Millipore, AB5622), overnight at 4 °C. Following washing, slips were incubated in secondary antibodies, Alexa 488 donkey anti-mouse pig IgG and Alexa 647 donkey anti-chicken IgG, or Alexa 488 donkey anti-mouse IgG and Alexa Fluor 647 donkey anti-rabbit (all 1:500, Jackson Immuno Research) and mounted using ProLong Gold Mounting medium (Life Technologies). Images were collected on a Zeiss LSM800, and proliferation index determined by quantifying percentage EdU-labelled glioma cells over total glioma cells (either nestin or HNA immunopositivity to identify glioma cells).

## Mice and housing conditions

All animal experiments were conducted in accordance with protocols approved by the Stanford University Institutional Animal Care and Use Committee (IACUC) and performed in accordance with institutional guidelines. Animals were housed according to standard guidelines with unlimited access to water and food, under a 12 h light: 12 h dark cycle, a temperature of 21 °C and 60% humidity. For brain tumour xenograft experiments, the IACUC has a limit on indications of morbidity (as opposed to tumour volume). Under no circumstances did any of the experiments exceed the limits indicated and mice were immediately euthanized if they exhibited signs of neurological morbidity or if they lost 15% or more of their initial body weight.

For the Bdnf-TMKI mice (C57BL/6 J background), knock-in mutations in three calcium regulatory element binding sites in the Bdnf promoter 4: CaRE1, CaRE2 and CaRE3−CRE (gift from M. Greenberg[32]) were bred to Thy1::ChR2$^{+/-}$ mice (line 18, The Jackson Laboratory, C57BL/6 J background) to produce the Bdnf-TMKI; Thy1::ChR2$^{+/-}$ genotype. These mice were then intercrossed with NSG mice (NOD-SCID-IL2R gamma chain-deficient, The Jackson Laboratory) to produce a Bdnf-TMKI; Thy1::ChR2$^{+/-}$; NSG genotype to facilitate to facilitate orthotopic xenografting.

## Orthotopic xenografting

For all xenograft studies, NSG mice (NOD-SCID-IL2R-gamma chain-deficient, The Jackson Laboratory) were used. Male and female mice were used in cohorts equally. For electrophysiological, immuno-electron microscopy and calcium imaging experiments, a single-cell suspension from patient-derived DIPG cultures SU-DIPG-VI and SU-DIPG-XIII-FL was injected into the hippocampal region. For survival analysis and proliferation immunohistological assays, patient-derived DIPG cultures (SU-DIPG-VI and SU-DIPG-XIII-p*) were injected into the pontine region and for patient-derived H3WT paediatric cortical glioblastoma (SU-pcGBM2), cells were injected into the cortex. For optogenetic stimulation, SU-DIPG-VI cells were stereotactically injected into wild-type and Bdnf-TMKI mouse premotor (M2) frontal cortex; ChR2$^+$ SU-DIPG-XIII-FL cells were similarly stereotactically injected into the M2 cortex. A single-cell suspension of all cultures was prepared in sterile culture medium (see 'Cell culture') immediately before surgery. Animals at P28–P35 were anaesthetized with 1−4% isoflurane and placed on stereotactic apparatus. Under sterile conditions, the cranium was exposed via a midline incision and a 31-gauge burr hole made at exact coordinates. For hippocampal injections the coordinates were as follows: 1.5 mm lateral to midline, 1.8 mm posterior to bregma, −1.4 deep to cranial surface. For pontine injections coordinates were: −1mm lateral to midline, 0.8 mm posterior to lambda, −5 deep to cranial surface. For cortical injections coordinates were: 1 mm anterior to bregma, 0.5 mm lateral to midline, 1.4 mm deep (for survival) and 1 mm deep (for optogenetics) to cranial surface. Cells were injected using a 31-gauge Hamilton syringe at an infusion rate of 0.4 μl min$^{-1}$ with a digital pump. At completion of infusion, the syringe needle was allowed to remain in place for a minimum of 2 min, then manually withdrawn. The wound was closed using 3 M Vetbond (Thermo Fisher Scientific) and treated with Neo-Predef with Tetracaine Powder.

## Fibre-optic placement and in vivo optogenetic stimulation

Experiments for the chronic neuronal optogenetic stimulation paradigm of glioma xenografts were performed as previously described[1]. In brief, a fibre-optic ferrule (Doric Lenses) was placed at M2 of the right hemisphere with the following coordinates: 1.0 mm anterior to bregma, 0.5 mm lateral to midline, −0.7 mm deep to the cranial surface at twelve weeks post glioma xenograft. Following seven days of recovery, all mice were connected to a 473 nm diode-pumped solid-state (DPSS) laser system via a mono fibre patch cord. In awake mice, pulses of light with a power measured at 10 mW were administered at a frequency

of 20 Hz for a period of 30 s, followed by 90 s recovery in a repeated cycle for a total of 10 min per day, for 7 consecutive days. The mice were monitored for their unidirectional ambulation response to light stimulation, confirming correct ferrule placement over M2 of the right hemisphere and effective neuronal stimulation. Animals confirmed as ChR2-negative had no response to light stimulation. Mice were sacrificed 24 h following the final stimulation session. Experiments for the chronic optogenetic stimulation paradigm of ChR2[+] glioma xenografts were performed as previously described[4]. In brief, ChR2[+] (pLV-ef1-ChR2(H134R)-eYFP WPRE) or control (pLV-ef1-eYFP WPRE) constructs were lentivirally transduced into SU-DIPG-XIII-FL cells. Five weeks after tumour engraftment, a fibre-optic ferrule was placed as described above and previously[1,4]. After a 7-day recovery, the surgically placed ferrules were connected to a 100-mW 473-nm DPSS laser system and received either 5 ms or 25 ms pulses of light (~5 mW output, 3–30 mW cm$^{-2}$ light density in the analysed region) at a frequency of 20 Hz over 30 s, followed by 90 s rest for a total of 10 min per day for 5 consecutive days. The mice were euthanized 24 h after the 5th stimulation.

## Survival studies

For survival studies, mice were xenografted at 4 to 5 weeks of age with the cultures SU-DIPG-VI (wild type and *NTRK2*-KO), SU-pcGBM2 (wild type and *NTRK2*-KO) and SU-DIPG-XIII-P*. After xenografts, mice were continuously monitored for signs of neurological deficits or health decline. For inhibitor treatment, SU-DIPG-XIII-P* was treated with 120 mg kg$^{-1}$ orally daily of entrectinib (HY-12678, MedChemExpress, 7% DMSO (Sigma), 10% Tween 80 (Sigma) in sterile H$_2$O) for 14 days, starting at 2 weeks post-xenograft. Morbidity criteria were a 15% reduction in weight or severe neurological motor deficits consistent with brain dysfunction (brainstem tumours exhibited circling and barrel rolls, cortical tumours displayed seizures and loss of gait). Statistical analyses were performed with Kaplan–Meier survival analysis using log rank testing.

## Mouse drug treatment studies

For all drug studies to assess proliferation index of xenografted glioma cells, NSG mice were xenografted as above and blind randomized to a treatment group. Four weeks post-xenograft of SU-DIPG-VI wild-type or *NTRK2*-KO glioma cells, mice were treated with oral administration of the AMPAR blocker perampanel (5 mg kg$^{-1}$; Adooq Biosciences; formulated in 10% DMSO, 60% PEG300, 30% water) via oral gavage for three weeks (5 days per week) and controls treated with equivalent volume of vehicle. Similarly, four weeks post-xenograft of SU-DIPG-VI wild-type or *NTRK2*-KO glioma cells, mice were treated with oral administration with the pan-Trk inhibitor entrectinib (120 mg kg$^{-1}$; HY-12678, MedChemExpress, 7% DMSO (Sigma), 10% Tween 80 (Sigma) in sterile H$_2$O) for 15 days and controls with equivalent volume of vehicle. For immunohistological analysis of glioma cell proliferation, mice were euthanized on the same day of the last drug dose.

## Slice preparation for electrophysiology and calcium imaging experiments

Coronal slices (300 μm thick) containing the hippocampal region were prepared from mice (4–8 weeks after xenografting) in accordance with a protocol approved by Stanford University Institutional Animal Care and Use Committee (IACUC). After rapid decapitation, the brain was removed from the skull and immersed in ice-cold slicing artificial cerebrospinal fluid (ACSF) containing (in mM): 125 NaCl, 2.5 KCl, 25 glucose, 25 NaHCO$_3$ and 1.25 NaH$_2$PO$_4$, 3 MgCl$_2$ and 0.1 CaCl$_2$. After cutting, slices were incubated for 30 min in warm (30 °C) oxygenated (95% O$_2$, 5% CO$_2$) recovery ACSF containing (in mM): 100 NaCl, 2.5 KCl, 25 glucose, 25 NaHCO$_3$, 1.25 NaH$_2$PO$_4$, 30 sucrose, 2 MgCl$_2$ and 1 CaCl$_2$ before being allowed to equilibrate at room temperature for an additional 30 min.

## Cerebral slice conditioned medium

Wild-type or *Bdnf*-TMKI mice expressing *Thy1::Chr2* were used at 4–7 weeks of age. Brief exposure to isoflurane rendered the mice unconscious before immediate decapitation. Extracted brains (cerebrum) were placed in an oxygenated sucrose cutting solution and sliced at 350um as described previously[1]. The slices were placed in ACSF (see 'Electrophysiology') and allowed to recover for 30 min at 37 C and 30 min at room temperature. After recovery the slices were moved to fresh ACSF and stimulated using a blue-light LED using a microscope objective. The optogenetic stimulation paradigm was 20-Hz pulses of blue light for 30 s on, 90 s off over a period of 30 min. The surrounding conditioned medium was collected and used immediately or frozen at −80 °C for future use.

## Preparation of cells for in vitro electrophysiology

A single-cell suspension of dissociated glioma cells (see 'Cell culture') was plated at a density of 20,000 cells per well of a 24-well plate containing glass coverslips (Electron Microscopy Services) pre-coated with poly-L-lysine (Sigma) and 5 μg ml$^{-1}$ mouse laminin (Thermo Fisher Scientific). The medium was supplemented with B27(+A) (10 μl ml$^{-1}$ serum-free medium plus growth factors; Invitrogen). Cells were incubated overnight prior to whole-cell recordings.

## Electrophysiology

Slices were transferred to a recording chamber and perfused with oxygenated, warmed (28–30 °C) recording ACSF containing (in mM): 125 NaCl, 2.5 KCl, 25 glucose, 25 NaHCO$_3$, 1.25 NaH$_2$PO$_4$, 1 MgCl$_2$ and 2 CaCl$_2$. Slices were visualized using a microscope equipped with DIC optics (Olympus BX51WI). Recording patch pipettes (2–3 MΩ) were filled with potassium gluconate-based pipette solution containing (in mM): 130 potassium gluconate, 20 KCl, 5 sodium phosphocreatine, 10 HEPES, 4 Mg-ATP, 0.3 GTP, and 50 μM Fluo-4, pH 7.3. Pipette solution additionally contained Alexa 568 (50 μM) to visualize the cell by dye-filling during whole-cell recordings. Glutamate (1 mM; Sigma) in recording ACSF was applied for a period of 250 ms via a puff pipette approximately 100 μm away from the patched cell and controlled by a Picospritzer II (Parker Hannifin). Tetrodotoxin (0.5 μM; Tocris) was perfused with the recording ACSF to prevent neuronal action potential firing in all glutamate puff experiments. Recombinant BDNF human protein (Peprotech, 450-02), or NLGN3 (OriGene Technologies, TP307955), was added to ACSF at 100 ng ml$^{-1}$ and perfused for 30 min to test changes in response to glutamate puff or evoked stimulation. Other drugs used for electrophysiology were NBQX (10 μM; Tocris), AP-5 (100 μM; Tocris), and TBOA (200 μM; Tocris), KN-93 (10 μM; Tocris), KN-92 (10 μM; Tocris) and perfused for 2 h. When used for in vitro slice application, drugs were made up as a stock in distilled water or dimethylsulfoxide (DMSO) and dissolved to their final concentrations in ACSF before exposure to slices. Synaptic responses were evoked with a bipolar electrode connected to an Iso-flex stimulus isolator (A.M.P.I.) placed in the strata radiatum. Signals were acquired with a MultiClamp 700B amplifier (Molecular Devices) and digitized at 10 kHz with an InstruTECH LIH 8 + 8 data acquisition device (HEKA).

For in vitro recordings, glioma cells expressing ChR2 were placed in an extracellular Tyrode medium (150 mM NaCl, 4 mM KCl, 2 mM CaCl$_2$, 2 mM MgCl$_2$, 10 mM HEPES pH 7.4, and 10 mM glucose). Borosilicate pipettes (Harvard Apparatus, with resistance of 4–6 mΩ) were filled with intracellular medium (140 mM potassium gluconate, 10 mM EGTA, 2 mM MgCl$_2$ and 10 mM HEPES pH 7.2). After break-in, cells were held for at least 5 min before recording to ensure cell health and stability of the recording. Light was delivered with the Lumencor Spectra X Light engine with 470 nm for blue-light delivery, respectively. Light stimulation with 1.0 mW mm$^{-2}$, 20 Hz light power density at varying light-pulse width (5 ms or 25 ms), and all recordings were performed in triplicate

to ensure stable and reproducible data. Recordings were randomized in order across conditions to counterbalance for unknown variables.

Data were recorded and analysed using AxoGraph X (AxoGraph Scientific) and IGOR Pro 8 (Wavemetrics). For representative traces, stimulus artifacts preceding the synaptic currents have been removed for clarity.

## Calcium imaging

SU-DIPG-VI and SU-DIPG-XIII-FL were transduced with lentivirus containing the genetically encoded calcium indicator GCaMP6s (pLV-ef1-GCAMP6s-P2A-nls-tdTomato) as described[4]. Cells were xenograft into the CA1 region of the hippocampus as described above.

Calcium imaging experiments performed on in situ SU-DIPG-XIII-FL xenograft slices were visualized using a microscope equipped with DIC optics (Olympus BX51WI). Excitation light was at 594 (for TdTomato) and 488 (for GCaMP6s) provided by pE-300 ULTRA (CoolLED). The recording software used was FlyCapture2 (Point Grey). Calcium imaging experiments performed on in situ SU-DIPG-VI xenograft slices were visualized using a Prairie Ultima XY upright two-photon microscope equipped with an Olympus LUM Plan FI W/IR-2 40x water immersion objective with Prairie View v5.6 software, as described[4]. A tunable Ti:Sapphire laser (Spectra Physics Mai Tai DeepSee) provided the excitation light (920 nm) for both tdTomato and GCaMP6s. PMTs were set to 750 V for each channel. These settings resulted in a power of approximately 30 mW at 920 nm. The emission filters wavelength ranges were for PMT1: 607 nm centre wavelength with 45 nm bandpass (full width at half maximum) and PMT: 525 nm with 70 nm bandpass (full width at half maximum). Image recordings were taken at 1 frame per second (1 Hz) over the course of the glutamate stimulation (approx. 1 min). Tumour cells located in the CA1 region of the hippocampus were identified via the expression for nuclear tdTomato. Slices were perfused with oxygenated aCSF, as described above, at a constant temperature of (28–30 °C) and containing (in mM): 125 NaCl, 2.5 KCl, 25 glucose, 25 NaHCO$_3$, 1.25 NaH$_2$PO$_4$, 1 MgCl$_2$ and 2 CaCl$_2$. Tetrodotoxin (0.5 μM) was perfused with the recording ACSF to prevent neuronal action potential firing. Glutamate (1 mM) in recording ACSF was applied via a puff pipette (250 ms) approximately 100 μm away from the tdTomato-expressing cells and cells were stimulated three times, with approximately-2 min intervals, to ensure a reliable response. Glutamate solution contained Alexa 568 (50 μM) to visualize and confirm reliable glutamate puff (for representative images, the tdTomato nuclear signal displayed is prior to ACSF puff for all frames). Recombinant BDNF human protein (Peprotech, 450-02) was added to ACSF at 100 ng ml$^{-1}$, in addition to tetrodotoxin (0.5 μM) and perfused for 30 min to test changes in response to glutamate puff or evoked stimulation. NBQX (10 μM; Tocris) was added to the ACSF (+tetrodotoxin, +BDNF) perfused for 20 min. Analysis was performed as previously described[4]. In brief, using imageJ (v.2.1.0/153c), regions of interest (ROIs) of nuclear tdTomato were defined manually for each glutamate responding GCaMP6s-expressing glioma cell. Mean intensity over the image time-course was used to measure the corresponding change in fluorescence and following background subtraction, the intensity values were calculated as $\Delta F/F$, where $F$ is the basal fluorescence of the ROI and $\Delta F$ is the change in fluorescence of the ROI at peak response relative to the $F$ value. Duration of response to glutamate was calculated as the length of time that an increase in GCaMP6s signal ($\Delta F/F$) was observed above 0 ($F$) in seconds.

## Biotinylation

Glioma cells (SU-DIPG-VI) were seeded on laminin coated wells of 6-well plates at a density of 500,000 cells per condition. One day after plating, the medium was changed to medium without growth factors to 'starve' the cells for three days. Cells were treated with 100 nM BDNF recombinant protein (Peprotech, 450-02, stock 0.25 μg μl$^{-1}$ in 0.1% BSA in H2O) compared to vehicle (equal volume added of 0.1% BSA in H$_2$O), or 100 nM of NLGN3 (OriGene Technologies, TP307955) compared to vehicle, for specified time periods. To label surface proteins,

the cells were washed twice with ice-cold PBS before adding 1 mg ml$^{-1}$ sulfo-NHS-SS-biotin (Thermo Fisher Scientific) for 10 min at 4 °C with continuous gentle shaking. The reaction was quenched (100 mM glycine, 25 mM Tris-HCL, pH 7.4) for 5 min and then washed in ice-cold PBS three times; all procedures were carried out at 4 °C. The biotinylated cells were then lysed in RIPA lysis buffer (Santa Cruz Biotechnology) supplemented with PMSF, protease inhibitor cocktail and sodium orthovanadate as per manufacturers recommendations. Insoluble material was removed by centrifugation at 10,000 g at 4 °C for 10 min and the supernatant was incubated with 50 μl NeutrAvidin agarose resin (Thermo Fisher Scientific) with gentle mixing overnight at 4 °C. Beads were washed with lysis buffer three times and proteins bound to the beads were eluted with NuPage LDS and sample reducing buffer in equal volumes (Life Technologies). Protein lysates were run on 4–20% Tris-Glycine Plus Gels (Novex, Thermo Fisher Scientific) and transferred to PDVF membranes using an iBlot 2 Gel Transfer Device (Thermo Fisher Scientific). Membranes were incubated in 5% BSA in 1× TBS/1% Tween 20 for 1 h. Primary antibodies against GluA4 (Cell Signaling Technology, 8070), GluA3 (Cell Signaling Technology, 4676) and GAPDH (Cell Signaling Technology, 5174) were added at a concentration of 1:1,000 and incubated overnight at 4 °C, before washing and addition of horseradish peroxidase-conjugated secondary antibody (Cell Signaling Technology, 7074). Chemiluminescent signal was detected using either SuperSignal West Femto Maximum Sensitivity Substrate (Thermo Fisher Scientific, PI34095) or Clarity Western ECL Substrate (Biorad, 1705061). Quantification was performed using imageJ (v.2.1.0/153c), where an ROI was manually drawn around the lanes, and plotted for the relative density of signal observed for each band (analyze>gels>plot lanes). The cell surface levels were normalized to their respective total protein level. The level of AMPAR at the cell surface was presented as a percentage of biotinylated cell surface GluA4 from the control average.

## Western blots

For ligand activation experiments, patient-derived cultures were incubated in medium supplemented with only B27 supplement minus vitamin A for three days. Cells were dissociated using 5 mM EDTA in HBSS and resuspended in medium with B27 supplement for 4 h before incubation with recombinant BDNF protein (100 nM, Peprotech, 450-02) or vehicle (equal volume of 0.1% BSA in H$_2$O). After ligand stimulation for the stated time points the cells were washed in PBS, before lysis using the RIPA Lysis Buffer System containing PMSF, protease inhibitor cocktail and sodium orthovanadate (Santa Cruz Biotechnology). Following quantification using the Pierce BCA Protein Assay Kit (Thermo Fisher Scientific), equal amounts of total protein were loaded onto for each sample for standard western blot. Protein lysates were run on 4–20% Tris-Glycine Plus Gels (Novex, Thermo Fisher Scientific) and transferred to PDVF membranes using an iBlot 2 Gel Transfer Device (Thermo Fisher Scientific). Membranes were incubated in 5% BSA in 1× TBS/1% Tween 20 for 1 h. Primary antibodies GluA4 (Cell Signaling Technology, 8070), phospho-GluA4 (Ser862; Invitrogen, PA5-36807), TrkB (Cell Signaling Technology, 4606), β-actin (Cell Signaling Technology, 4970), phospho-p44/42 MAPK (ERK1/2, Thr202/Tyr204; Cell Signaling Technology, 4370), p44/42 MAPK (ERK1/2, Thr202/Tyr204; Cell Signaling Technology, 9102), phospho-AKT (Ser473; Cell Signaling Technology, 4060), AKT (Cell Signaling Technology, 9272), phospho-CAMKII (Thr286; Cell Signaling Technology, 12716), CAMKII (Cell Signaling Technology, 4436), phospho-TrkB (Tyr515; Sigma, SAB4503785) were added at a concentration of 1:1,000 and incubated overnight at 4 °C, before washing and addition of horseradish peroxidase-conjugated secondary antibody (Cell Signaling Technology, 7074). Chemiluminescent signal was detected using either SuperSignal West Femto Maximum Sensitivity Substrate (Thermo Fisher Scientific, PI34095) or Clarity Western ECL Substrate (Biorad, 1705061). Quantification was performed using ImageJ (v.2.1.0/153c), where an ROI was manually drawn around the lanes, and plotted for the relative density of signal

observed for each band (analyze>gels>plot lanes). The conditions were normalized to loading control.

## Synaptic puncta staining and quantification

Fixed neuron–glioma co-culture coverslips were incubated in blocking solution (3% normal donkey serum, 0.3% Triton X-100 in TBS) at room temperature for 1 h. Primary antibodies guinea pig anti-synapsin1/2 (1:500; Synaptic Systems, 106-004), mouse anti-nestin (1:500; Abcam, ab6320), chicken anti-neurofilament (M + H, 1:1,000; Aves Labs, NFM and NFH) or rabbit anti-RFP (1:500; Rockland, 600-401-379) diluted in diluent (1% normal donkey serum in 0.3% Triton X-100 in TBS) and incubated at 4 °C overnight. Following washing, the slides were incubated in secondary antibody (Alexa 594 donkey anti-rabbit IgG; Alexa 405 donkey anti-guinea pig IgG; Alexa 647 donkey anti-mouse IgG and Alexa 488 donkey anti-chicken IgG all used at 1:500 (Jackson Immuno Research)) overnight at 4 °C. Following washing, coverslips were mounted using ProLong Gold Mounting medium (Life Technologies). Images were collected on a Zeiss LSM800 confocal microscope using a 63× oil-immersion objective and post-processed with Airyscan. Co-localization of synaptic puncta images were performed as previously described[4] using a custom ImageJ (v.2.1.0/153c) processing script (please refer for extended details). In brief, the quantification determines co-localization of presynaptic synapsin and postsynaptic PDS95–RFP within a defined proximity of 1.5 µm. Background fluorescence is removed using rolling ball background subtraction and peaks detected using imglib2 DogDetection plugin which determines the region of interest for each channel. The percentage of total glioma ROIs that are within 1.5 µm of a neuron ROI is reported. The script was implemented in ImageJ (v.2.1.0/153c).

## Cloning constructs

For SEP–GluA2(Q)–TagBFP, Addgene plasmid EFS-Cas9-Puro (#138317) was digested with AgeI, MluI, and EcoRV; the 6 kb lentiviral backbone was isolated via gel extraction. The SEP fragment with Gibson overhangs was amplified from Addgene plasmid pCI-SEP-GluR2(Q) (#24002) with primers: 5′-AACGGGTTTGCCGCCAGAACACAGGACCGGTGCCAC CATGCAAAAGATTATGCATATTTC-3′ and 5′-CCCCCTATCTGTATGCT GTTGCTAGCTTTGTATAGTTCATC-3′. Human GRIA2 (GluA2) with Gibson overhangs was amplified from pLV-EF1a-GFP-GRIA2[4] in two parts to introduce R583Q. GRIA-part 1 was amplified with primers: 5′-CAAAGCTAGCAACAGCATACAGATAGGG-3′ and 5′-TTGGC GAAATATCGCATCCCTGCTGCATAAAGGCACCCAAGGA-3′. GRIA-part 2 was amplified with primers: 5′-TTATGCAGCAGGGATGCG ATATTTCGCCAA-3′ and 5′-TCTTCGACATCTCCGGCTT GTTTCAGCAGAGAGAAGTTTGTTGCGCCGGATCCAATTTTAACACTTTC GATGC-3′. TagBFP2 with Gibson overhangs was amplified from Addgene plasmid pLenti6.2-TagBFP (#113724) with primers: 5′-TCTGC TGAAACAAGCCGGAGATGTCGAAGAGAATCCTGGACCGATGAGCGAG CTGATTAAG-3′ and 5′-TTGTAATCCAGAGGTTGATTGTCGACTTA ACGCGTTTAATTAAGCTTGTGCCC-3′. DNA fragments above were stitched together using Gibson Assembly and transformed.

For PSD95-PURO, Addgene plasmid EFS-Cas9-Puro (#138317) was digested with AgeI, BamHI, and EcoRV; the 6.7 kb lentiviral backbone was isolated via gel extraction. PSD95-RFP with Gibson overhangs was amplified from PSD95-RFP[4] with primers: 5′-TCGCAACGGGTTTGCCGC CAGAACACAGGTCTAGAGCCACCATGGACTGTCTCTGTATAG-3′ and 5′-TGTTTCAGCAGAGAGAAGTTTGTTGCGCCGGATCCATTAAGTTTGT GCCCCAG. DNA fragments above were stitched together using Gibson Assembly and transformed.

## CRISPR deletion and shRNA knockdown

Target sequencing for single guide RNA (sgRNA) was generated using the online predictor at https://cctop.cos.uni-heidelburg.de. The validated sgRNA sequence used for NTRK2 deletion in all cultures was 5′-GTCGCTGCACCAGATCCGAG-3′. The scrambled control sequence used was 5′-GGAGACGTGACCGTCTCT-3′. The custom oligonucleotides were purchased from Elim Biopharmaceuticals. The oligonucleotides were phosphorylated in a reaction with the oligonucleotide (10 µM), 1× T4 DNA ligase buffer (B0202, NEB) and T4 PNK (M020, NEB) with a program 45 min 37 °C, 2 min 30 s at 95 °C, cool 0.1 °C s$^{-1}$ to 22 °C. The sgRNA was cloned into the Lenti vector (pL-CRISPR.EFS.RFP, Addgene #57819). First the vector was digested in a reaction with Fast Digest Buffer (B64, Thermo Fisher Scientific), BsmBI restriction enzyme (FD0454, Thermo Fisher Scientific), DTT (10 mM) with program 45 min at 37 °C, heat inactivate 10 min at 65 °C. The digested vector backbone was dephosphorylated using Antartica phosphatase (M0289, NEB) in Antarctic phosphatase buffer (B0289, NEB) at 37 °C for 30 min, before purifying after running on a 1% agarose gel. The phosphorylated oligonucleotide duplexes were ligated into the vector backbone in a reaction with T4 DNA ligase buffer and T4 DNA ligase and incubated at room temperature for 1 h. Stabl3 (Invitrogen) cells were transformed with the assembled plasmids and individual colonies picked the next day for propagation and sanger sequencing (ElimBio). Lentiviral particles for were produced following transfection of the lentiviral packaging vectors (pΔ8.9 and VSV-g) and either the NTRK2 CRISPR vector or the control scramble vector into HEK293T cells and collected 48 h later. The viral particles were concentrated using Lenti-X Concentrator (Takara Bio) and resuspended in TSM base and stored at −80 °C for future use. The RFP-positive cells were FACs sorted for purity and returned to culture.

Lentiviral particles for shRNA knockdown of NTRK2 were produced following transfection of the lentiviral packaging vectors (pΔ8.9 and VSV-g) and either the NTRK2 shRNA vector (TRCN0000197207; Sigma) or the control shRNA vector (SHCOO2; Sigma) into HEK293T cells and collected 48 h later. The viral particles were concentrated using Lenti-X Concentrator and resuspended in TSM base and stored at −80 °C for future use. Control or NTRK2 shRNA lentiviral particles were transduced into SU-DIPG-VI cultures and the transduced cells were selected with puromycin (4 µg ml$^{-1}$) from day 3.

## pHluorin live imaging

Glioma cells (SU-DIPG-VI) expressing the SEP-GluA2(Q)-TagBFP and PSD95-RFP-Puro constructs (see 'Cloning constructs') were cultured as adherent cells, with mouse neurons, on laminin coated 27 mm glass bottom plates (150682, Thermo Fisher Scientific). ACSF was made at pH 7.4 (see 'Electrophysiology') and at pH 5.5 using the membrane impermeable acid MES hydrate (Sigma) to replace NaHCO$_3$ at equimolar concentration. The ACSF was perfused onto the culture dish using a 3D-printed custom-built stage and tubing for manual perfusion of the solution. Images were collected using a Zeiss LSM980 confocal microscope equipped with a plexiglass environmental chamber, heated stage and CO$_2$ module, and post-processed with Airyscan. The cells were kept at 37 °C with 5% CO$_2$ for the duration of the imaging period. SEP puncta (channel setting for Alexa 488) were identified on the glioma cells as co-localized to the PSD95–RFP (channel setting for Alexa 594) puncta signal to the GFP signal from the SEP-GluA2(Q) puncta. In ImageJ (v.2.1.0/153c), an ROI was manually drawn over the PSD95–RFP signal and used to measure mean intensity of the both RFP and the SEP signal, thus blinding the area chosen for the SEP–GluA2(Q) signal. All puncta analysed were identified for the first timepoint and quantified for all the subsequent time points, thus the choice was blind with respect to outcome. The mean fluorescence intensity of SEP–GluA2(Q) were represented as a ratio to the levels of PSD95–RFP, to account for any fluorescence intensity changes that may occur due to photobleaching or z-axis drifting during the imaging time course. For BDNF perfusion experiments, ACSF (pH 7.4) containing 100 nM BDNF (Peprotech, 450-02) was perfused into the chamber. After imaging the signal in response to BDNF, the signal was then quenched with pH 5.5 to confirm the puncta of interest were membrane-bound GluA subunits. The fluorescence intensity was measured using ImageJ (v.2.1.0/153c).

For the pH validation experiments, the ratio of SEP–GluA2(Q)/PSD95–RFP fluorescence intensity was reported. For the BDNF perfusion experiments, the intensity values were calculated as $\Delta F/F$, where $F$ is the basal SEP–GluA4(Q)/PSD95–RFP ratio and $\Delta F$ is the change in fluorescence of the SEP/PSD95 signal at each subsequent timepoint relative to the $F$ value.

## Immuno-electron microscopy

Twelve weeks post xenografting, mice were euthanized by transcardial perfusion with Karnovsky's fixative: 4% PFA (EMS 15700) in 0.1 M sodium cacodylate (EMS 12300), 2% glutaraldehyde (EMS 16000), p.H 7.4. For all xenograft analysis, transmission electron microscopy was performed in the tumour mass located in the CA1 region of the hippocampus. At room temperature the samples were post fixed in 1% osmium tetroxide (EMS 19100) for 1 h, washed 3 times with ultrafiltered water, before 2-h en bloc staining. The samples were dehydrated in graded ethanol (50%, 75% and 95%) for 15 min each at 4 °C before equilibrating to room temperature and washed in 100% ethanol twice, followed by a 15 min acetonitrile wash. Samples were immersed for 2 h in Embed-812 resin (EMS 14120) with 1:1 ratio of acetonitrile, followed by a 2:1 Embed-812:acetonitrile for 2 h, then in Embed-812 for 2 h. The samples were moved to TAAB capsules with fresh resin and kept at 65 °C overnight. Sections of 40 and 60 nm were cut on an Ultracut S (Leica) and mounted on 100-mes Ni grids (EMS FCF100-Ni). For immunohistochemistry, microetching was done with 10% periodic acid and eluting of osmium with 10% sodium metaperiodate for 15 min at room temperature on parafilm. Grids were rinsed with water three times, followed by 0.5 M glycine quench, and then incubated in blocking solution (0.5% BSA, 0.5% ovalbumin in PBST) at room temperature for 20 min. Primary rabbit anti-GFP (1:300; MBL International) was diluted in the same blocking solution and incubated overnight at 4 °C. The next day, grids were rinsed in PBS three times, and incubated in secondary antibody (1:10 10-nm gold-conjugated IgG TED Pella15732) for 1 h at room temperature and rinsed with PBST followed by water. For each staining set, samples that did not contain any GFP-expressing cells were stained simultaneously to control for any non-specific binding. Grids were contrast stained for 30 s in 3.5% uranyl acetate in 50% acetone followed by staining in 0.2% lead citrate for 90 s. Samples were imaged using a JEOL JEM-1400 TEM at 120 kV and images were collected using a Gatan Orius digital camera. Secondary antibody-only controls were used to compare for specific binding and quantification of images was performed by a blinded investigator.

## Electron microscopy data analysis

Sections of hippocampal preparations bearing xenografted SU-DIPG-VI cells were imaged as above using TEM imaging. Overall, 280 sections of SU-DIPG-VI wild-type across 7 mice and 253 sections of *NTRK2*-KO across 7 mice were analysed. Electron microscopy images were captured at 6,000×, with a 15.75 µm$^2$ field of view. Identified synapses were verified by 2 independent, blinded investigators. Glioma cells were counted and confirmed after unequivocal identification of immunogold particle labelling with four or more particles. To confirm clear synaptic structures, the following three criteria needed to be clearly met: (1) presence of synaptic vesicle clusters; (2) visually apparent synaptic cleft; and (3) clear postsynaptic density in the glioma cell. The number of confirmed glioma–neuron synapses identified was divided by the total number of glioma cells identified to provide the percentage of synaptic structures present. Overall, the analyses identified 0–6 glioma processes per section and 0–2 neuron–glioma synaptic structures per section.

## Immunohistochemistry

Mice were anaesthetized with intraperitoneal avertin (tribromoethanol), then transcardially perfused with 20 ml of PBS. Brains were fixed in 4% PFA overnight at 4 °C, then transferred to 30% sucrose

for cryoprotection. Brains were then embedded in Tissue-Tek O.C.T. (Sakura) and sectioned in the coronal plane at 40 µm using a sliding microtome (AO 860, American Optical).

For immunohistochemistry, coronal or sagittal sections were incubated in blocking solution (3% normal donkey serum, 0.3% Triton X-100 in TBS) at room temperature for 30 min. Mouse anti-human nuclei clone 235-1 (1:200, Millipore), rabbit anti-Ki67 antibody (1:500, Abcam, ab15580) or chicken anti-GFP (1:500, Abcam, ab6320) were diluted in antibody diluent solution (1% normal donkey serum in 0.3% Triton X-100 in TBS) and incubated overnight at 4 °C. Sections were then rinsed once with TBS, before an incubation with DAPI (1 µg ml$^{-1}$ in TBS, Thermo Fisher Scientific) and then another rinse with TBS. Slices were incubated in secondary antibody solution; Alexa 594 donkey anti-rabbit IgG, Alexa 647 donkey anti-mouse IgG or Alexa 488 donkey anti-chicken all used at 1:500 (Jackson Immuno Research) in antibody diluent at 4 °C overnight. Sections were washed three times with TBS and mounted with ProLong Gold Mounting medium (Life Technologies). Confocal images were acquired on either a Zeiss Airyscan1 800 or Zeiss Airyscan LSM980 using Zen 2011 v8.1. Proliferation index was determined by quantifying the fraction of Ki67-labelled cells/HNA-labelled cells using confocal microscopy at 20x magnification. Quantification of images was performed by a blinded investigator.

## EdU incorporation assay

EdU staining was performed on in vitro cell culture slides or on glass coverslips in 24-well plates which were pre-coated with poly-L-lysine (Sigma) and laminin (Thermo Fisher Scientific). Neurosphere culture were dissociated with TrypLE and plated onto coated slides, once the cells had adhered the medium was replaced with growth factor-depleted medium for 72 h. Recombinant proteins BDNF (100 nM, Peprotech, 450-02), NGF (100 nM, Peprotech 450-01), NT-3 (100 nM, Peprotech 450-03) or NT-4 (100 nM, Peprotech, 450-04), inhibitors (500 nM entrectinib HY-12678 and 500 nM larotrectinib HY-12866, both MedChem Express) and vehicle (0.1% BSA and/or DMSO) were added for specified times with 10 µM EdU. After a further 24 h the cells were fixed with 4% PFA in PBS for 20 min and then stained using the Click-iT EdU kit and protocol (Invitrogen) and mounted using Prolong Gold mounting medium with DAPI (Life Technologies). Confocal images were acquired on either a Zeiss Airyscan1 800 or Zeiss Airyscan LSM980 using Zen 2011 v8.1. Proliferation index was determined by quantifying the fraction of EdU-labelled cells divided by DAPI-labelled cells, HNA-labeled cells or nestin-labeled cells using confocal microscopy at 20× magnification. Quantification of images was performed by a blinded investigator.

## Bioinformatic analysis

Single-cell expression (in transcripts per million units) and metadata were downloaded from the 3CA Curated Cancer Cell Atlas[61] website (https://www.weizmann.ac.il/sites/3CA/) and the analysis was performed using R version 4.1.1. Gene-expression values were divided by 10 and log$_2$-transformed. Genes were considered analysable and included in the analysis (overall 11,518 genes) in case their average log$_2$ expression was greater than 0.25. Cells were separated into malignant and non-malignant populations according to the metadata file downloaded from the 3CA website. Within each tumour sample the malignant cells were scored using the sigScores function (scalop package available at https://github.com/jlaffy/scalop) for the cell-state programmes from Filbin et al.[36] (that is, AC-like, OC-like and OPC-like) and for two gene signatures reflecting synaptic transmission (SYN) and tumour microtube structure (TM) that were manually curated. Each malignant cell was assigned with a cell state from Filbin et al.[36] if the maximal state score was greater than 0.5. Cells that did not achieve such a score for any of the states were assigned with an 'unresolved' state. Lineage and stemness coordinates were computed for each cell as described[36] This basic data object was used for generating the panels

of Extended Data Fig. 4. *NTRK2* detection rate (Extended Data Fig. 4a) was computed for each state by summing the number of cells with *NTRK2* expression level greater than zero and dividing by the number of cells assigned to that state. *NTRK2* expression level (Extended Data Fig. 4b) was smoothened for the purpose of data visualization by assigning each cell with the average *NTRK2* expression of its 10 nearest neighbours (using the *k*-nearest-neighbours algorithm, FNN package) in the 2-dimensional lineage versus stemness space. Pearson correlation between *NTRK2* and analysable genes was computed for each state by centring the matrix across the assignable cells (351 AC-like, 201 OC-like and 720 OPC-like) and computing the Pearson correlation coefficient for each cell state between *NTRK2* and the each of the analysable genes (across the cells classified to the particular state). Genes were included in the analysis (Extended Data Fig. 4d) in case the absolute Pearson correlation coefficient was greater than 0.25 in at least one state (overall 519 genes passed this threshold). Gene Ontology enrichment analysis (Extended Data Fig. 4e,f,g) was computed for the positively correlated genes in each state (145, 138 and 97 genes with Pearson correlation coefficient greater than 0.25 for the AC-like, OC-like and OPC-like states, respectively) using the function enrichGo (package clusterProfiler). For scRNAseq processing of individual biopsy samples, RSEM-normalized gene abundances for the Filbin dataset[36] were downloaded from the Single Cell Portal (https://singlecell.broadinstitute.org/single_cell, Gene Expression Omnibus (GEO) accession: GSE102130). The data were log-transformed with a pseudocount of one.

For *NTRK2* and *BDNF* expression levels in DIPG patient-derived cell cultures, fragments per kilobase of transcript per million mapped reads (FPKM) data were analysed from datasets kept in-house and are publicly available to download (GEO accession: GSE94259) and (GEO accession: GSE222560). Cultures included were SU-DIPG-IV, SU-DIPG-VI, SU-DIPG-XIII-p, SU-DIPG-XVII, SU-DIPG-XXI, SU-DIPG25 and SU-DIPG-XIII-FL.

For TrkB isoform analysis, our previously published scRNASeq dataset[4] of patient-derived orthotopic xenograft models was used. NTRK2 isoform abundances were quantified from FASTQ files using Kallisto[62] (v.0.46.1). Reads were pseudoaligned against a reference transcriptome created from the hg38 reference genome using Ensembl hg38 transcript annotations. Estimated counts were library size normalized to counts per million and log-transformed with a pseudocount of one. Cells with libraries containing irregular GC content were identified and removed from analysis, resulting in a total of 321 cells.

For transcriptome analysis of BDNF-treated tumour cells, cultures were grown in serum-free medium without growth factors for three days before addition of 100 nM BDNF recombinant protein (Peprotech, 450-02) and incubated for 16 h before harvesting RNA. RNA was extracted from pelleted cell culture samples using the RNeasy Isolation kit (Qiagen) as per manufacturer's instructions. Total RNA samples were submitted to Stanford Functional Genomics Facility. RNA integrity was established with Bioanalyzer trace (Agilent). The mRNA was prepared for sequencing using the KAPA Stranded mRNAseq Library prep kit (KK8420), and libraries were indexed with Truseq RNA UD from Illumina (20021454) as per manufacturer's instructions. The sequencing was performed on the Illumina NextSeq 500.

Reads were mapped to hg19 annotation using Tophat2[63] (version 2.0.13) and transcript expression was quantified against RefSeq gene annotations using featureCounts[64] (v2.0.3). Differential gene expression and log$_2$ fold change calculations were determined using the DESeq2 (v.1.36.0) package in R[65]. Mitochondrial genes were excluded from analysis. Volcano plot analysis was conducted using the R-based EnhancedVolcano (v1.14.0) package (https://github.com/kevinblighe/EnhancedVolcano). The included volcano plot analysis was filtered for *P* values <0.4 and log$_2$FC< 1.5 to highlight the top differentially expressed genes.

## Statistics and reproducibility

Statistical tests were conducted using Prism v9.1.0 (GraphPad) software unless otherwise indicated. Gaussian distribution was confirmed by the Shapiro−Wilk normality test. For parametric data, unpaired two-tailed Student's *t*-test or one-way ANOVA with Tukey's post hoc tests to examine pairwise differences were used as indicated. Paired two-tailed Student's *t*-tests were used in the case of same-cell experiments (as in electrophysiological recordings). For data normalized to a control mean (as in western blot or pHluorin analyses) one-sample *t*-test were used against the mean of the control (either 0 or 1), with Wilcoxon signed-rank test for non-parametric data. For non-parametric data, a two-sided unpaired Mann−Whitney test was used as indicated, or a one-tailed Wilcoxon matched pairs signed-rank test was used for same-cell experiments. Two-tailed log rank analyses were used to analyse statistical significance of Kaplan−Meier survival curves. Statistical test results are reported in the figure legends and in Supplementary Table 1. On the basis of variance of xenograft growth in control mice, we used at least three mice per genotype to give 80% power to detect effect size of 20% with a significance level of 0.05. All in vitro experiments have been performed in at least three independent coverslips for each experiment and performed in at least two independent experiments. The number of biological replicates (mice for in vivo growth, calcium imaging, electrophysiological recording, optogenetic stimulation and immuno-electron microscopy experiments) is indicated in the figure legends and was three or greater for all experiments. The comparison of wild-type glioma to *NTRK2*-KO glioma growth was tested in three independent in vivo experiments and in vivo synaptic connectivity in two independent experiments. The effect of entrectinib on glioma growth was tested in two independent in vivo experiments. The increase of glioma synaptic/glutamatergic current strength in response to BDNF application was tested in three independent electrophysiological experiments. All protein experiments assayed by western blot analysis were performed 3 independent times, except for *NTRK2* knockdown, which was performed twice, with data shown from one experiment and entrectinib administration, which was performed once. The ChR2+ glioma in vivo optogenetic stimulation experiment was performed in two independent experiments, with data shown from one experiment.

## Materials availability

All unique materials such as patient-derived cell cultures are freely available and can be obtained by contacting the corresponding author with a standard materials transfer agreement with Stanford University.

## Reporting summary

Further information on research design is available in the Nature Portfolio Reporting Summary linked to this article.

## Data availability

Single-cell and bulk RNA-sequencing data in Extended Data Fig. 1, Extended Data Fig. 4 and Extended Data Fig. 7 were analysed from publicly available datasets on the Gene Expression Omnibus (GEO) (GSE102130, GSE134269, GSE94259 and GSE222560). Bulk RNA-sequencing data used in Extended Data Fig. 9 are publicly available on GEO (GSE222481). Patch-seq data referenced in the rebuttal is available on GEO (GSE222398). The hg38 and hg19 reference genomes were used for transcriptome annotation. All source data and original western blots are included in the source data file and Supplementary Fig. 1.

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

**Acknowledgements** The authors thank R. C. Malenka for helpful input and advice, G. Wang for his expertise and help in two-photon and confocal microscopy, M.E. Greenberg for providing *Bdnf*-TMKI mice, and S. Nagaraja for bulk RNA sequencing of DIPG cultures. All graphics were created with BioRender.com. This work was supported by grants from the National Institute of Neurological Disorders and Stroke (R01NS092597 to M.M.), NIH Director's Pioneer Award (DP1NS111132 to M.M.), National Cancer Institute (P50CA165962 to M.M. and M.L.S., R01CA258384 to M.M., and U19CA264504 to M.M. and M.L.S.), Abbie's Army (to M.M.), Robert J. Kleberg Jr and Helen C. Kleberg Foundation (to M.M.), Gatsby Charitable Foundation (to M.M.), Cancer Research UK (to M.M.), Damon Runyon Cancer Research (to K.R.T.), ChadTough Defeat DIPG (to M.M. and T.B.), Stanford Maternal & Child Health Research Institute (to K.R.T.), N8 Foundation (to M.M.), McKenna Claire Foundation (to M.M.), Kyle O'Connell Foundation (to M.M.), Virginia and D.K. Ludwig Fund for Cancer Research (to M.M.), Waxman Family Research Fund (to M.M.) and Will Irwin Research Fund (to M.M.). A.S. is partially supported by the Israeli Council for Higher Education (CHE) via the Weizmann Data Science Research Center.

**Author contributions** K.R.T. and M.M. designed, conducted and analysed experiments. T.B. conducted in slice electrophysiology experiments and Y.S.K. performed in vitro electrophysiology experiments. K.R.T. conducted calcium imaging experiments, xenografting, drug treatment experiments, pHluorin confocal imaging, in vitro and in vivo experiments, data collection and analyses. L.N. conducted electron microscopy data acquisition, and K.R.T and H.S.V. performed analyses. P.P.D. and A.S. performed single-cell transcriptomic analyses. R.M. performed RNA-sequencing data analyses. K.R.T., M.S., S.M.J., A.E.I. and S.M. contributed to neuron–glioma co-culture experiments. K.R.T., S.M.J. and H.Z. contributed to synaptic puncta confocal imaging. K.R.T. performed western blot analyses. G.G.H. generated plasmid constructs. A.P. performed CRISPR deletion. K.R.T., A.H., B.Y., A.C.G. and M.B.K., performed optogenetic experiments. P.J.W. contributed to xenograft and drug treatment experiments. A.E.I. contributed to in vitro data collection. A.H., B.Y., A.C.G., A.S.-T., R.A.-B., H.Z. and I.C. contributed to in vivo data collection. M.A., S.G., T.B. and K.S. contributed to Patch-seq experiments included in the rebuttal. I.T. and M.L.S. provided conceptual and analytical support. K.R.T., H.S.V., T.B. and M.M. contributed to manuscript editing. K.R.T. and M.M. wrote the manuscript. M.M. conceived of the project and supervised all aspects of the work.

**Competing interests** M.M. was on the scientific advisory board for Cygnal Therapeutics, is on the scientific advisory board for TippingPoint Biosciences, and holds equity in MapLight Therapeutics. M.L.S. is an equity holder, scientific co-founder and advisory board member of Immunitas Therapeutics. The other authors declare no competing interests.

**Additional information**
**Correspondence and requests for materials** should be addressed to Michelle Monje.

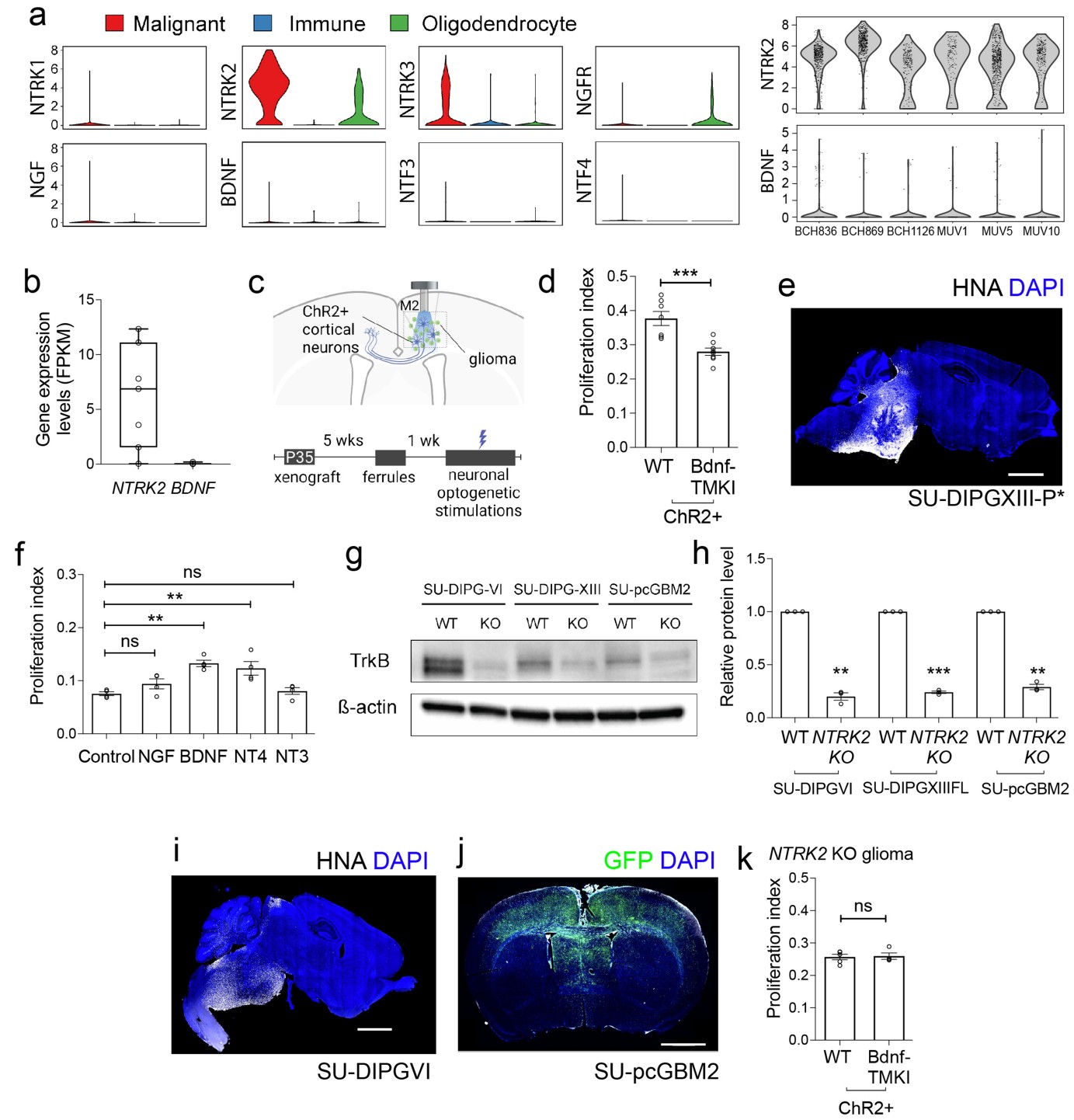

**Extended Data Fig. 1** | See next page for caption.

**Extended Data Fig. 1 | TrkB is the key receptor mediating neuronal BDNF signaling in glioma. a**, Left, Primary human biopsy single cell transcriptomic data[36] illustrating the expression of the neurotrophin family genes in H3K27M+ DMG (red; $n$ = 2,259 cells, 6 study participants), tumor associated, non-malignant immune cells (blue; n = 96 cells, 5 participants) and oligodendrocytes (green; $n$ = 232 cells). Right, *NTRK2* and *BDNF* expression in H3K27M+ DMG malignant single cells primary human biopsy single-cell transcriptomic data from each of 6 study participants (case numbers denoted on $x$ axis). For each individual violin plot, the $y$ axis represents expression $\log_2$ (transcripts per million) and the $x$ axis represents number of individual cells with indicated expression value. **b**, Expression levels of neurotrophin receptors analysed from previously published[59,66,67] and newly reported (GEO# GSE222560) bulk RNA sequencing of human autopsy pediatric DMG ($n$ = six patient-derived glioma samples SU-DIPG-IV, SU-DIPG-VI, SU-DIPG-XIII-P, SU-DIPG-XIII-FL, SU-DIPG-21 and SU-DIPG-25; means 2.36 *NTRK1*, 22.73 *NTRK2*, 8.688 *NTRK3*, 5.439 *NGFR* FPKM; *NTRK2* minimum 0.03273, 25% percentile 1.537, median 6.873, 75% percentile 11.10, maximum 12.34; BDNF minimum 0.01429, 25% percentile 0.01499, median 0.03367, 75% percentile 0.04565, maximum 0.1951). **c** Model for optogenetic stimulation of ChR2-expressing neurons (blue) in microenvironment of glioma xenograft (green); light blue rectangle denotes region of analysis. P, postnatal day. **d**, Proliferation index of SU-DIPG-XIII-FL glioma xenografted to mice with neurons expressing Channelrhodopsin (ChR2 + ) in a wild-type or Bdnf-TMKI genetic background (Fig. 1a) after neuronal optogenetic stimulation (quantified by confocal microscopy of EdU + /HNA cells, as in representative Fig. 1c, n = 7 wild-type ChR2+ mice, 8 Bdnf-TMKI ChR2+ mice, P = 0.0007). **e**, Representative image of tumor burden in a mouse brain (sagittal section) bearing orthotopic

xenograft of SU-DIPG-XIII-P* xenografted to the pons at endpoint. Survival analysis presented in Fig. 1e. White denotes HNA (tumor cells); DAPI nuclei are shown in blue (Scale bar = 2000 μm). **f**, Proliferation rate of SU-DIPG-XIII-FL cultures treated with recombinant proteins NGF, BDNF, NT3, NT4 (100 μM each), compared to vehicle control (quantified by confocal microscopy of EdU + /DAPI cells, as in representative Fig. 1h, n = 4 coverslips/group, Control vs BDNF P = 0.016, Control vs NT4 P = 0.0074). **g**, Representative western blot analysis of TrkB protein levels in wild-type, Cas9-control and *NTRK2* KO cultures (SU-DIPG-VI, SU-pcGBM2, SU-DIPG-XIII-FL), using indicated antibodies. **h**, Quantification of **g**, with levels of TrkB normalized to total protein loading using ß-actin levels and compared to wild-type, Cas9-scramble control, cultures ($y$ axis is in arbitrary units, $n$ = 3 technical replicates, DIPGVI WT vs *NTRK2* KO P = 0.0019, DIPGXIII WT vs *NTRK2* KO P = 0.0002, pcGBM2 WT vs *NTRK2* KO P = 0.0013). **i-j**, Representative images of tumors at survival endpoint for Fig. 1f. **i**, Orthotopic xenograft of SU-DIPG-VI into pons (sagittal section of mouse brain; scale bar = 2000 μm), and in **j**, cortical orthotopic xenograft of SU-pcGBM2 (coronal section of mouse brain). White denotes HNA (tumor cells); Green denotes GFP (tumor cells); DAPI nuclei are shown in blue (scale bar = 2000 μm). **k**, Proliferation index of *NTRK2* KO SU-DIPG-VI glioma xenografted to mice with neurons expressing Channelrhodopsin (ChR2 + ) in a wild-type or Bdnf-TMKI genetic background after neuronal optogenetic stimulation (quantified by confocal microscopy of EdU + /HNA cells, as in representative Fig. 1c, n = 5 wild-type ChR2+ mice, $n$ = 4 BDNF-TMKI ChR2+ mice). Data are mean ± s.e.m. *P < 0.05, **P < 0.01, ***P < 0.001, ns = not significant. Two-tailed unpaired Student's $t$-test for **d** and **k**, one-way analysis of variance (ANOVA) with Tukey's post hoc analysis for **f** and two-tailed one sample $t$-test for **h**.

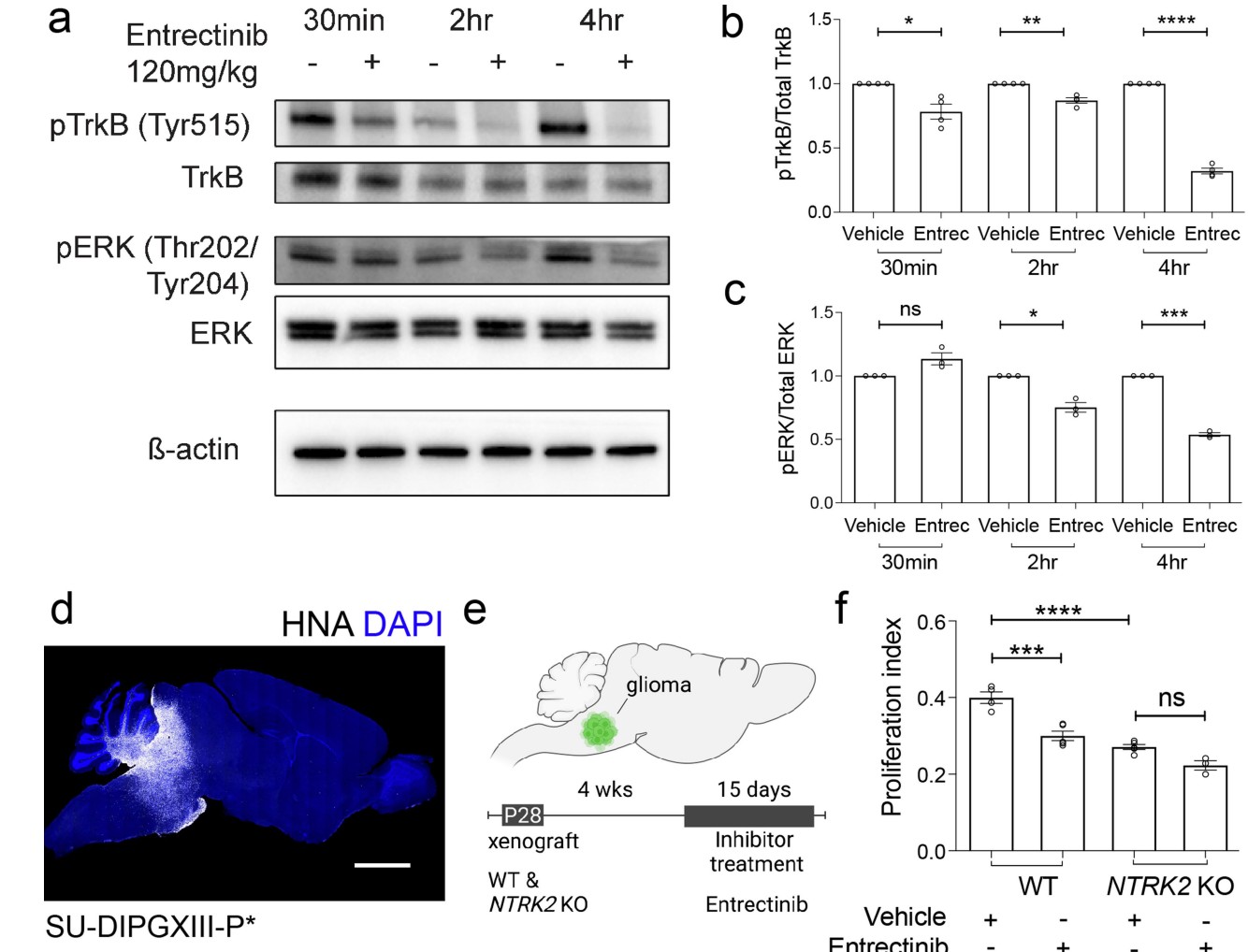

**Extended Data Fig. 2 | Effect of the pan-Trk inhibitor entrectinib on glioma proliferation is mediated by TrkB inhibition. a**, Western blot of whole brain protein lysate collected from NSG mice that were either treated with one PO dose of 120 mg/kg of entrectinib or one dose of vehicle control (PO). The mouse brains were harvested after transcardial perfusion and mice were collected at either 30 min, 2 h and 4 h after vehicle or entrectinib dosing. The protein lysate was probed for the indicated antibodies to demonstrate inhibition of BDNF-TrkB signaling as an indication of effective drug penetration into brain tissue. **b**, Quantification of TrkB phosphorylation by comparing the ratio of the normalized phospho-TrkB (Tyr515) levels to corresponding total TrkB protein levels between the entrectinib treated and vehicle control mice (*y* axis is in arbitrary units, *n* = 4 technical replicates, pTrkB vehicle vs entrectinib 30 min P = 0.036, 2 h P = 0.0082, 4 h P < 0.0001). **c**, Quantification of MAPK pathway activation by comparing the ratio of the normalized phospho-ERK (T202/Y204) to corresponding total protein levels between the entrectinib treated and vehicle control treated mice (*y* axis is in arbitrary units, *n* = 3 technical replicates, pErk vehicle vs entrectinib 2 h P = 0.0220, 4 h P = 0.0010).

**d**, Representative image of mouse brain (sagittal section) from SU-DIPG-XIII-P* xenografted to the pons treated with entrectinib (120 mg/kg PO) at endpoint in survival analyses (presented in Fig. 1g). White denotes HNA (tumor cells); DAPI nuclei are shown in blue (scale bar = 2000 μm). **e**, Experimental model of pontine xenografted WT and *NTRK2* KO glioma (SU-DIPG-VI) treated with the Pan-Trk inhibitor, entrectinib, or vehicle control. **f**, Proliferation index of wild-type and *NTRK2* KO SU-DIPG-VI glioma xenografted to the pons of NSG mice and treated with vehicle or entrectinib (120 mg/kg PO). Quantification by confocal microscopy analysis of EdU + /HNA+ co-positive tumor cells, as in representative Fig. 1c, n = 4 wild-type glioma xenografted, vehicle-treated mice, 5 wild-type glioma xenografted, entrectinib-treated mice, 5 *NTRK2* KO glioma xenografted, vehicle-treated mice, 3 *NTRK2* KO glioma xenografted, entrectinib-treated mice, WT vehicle vs WT entrectinib P = 0.0002, WT vehicle vs *NTRK2* KO vehicle P < 0.0001). Data are mean ± s.e.m. *P < 0.05, **P < 0.01, ***P < 0.001, ****P < 0.0001, ns = not significant. Two-tailed one sample *t* test for **b** and **c**, one-way analysis of variance (ANOVA) with Tukey's post hoc analysis for **f**.

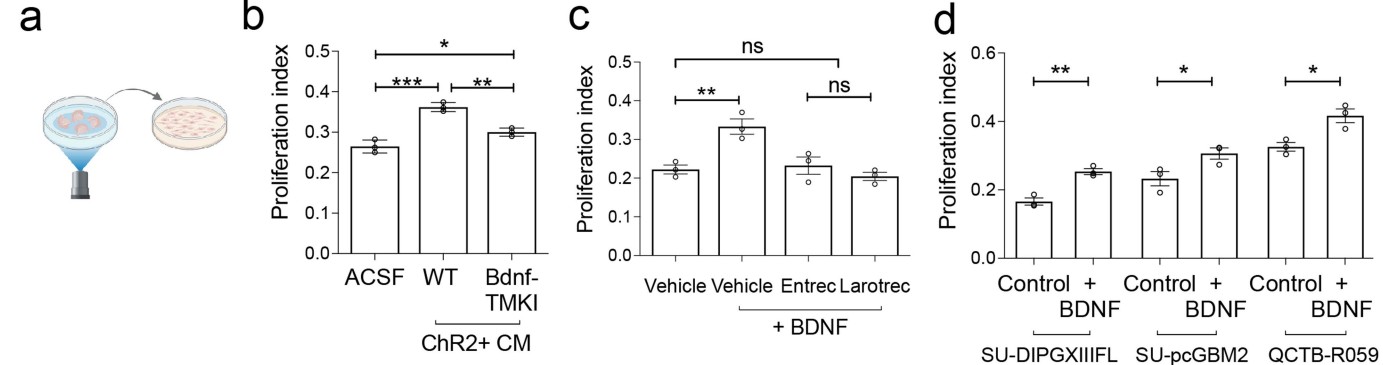

**Extended Data Fig. 3 | Mitogenic effect of BDNF on glioma proliferation.**
**a**, Collection of conditioned medium (CM) from optogenetically stimulated acute cortical slices. **b**, Proliferation index of SU-DIPG-VI cells exposed to wild-type or Bdnf-TMKI CM ($n$ = 3 coverslips/group, quantified by confocal microscopy of EdU + /DAPI cells, as in representative Fig. 1h, ACSF vs WT CM P = 0.0002, ACSF vs *Bdnf*-TMKI CM P = 0.0334, WT CM vs *Bdnf*-TMKI CM P = 0.0024). **c**, Proliferation index of SU-DIPG-VI cells treated with 100 µM of BDNF protein in the presence of pan-Trk inhibitors, entrectinib and larotrectinib at 500 nM (quantified by confocal microscopy of EdU + /DAPI cells, as in representative Fig. 1h, n = 3 coverslips/group, P = 0.0068).

**d**, Proliferation index of DIPG (SU-DIPG-XIII-FL), cortical (SU-pcGBM2) and thalamic (QCTB-R059) pediatric glioblastoma cultures treated with BDNF recombinant protein (100 nM) compared to control cells (vehicle-treated; quantified by confocal microscopy of EdU + /DAPI+ cells, as in representative Fig. 1h, n = 3 coverslips/group, DIPGXIII Control vs BDNF P = 0.0029, pcGBM2 Control vs BDNF P = 0.0494, R059 Control vs BDNF P = 0.0186). Data are mean ± s.e.m. *P < 0.05, **P < 0.01, ***P < 0.001, ns = not significant. One-way analysis of variance (ANOVA) with Tukey's post hoc analysis for **b** and **c**, and two-tailed unpaired Student's *t*-test for **d**.

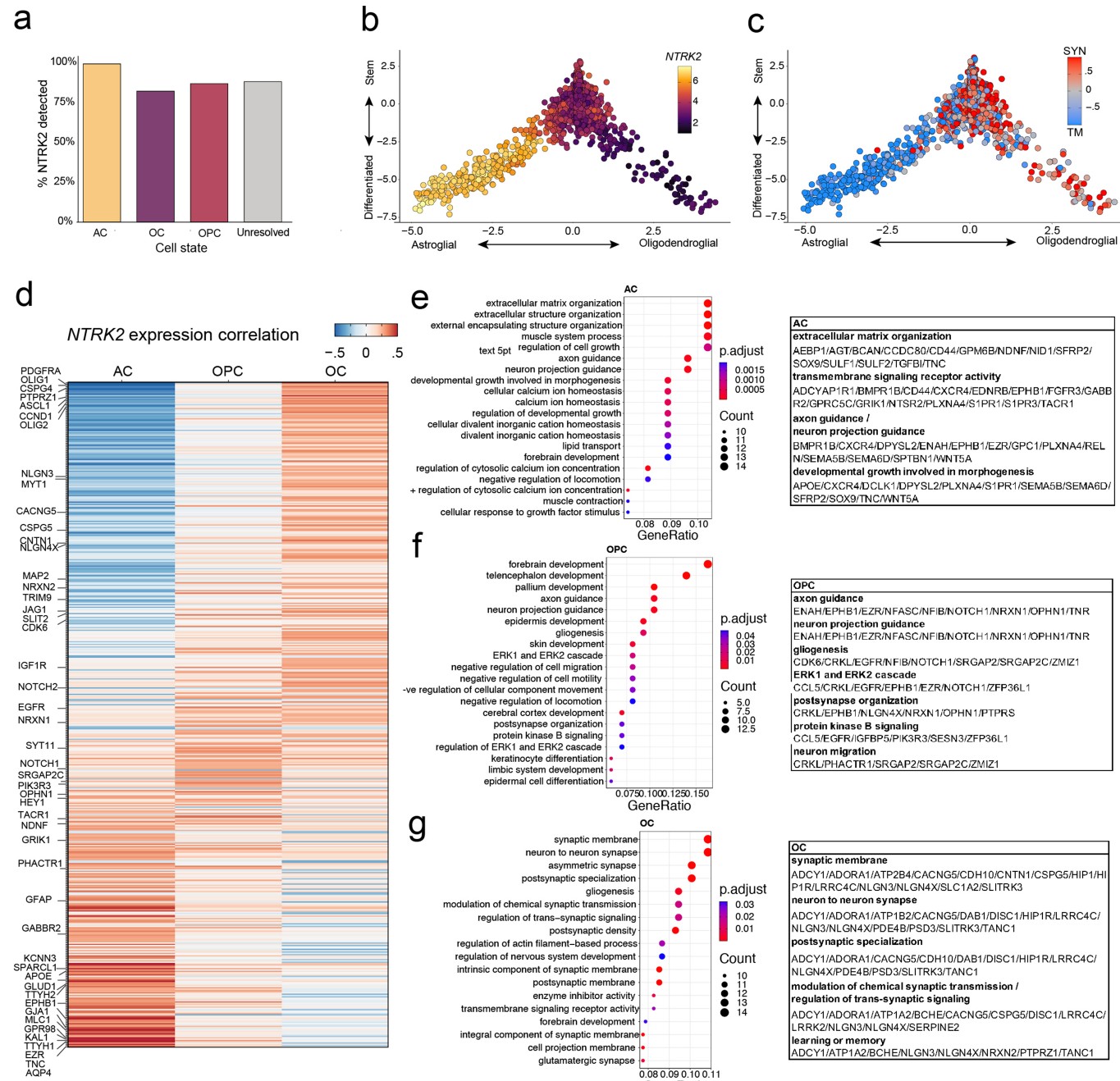

**Extended Data Fig. 4 | *NTRK2* correlates with unique cellular mechanisms in distinct cell-state subgroups of pediatric DMG. a**, Analysis of previously published H3K27M⁺ DMG single-cell RNASeq data[36] quantifying the percentage of tumor cells in which *NTRK2* (TrkB) was captured in either the astrocyte-like (AC), oligodendrocyte-like (OC) and oligodendroglial precursor cell-like (OPC) glioma cells. **b**, *NTRK2* expression level in malignant H3K27M⁺ malignant single cells projected on the glial-like cell lineage (x axis) and stemness (stem to differentiated; y axis) scores. *NTRK2* expression level was smoothened (for the purpose of data visualization only) for each cell by assigning each cell with the average *NTRK2* expression of its nearest neighbors in the Lineage vs. Stemness 2-dimensional space. **c**, Difference between the scores of the synaptic (SYN) and tumor microtube (TM) gene signatures (i.e. SYN – TM) in H3K27M⁺

malignant single cells projected on the lineage (x axis) and stemness (stem to differentiated; y axis) scores. **d**, Heatmap of genes correlating with *NTRK2* expression in distinct cellular subgroups (astrocyte-like, oligodendroglial precursor cell-like, oligodendrocyte-like) of H3K27M⁺ malignant single cells. Genes were ordered according to the AC-OC score difference. **e-g**, Gene Ontology (GO) enrichment analysis for the top genes correlated with *NTRK2* expression in distinct cellular subgroups within H3K27M⁺ diffuse midline glioma tumors (145, 138, 97 genes with Pearson's correlation coefficient greater than 0.25 for the AC-like, OC-like and OPC-like malignant cell states respectively) (**e**, astrocyte-like; **f**, oligodendroglial precursor cell-like; **g**, oligodendrocyte-like). Right, tables depicting the genes associated with the biological processes identified (GO terms) for each cellular subgroup.

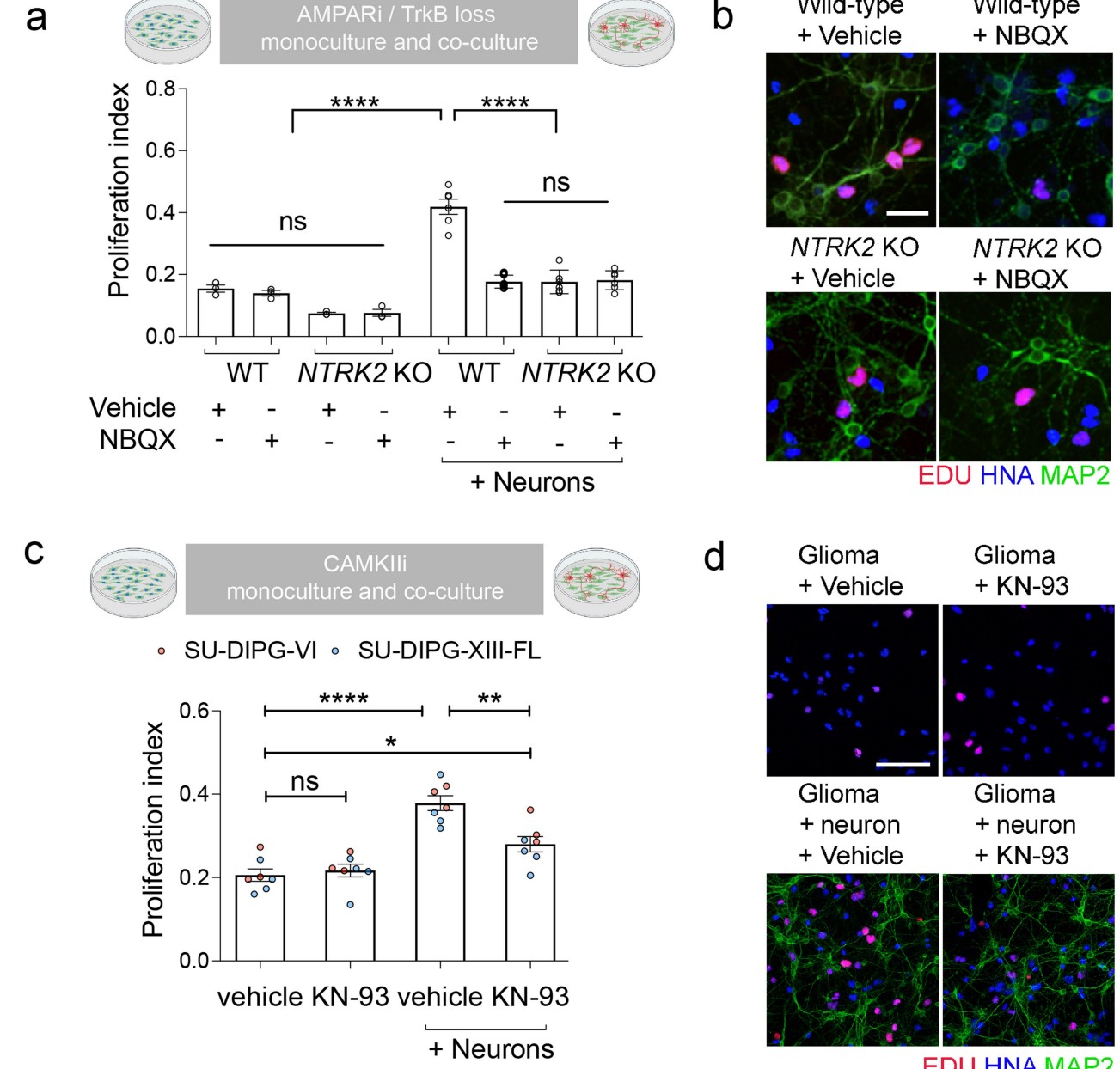

**Extended Data Fig. 5 | Targeting AMPAR, TrkB and CAMKII reduces glioma proliferation in the context of neurons. a**, Proliferation index of SU-DIPG-VI WT and *NTRK2* KO glioma monoculture (left), or glioma co-culture with neurons (right, as in Fig. 1i), in the presence and absence of the AMPAR blocker NBQX (10 μM) (quantified as fraction of EdU+/HNA+ co-positive tumor cells assessed by confocal microscopy, *n* = 3 coverslips/group for glioma monoculture experiments and 6 coverslips/group for neuron-glioma co-culture; experiment replicated in Fig. 1j, WT vehicle vs WT + neurons vehicle P < 0.0001, WT + neurons vehicle vs WT + neurons NBQX P < 0.0001, WT + neurons vs *NTRK2* KO + neurons P < 0.0001). **b**, Representative images of data quantified in **a**; wild-type and *NTRK2* KO glioma cells (SU-DIPG-VI) co-cultured with neurons in the presence and absence of NBQX (10 μM). Blue denotes HNA positive glioma cells; red denotes EdU (proliferative marker); green denotes MAP2 (neurons). Scale

bar = 30 μm. **c**, Proliferation index of SU-DIPG-VI (red data points) and SU-DIPG-XIII-FL (blue data points) as a monoculture or cocultured with neurons in the presence of a CAMKII inhibitor, KN-93 (10 μM) or vehicle control (quantified as fraction of EdU+/HNA+ glioma cells; *n* = 7 coverslips/group, vehicle vs vehicle + neurons P < 0.0001, vehicle + neurons vs KN-93 + neurons P = 0.0017, vehicle vs KN93 + neurons P = 0.0212). **d**, Representative images of data quantified in **c**; glioma cells (SU-DIPG-VI) in monoculture, or co-cultured with neurons, in the presence and absence of KN-93 (10 μM). Blue denotes HNA positive glioma cells; red denotes EdU (proliferative marker); green denotes MAP2 (neurons). Scale bar = 100 μm. Data are mean ± s.e.m., *P < 0.05, **P < 0.01, ****P < 0.0001, ns = not significant, one-way analysis of variance (ANOVA) with Tukey's post hoc analysis.

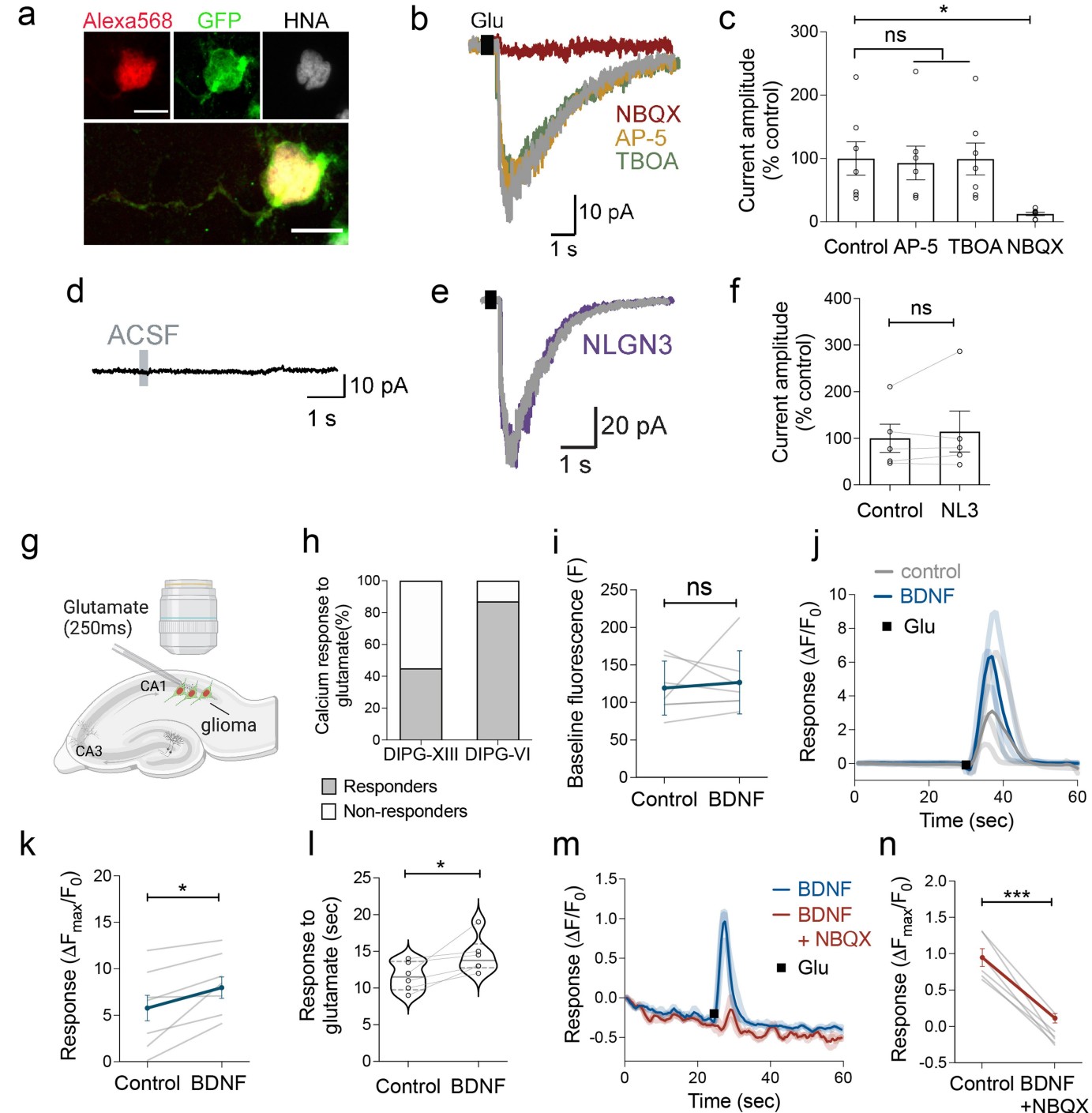

**Extended Data Fig. 6** | See next page for caption.

**Extended Data Fig. 6 | Heterogeneity and specificity of glioma electrophysiological response to glutamate. a**, Representative image of Alexa 568 (red)- filled GFP+ glioma cell following whole-cell patch clamp recording. Co-labelled with GFP (green) and human nuclear antigen (HNA, grey). Scale bars = 10 μm. **b**, Representative voltage-clamp traces of whole cell patch-clamp electrophysiological recordings in glioma cells. Hippocampal slices were perfused with ACSF containing tetrodotoxin (TTX, 0.5 μM), and response to a local puff (250 msec) application of 1 mM glutamate (black square) was recorded from xenografted glioma cells with sequential application of NMDAR blocker (AP-5, 100 μM), TBOA (200 μM), AMPAR blocker (NBQX, 10 μM). **c**, Quantification of data in **b** (*n* = 7 glioma cells, 4 mice, P = 0.0165). **d**, Whole cell patch-clamp electrophysiological recording of glioma cell with ACSF puff, representative voltage clamp trace. **e**, Representative traces of glutamate-evoked inward currents (black square) in patient-derived glioma xenografted cells before (grey) and after 30-minute perfusion with NLGN3 recombinant protein (100 ng/ml) in ACSF (containing TTX, 0.5 μM) (purple). **f**, Quantification of data in **e** (*n* = 5 glioma cells, 3 mice). **g**, Model of calcium imaging of tdTomato nuclear tagged (red nuclei), GCaMP6s-expressing (green calcium transients) glioma cells xenografted into the mouse hippocampal region. **h**, Quantification of number of xenografted SU-DIPG-XIII-FL or SU-DIPG-VI cells glioma cells demonstrating a calcium transient (as depicted in Fig. 2i, j and Extended Data Fig. 6j) in response to a glutamate puff (responders, grey, non-responders, white). **i**, Baseline GCaMP6s intensity in SU-DIPG-VI glioma cells before and 30-min after BDNF exposure, in the absence of glutamate puff (*n* = 7 cells, 3 mice). **j**, GCaMP6s intensity trace of SU-DIPG-VI glioma cells response to glutamate puff before (3 cells, 3 mice: light grey, average: dark grey) and after BDNF perfusion (three cells: light blue, average intensity: dark blue). **k**, SU-DIPG-VI GCaMP6s cell response to glutamate puff at baseline and after BDNF perfusion (100 ng/ml, 30 min, *n* = 7 cells, 4 mice, P = 0.0174). **l**, Duration of calcium transient response to glutamate puff in SU-DIPG-VI hippocampal xenografted cells, before and after perfusion with BDNF (100 ng/ml, 30 min, *n* = 6 cells, 4 mice, P = 0.0302). **m**, Representative traces of SU-DIPG-XIII glioma GCaMP6s intensity in the presence of BDNF (100 ng/ml, 30 min). Response to glutamate application (black) recorded with BDNF perfusion (3 cells, 2 mice: light blue, average: dark blue) or with BDNF and NBQX (10 μM, 3 cells: light red, average: red). **n**, Response of GCaMP6s cells to glutamate puff with BDNF application, in the presence and absence of NBQX (*n* = 6 cells, 3 mice, P = 0.0002). Data are mean ± s.e.m., *P < 0.05, ***P < 0.001, ns = not significant, two-tailed paired Student's *t*-test for **k**, **l**, **n**, and two-tailed Wilcoxon signed pairs matched rank test for **d**, **f** and **i**.

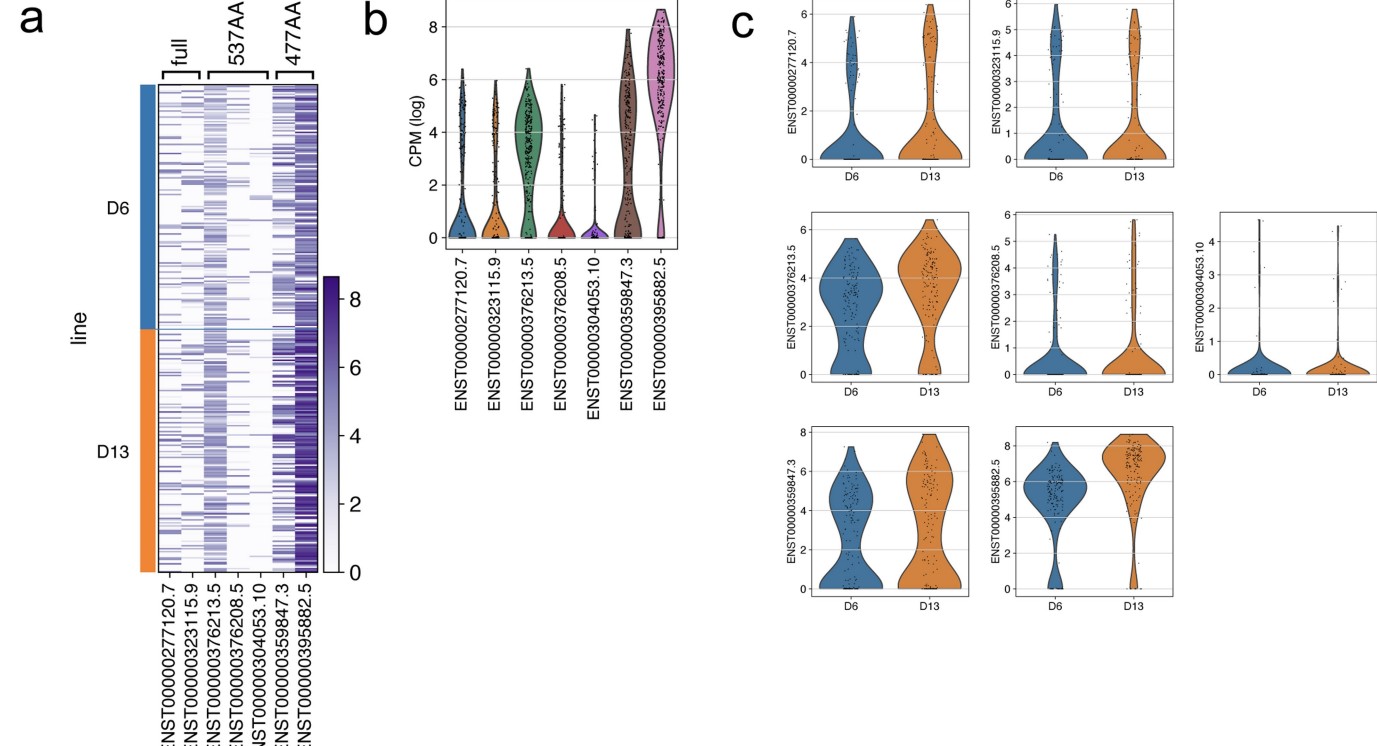

**Extended Data Fig. 7 | TrkB isoform expression in DIPG. a**, Heatmap of single-cell RNASeq data of patient derived glioma xenograft models (SU-DIPG-VI and SU-DIPG-XIII) cells (*n* = 321 cells, 4 mice) demonstrating relative expression of TrkB (*NTRK2*) isoforms; Full (ENST-277120.7 and ENST-323115.9) and Truncated (527aa ENST-376208.5 and ENST-376208.5; 477aa ENST-359847 and ENST-395882.5), with representative Ensembl codes depicted below. **b**, Violin plots of relative expression level of TrkB isoforms (depicted in **a**,) for both SU-DIPG-VI and SU-DIPG-XIII cells combined, shown as log-transformed counts per million (CPM). **c**, Violin plots of relative expression of TrkB isoforms (depicted in **a**,) separated by patient-derived xenograft model type (SU-DIPG-VI (D6) and SU-DIPG-XIII (D13)). *Y*-axis is in log-transformed counts per million (CPM).

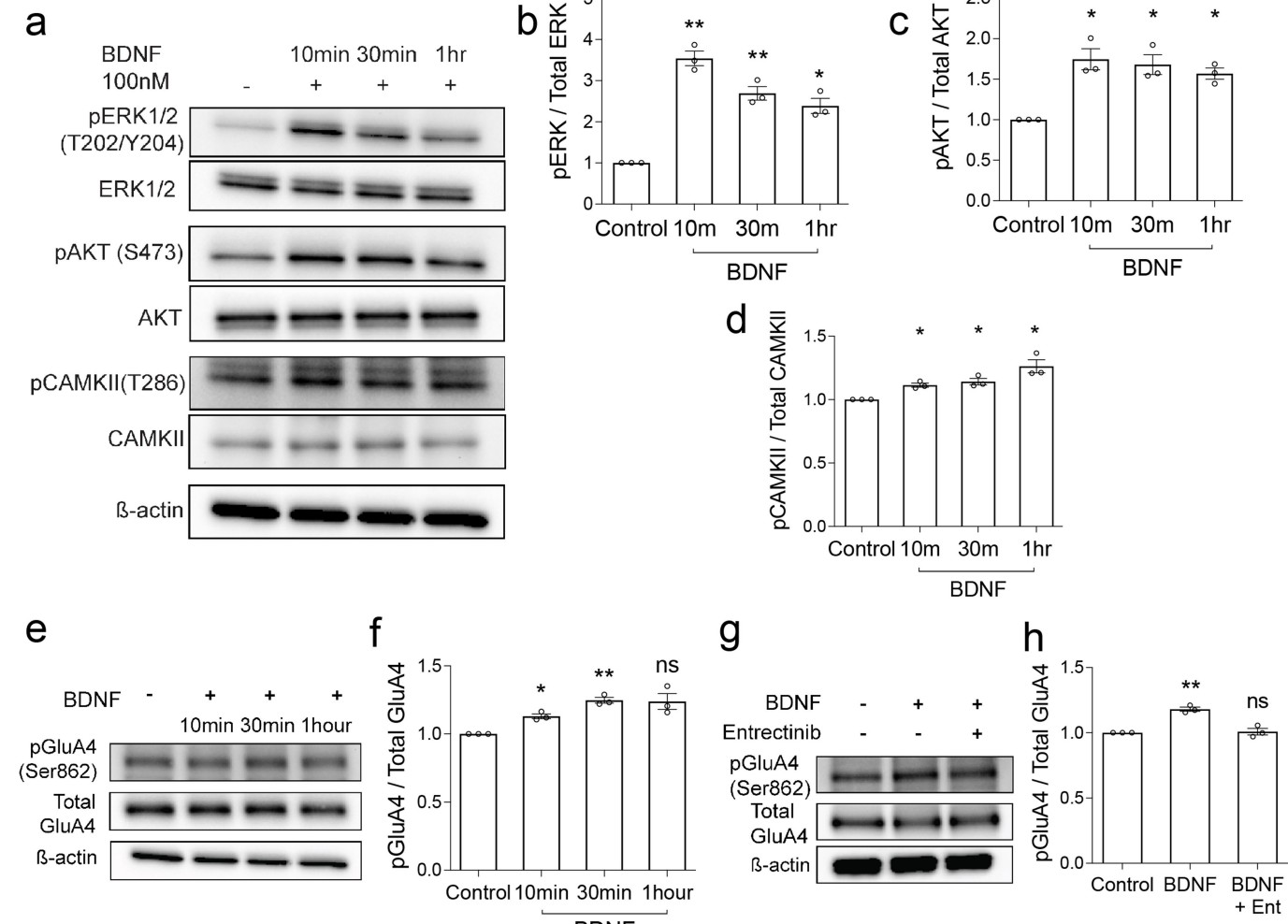

**Extended Data Fig. 8 | BDNF exposure induces PI3K, MAPK and CAMKII activation and AMPAR phosphorylation in pediatric glioma. a**, Western blot of proteins from SU-DIPG-VI cells treated with BDNF recombinant protein (100 nM) over a time course and probed for the indicated antibodies to demonstrate activation of downstream signaling pathways in comparison to untreated cells (vehicle only). **b**, Quantification of MAPK pathway activation in **a**, as the ratio of the normalized phospho-ERK (T202/Y204) levels to corresponding total protein levels for BDNF treated (100 nM) cultures compared to control (y-axis is in arbitrary units, n = 3 independent biological replicates, pERK Control vs 10 m P = 0.0048, Control vs 30 m P = 0.0094, Control vs 1 h P = 0.0167). **c**, Quantification of PI3K pathway activation in **a**, by comparing the ratio of the normalized phospho-AKT (S473) to corresponding total protein levels for BDNF treated (100 nM) cultures compared to control (y-axis is in arbitrary units, n = 3 independent biological replicates, pAKT Control vs 10 m P = 0.0293, Control vs 30 m P = 0.0307, Control vs 1 h P = 0.0148). **d**, Quantification of calcium pathway activation in **a**, by comparing

the ratio of the normalized phospho-CAMKII (T286) to corresponding total protein levels for BDNF treated (100 nM) compared to control (y axis is in arbitrary units, n = 3 independent biological replicates, pCAMKII Control vs 10 m P = 0.0197, Control vs 30 m P = 0.0310, Control vs 1 h P = 0.0374). **e**, Representative Western blot analysis of primary patient-derived glioma culture, SU-DIPG-VI, treated with 100 nM BDNF at several timepoints using indicated antibodies. **f**, Quantification of the phospho-immunoblots ratio to corresponding total protein levels and normalized to vehicle treated control (y axis is in arbitrary units, n = 3 independent biological replicates, pGluA4 Control vs 10 m P = 0.0174, Control vs 30 m P = 0.0072), 8 h (pGluA4 Control vs BDNF P = 0.078). **g**, Representative Western blot analysis of 100 nM BDNF treated glioma cells (as in **e**,) at 30 min with and without entrectinib treatment (Ent, 5 µM). **h**, quantification of phospho-immunoblots (y axis is in arbitrary units, n = 3 independent biological replicates). Data are mean ± s.e.m. *P < 0.05, **P < 0.01, ns = not significant. Two-tailed one Sample t test for **b**, **c**, **d**, **f**, **h**.

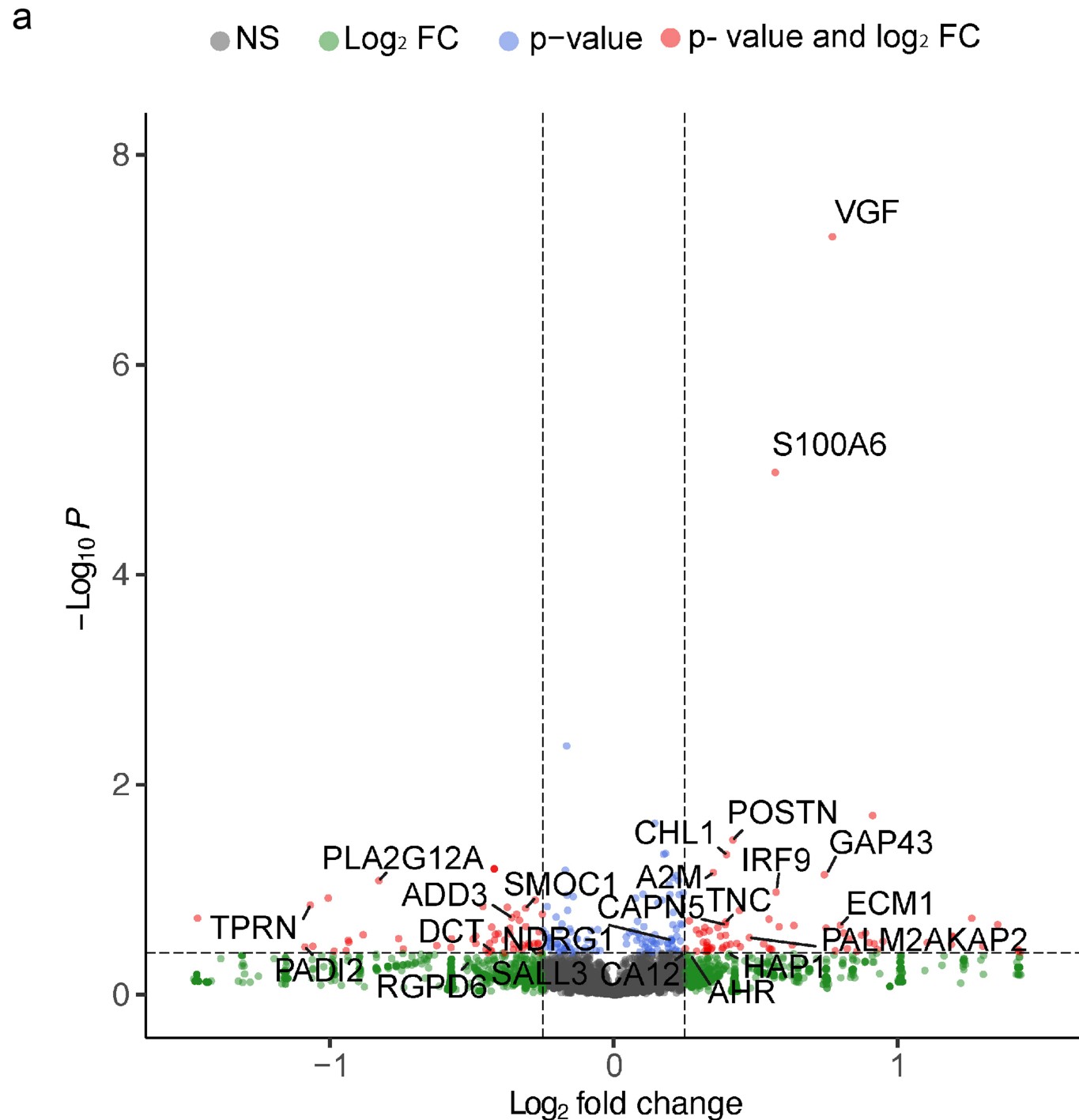

**Extended Data Fig. 9 | Gene expression changes induced in pediatric glioma upon BDNF exposure. a**, Volcano plot demonstrating gene expression changes in SU-DIPG-VI after 16 h of treatment in vitro with and without BDNF recombinant protein (100 nM). The x axis demonstrates the $\log_2$ fold change in gene expression (of BDNF-treated compared to vehicle-treated SU-DIPG-VI samples) and the y axis demonstrates $\log_{10}P$ significance (log10-transformed two-tailed p value significance, as calculated by the Wald test) of the gene expression change following analysis with DESeq2 in R.

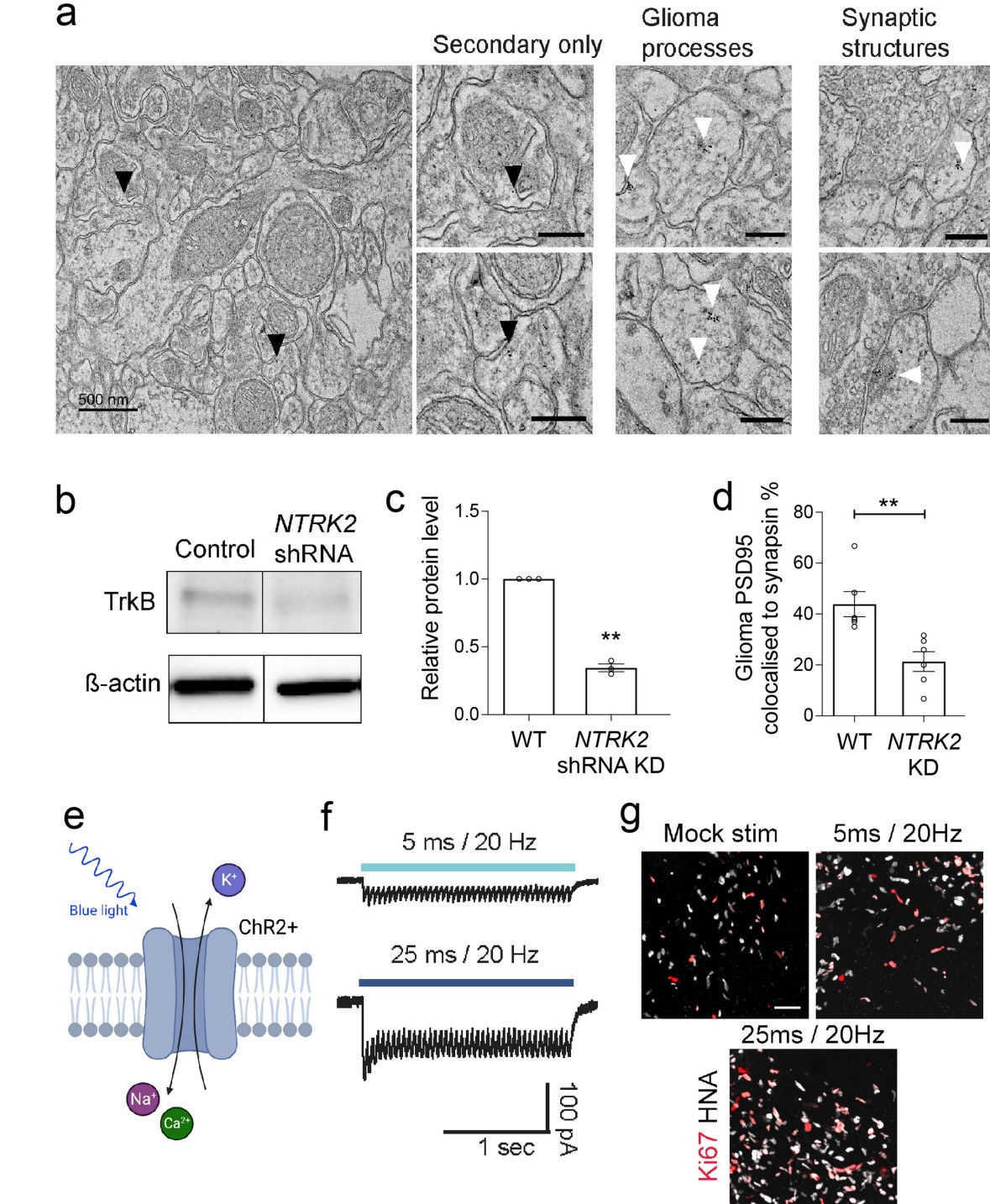

**Extended Data Fig. 10 | *NTRK2* knockdown reduces colocalisation of neuron-to-glioma synaptic puncta, and optogenetic modeling of glioma membrane depolarization. a**, Electron microscopy images of glioma xenografted mouse hippocampal tissue sections with immuno-gold particle labeling of GFP. Left, secondary only stains to show background levels of non-specific gold particle labeling (black arrows). Right, examples of glioma processes and additional examples of neuron-to-glioma synapses positive for >4 immuno-gold particles (white arrows) in patient-derived SU-DIPG-VI cells xenografted to the hippocampus. Scale bar = 500 nm (left), all other scale bars = 200 nm. **b**, Representative western blot analysis of TrkB protein levels in control scramble shRNA and *NTRK2* shRNA knockdown (KD) cultures (SU-DIPG-VI), using indicated antibodies. **c**, Quantification of western blot analysis with levels of TrkB normalized to total protein loading using ß-actin levels and compared to wild-type, Cas9-scramble control, cultures (*y* axis is in arbitrary units, *n* = 3 technical replicates, P < 0.0001). **d**, Quantification of the colocalization of postsynaptic glioma-derived PSD95-RFP with neuronal presynaptic synapsin in co-cultures of wild-type (*n* = 6 cells, 3 coverslips, P = 0.0050), or *NTRK2* KD glioma cells (SU-DIPG-VI, *n* = 6 cells, 3 coverslips); replicates experiment in Fig. 4f, g using shRNA knockdown. **e**, The cation channel, Channelrhodopsin-2, is gated by blue light, inducing membrane depolarization of the cell. **f**, Electrophysiological traces of patch-clamped glioma cells stimulated with 470 nm light at 20 Hz, 1.0 mW/mm2 for 2 s (blue lines) at either 5 ms (light blue) or 25 ms (dark blue) light-pulse width. Note the difference in current amplitude elicited by 5 ms vs 25 ms light-pulse widths. **g**, Representative images of xenografted ChR2+ glioma cells quantified in **4k** after mock stimulation, or optogenetic stimulation at 5 ms and 25 ms light-pulse width, gray denotes HNA-positive glioma cells; red denotes Ki67. Scale bar = 50 μm. Data are mean ± s.e.m. **P < 0.01. Two-tailed one sample *t*-test for **c** and two-tailed unpaired Student's *t*-test for **d**.

# Reporting Summary

## Statistics

For all statistical analyses, confirm that the following items are present in the figure legend, table legend, main text, or Methods section.

| n/a | Confirmed | |
|---|---|---|
| ☐ | ☒ | The exact sample size (*n*) for each experimental group/condition, given as a discrete number and unit of measurement |
| ☐ | ☒ | A statement on whether measurements were taken from distinct samples or whether the same sample was measured repeatedly |
| ☐ | ☒ | The statistical test(s) used AND whether they are one- or two-sided <br> *Only common tests should be described solely by name; describe more complex techniques in the Methods section.* |
| ☐ | ☒ | A description of all covariates tested |
| ☐ | ☒ | A description of any assumptions or corrections, such as tests of normality and adjustment for multiple comparisons |
| ☐ | ☒ | A full description of the statistical parameters including central tendency (e.g. means) or other basic estimates (e.g. regression coefficient) AND variation (e.g. standard deviation) or associated estimates of uncertainty (e.g. confidence intervals) |
| ☐ | ☒ | For null hypothesis testing, the test statistic (e.g. *F*, *t*, *r*) with confidence intervals, effect sizes, degrees of freedom and *P* value noted <br> *Give P values as exact values whenever suitable.* |
| ☒ | ☐ | For Bayesian analysis, information on the choice of priors and Markov chain Monte Carlo settings |
| ☒ | ☐ | For hierarchical and complex designs, identification of the appropriate level for tests and full reporting of outcomes |
| ☐ | ☒ | Estimates of effect sizes (e.g. Cohen's *d*, Pearson's *r*), indicating how they were calculated |

*Our web collection on statistics for biologists contains articles on many of the points above.*

## Software and code

Policy information about availability of computer code

| Data collection | Electrophysiology data were collected with a MultiClamp 700B amplifier (Molecular Devices) and digitzed at 10 kHz with InstruTECH LIH 8+8 data acquisition device (HEKA) and analyzed using AxoGraph X and IgorPro v.8. Calcium imaging data was either collected on a microscope equipped with DIC optics (Olympus BX51WI) and FlyCapture 2.13 release 61 software, or a Prairie Ultima XY upright two-photon microscope equipped with an Olympus LUM Plan FI W/IR-2 40x water immersion objective and Prairie View 5.6 software. Confocal images were acquired on either a Zeiss Airyscan1 LSM800 or Zeiss Airyscan2 LSM980 using Zen 2011 v8.1. |
|---|---|
| Data analysis | Statistical tests were conducted using GraphPad Prism v9.1.0 software for most analysis. For analysis of previously published scRNAseq data, R studio 1.4.1106-5 was used, with packages clusterProfiler v3.18.1, Kallisto v.0.46.1, Tophat2 v2.0.1.3, featurecounts v2.0.3 and DESeq2 v1.36.0 and EnhancedVolcano v1.14.0. Calcium imaging and pHluorin analyses were performed using ImageJ v.2.1.0/1.53c with standard ROI intensity plugin. |

For manuscripts utilizing custom algorithms or software that are central to the research but not yet described in published literature, software must be made available to editors and reviewers. We strongly encourage code deposition in a community repository (e.g. GitHub). See the Nature Portfolio guidelines for submitting code & software for further information.

## Data

Policy information about availability of data

All manuscripts must include a data availability statement. This statement should provide the following information, where applicable:

- Accession codes, unique identifiers, or web links for publicly available datasets
- A description of any restrictions on data availability
- For clinical datasets or third party data, please ensure that the statement adheres to our policy

All unique materials such as patient-derived cell cultures are freely available and can be obtained by contacting the corresponding author with a standard MTA with Stanford University. Single cell and bulk RNAseq data in Extended Data Figure 1, Extended Data Figure 4 and Extended Data Figure 7 were analyzed from publicly available datasets on GEO (GSE102130, GSE134269, GSE94259 and GSE222560). Bulk RNASeq data used in Extended Data Figure 9 is available on GEO (GSE222481). Glioma Patch-Seq data referenced in the rebuttal is available on GEO (GSE222398). The hg38 and the hg19 reference genome was used for transcriptome annotations as stated in the methods. Data for all figures in the manuscript can be found in the Source Data, Supplementary Information and Supplementary Figure 1.

# Field-specific reporting

Please select the one below that is the best fit for your research. If you are not sure, read the appropriate sections before making your selection.

☒ Life sciences        ☐ Behavioural & social sciences        ☐ Ecological, evolutionary & environmental sciences

For a reference copy of the document with all sections, see nature.com/documents/nr-reporting-summary-flat.pdf

# Life sciences study design

All studies must disclose on these points even when the disclosure is negative.

| | |
|---|---|
| Sample size | Based on the variance of xenograft growth in control mice, power calculations indicated use of at least 3 mice per treatment group/genotype to give 80% power to detect an effect size of 20% with a significance level of 0.05. For electrophysiological and calcium imaging studies, all studies were replicated across multiple cohorts of mice to verify reproducibility. |
| Data exclusions | No data were excluded from analyses. |
| Replication | All in vitro experiments have been performed in at least three independent coverslips for each experiment and performed in at least two independent experiments. The number of biological replicates (mice for in vivo growth, calcium imaging, electrophysiological recording, optogenetic stimulation and immuno-electron microscopy experiments) is indicated in the figure legends and was three or greater for all experiments and replicated in at least two independent experiments. |
| Randomization | All mice xenografted with tumor were randomized in assigning to treatment and control groups for all experiments. |
| Blinding | The experimenters performing histological quantifications were blinded to group allocation. |

# Reporting for specific materials, systems and methods

We require information from authors about some types of materials, experimental systems and methods used in many studies. Here, indicate whether each material, system or method listed is relevant to your study. If you are not sure if a list item applies to your research, read the appropriate section before selecting a response.

### Materials & experimental systems

| n/a | Involved in the study |
|---|---|
| ☐ | ☒ Antibodies |
| ☐ | ☒ Eukaryotic cell lines |
| ☒ | ☐ Palaeontology and archaeology |
| ☐ | ☒ Animals and other organisms |
| ☒ | ☐ Human research participants |
| ☒ | ☐ Clinical data |
| ☒ | ☐ Dual use research of concern |

### Methods

| n/a | Involved in the study |
|---|---|
| ☒ | ☐ ChIP-seq |
| ☒ | ☐ Flow cytometry |
| ☒ | ☐ MRI-based neuroimaging |

## Antibodies

| | |
|---|---|
| Antibodies used | Primary antibodies used in immunohistochemistry: chicken anti-GFP (1:500, #ab13970, Lot #GR3361051), mouse anti-Human Nuclei, clone 235-1 (1:200, MAB1281, Lot #3543073), guinea pig anti-Synapsin (1:500, #106-004 Synaptic Systems), rabbit anti-RFP (1:500, #600-401-379, Lot # 42872), mouse anti-nestin (1:500, #ab6320, Abcam) chicken anti-Neurofilament-H (1:1000, #NFH, |

Lot#NFH957980), chicken anti-Neurofilament-M (1:1000, #NFM, Lot#NFM87837981), rabbit anti-microtubule-associated protein 2 (1:500, MAP2; EMD Millipore, #AB5622, LOT# 3897281), rabbit anti-Ki67 (1:500, ab15580, Abcam, LOT#GR3445754-1).
For secondary antibodies: Alexa 488 donkey anti-mouse IgG (#715-545-150), Alexa 594 donkey anti-rabbit IgG (#711-585-152), Alexa 647 donkey anti-mouse IgG (#715-605-150), Alexa 406 donkey anti-guinea pig IgG (#706-475-148) all used at 1:500 (Jackson Immuno Research).
For antibodies used in western blot: anti-GluA4 (#8070, Lot# 1), anti-GluA3 (#4676, Lot#1), anti-TrkB (#4606, Lot# 1), anti-GAPDH (#5174, Lot# 8), anti-beta-actin (#4970, Lot# 15), anti-phospho Erk (#4370, Lot# 17), anti-Erk (#9102, Lot# 26), anti-phospho Akt (#4060, Lot# 25), anti-Akt (#9272, Lot# 28), anti-phospho CAMKII (#12716, Lot# 4), anti-CAMKII (#4436, Lot# 3) all rabbit primary antibodies used at 1:1000 from Cell Signaling Technologies. Antibodies sourced from other companies were: rabbit anti_phospho-GluA4 (#PA5-36807, Invitrogen, Lot# SI2447842A) and rabbit anti-phospho TrkB (#SAB4503785, Lot# 110035).

| Validation | All antibodies have been validated either in the literature, by manufacturers and/or in Antibodypedia for use in mouse immunohistochemistry and human western blot. To further validate the antibodies in our hands, we confirmed each antibody had either staining in the expected cellular patterns and brain-wide distributions for immunohistochemistry or for western blot, bands appearing at the correct weight according to the protein ladders run alongside the samples (LI-COR Chemiluminescent Protein Ladder (#NC0986471, Fisher Scientific) and Precision Plus Protein Dual Color Standards (#1610374, Bio-Rad).

Antibody validation information : Chicken anti-GFP (#ab13970, Reactivity : Species independent. Manufacturer validation: ICC: GFP-transfected NIH/3T3 Mouse embryo fibroblast cell line. Publication Figure : ExtData Fig1a and 6a, IF of human DIPG xenograft), mouse anti-Human Nuclei, clone 235-1 (MAB1281, Reactivity : human only, Manufacturer validation: IF neural stem cells transplanted into rat brain. Publication Figure : Fig1c, 1l, ExtData Fig5b, 5d, 6a, 10g, IF of human DIPG xenograft), guinea pig anti-Synapsin (#106-004, Reactivity : human, mouse, hamster, cow, zebrafish. Manufacturer validation: ICC immunostaining of PFA fixed rat hippocampus neurons. Publication Figure : 4e, IF of mouse neuron culture), rabbit anti-RFP (#600-401-379, Reactivity : RFP, Manufacturer validation: IF HopERCre/+; R26Tom/+ mice tissue. Publication Figure : 4e, IF of patient-derived PSD95-RFP+ DIPG culture), mouse anti-nestin (#ab6320, Reactivity : human, Manufacturer validation: IF human fetal neural progenitor cells, Publication Figure : Fig1i IF of human DIPG culture), chicken anti-Neurofilament-H (#NFH, Reactivity : human, mouse, rat, chicken, Manufacturer validation: IF rat cortical neurons and glia. Publication Figure : 4e, IF of mouse neuron culture), chicken anti-Neurofilament-M (#NFM, Reactivity : human, mouse, rat, bovine, chicken. Publication Figure : 4e, IF of mouse neuron culture), rabbit anti-microtubule-associated protein 2 (#AB5622, Reactivity : human, mouse, rat. Manufacturer validation: culture Rat hippocampal neurons. Publication Figure : 5b and 5d, IF of mouse neuron culture), rabbit anti-Ki67 (ab15580, Reactivity : mouse, human. Manufacturer validation: IF in Mef1 and HeLa cultures, Human skin tissue. Publication Figure : Fig1c, 1l, ExtData Fig 10g, IF of human DIPG xenograft). All primary antibodies used for IF in this publication have been used and validated in similar experimental paradigms previously (Venkatesh et al, Nature, 2019).

For antibodies used in western blot: anti-GluA4 (#8070, Reactivity : human, mouse, rat. Manufacturer validation: WB analysis of extracts from mouse brain, NIH/3T3 cells, rat brain, and C6 cells. Publication Figure : Fig3b, 3d, 3f and ExtData Fig. 6e, 6g WB analysis of human DIPG cell culture), anti-GluA3 (#4676, Reactivity : human, mouse, rat. Manufacturer validation: WB analysis of extracts from mouse and rat brains. Publication Figure : Fig3d, WB analysis of human DIPG cell culture), anti-TrkB (#4606, Reactivity : human. Manufacturer validation: Western blot analysis of extracts from NIH/3T3, NIH/3T3-TrkA, NIH/3T3-TrkB and NIH/3T3-TrkC cells. Publication Figure : ExtData Fig 1g, 2a, 10b, WB analysis of human DIPG cell culture), anti-GAPDH (#5174, Reactivity : human, mouse, rat, monkey. Manufacturer validation: Western blot analysis of extracts from HeLa, NIH/3T3, C6 and COS-7 cells. Publication Figure : Fig 3b, 3d, 3f , WB analysis of human DIPG cell culture), anti-beta-actin (#4970, Reactivity : human, mouse, rat, monkey, bovine, pig. Manufacturer validation: Western blot analysis of extracts from HeLa, NIH/3T3, C6 and COS-7 cells. Publication Figure : ExtData Fig 1g, 2a, 10b, 8a, 8e, 8g, 10b, WB analysis of human DIPG cell culture), anti-phospho Erk (#4370, Reactivity : human, mouse, rat, monkey. Manufacturer validation: Western blot analysis of extracts from NIH/3T3, C6 and COS-7 cells. Publication Figure : ExtData Fig : 2a, 8a, WB analysis of human DIPG cell culture), anti-Erk (#9102, Reactivity : human, mouse, rat. Manufacturer validation: Western blot analysis of extracts from HeLa cells. Publication Figure : ExtData Fig : 2a, 8a, WB analysis of human DIPG cell culture), anti-phospho Akt (#4060, Reactivity : human, mouse, rat. Manufacturer validation: Western blot analysis of extracts from PC-3 cells. Publication Figure : ExtData Fig : 8a, WB analysis of human DIPG cell culture), anti-Akt (#9272, Reactivity : human, mouse, rat. Manufacturer validation: Western blot analysis of extracts from CHO cells. Publication Figure : ExtData Fig : 8a, WB analysis of human DIPG cell culture), anti-phospho CAMKII (#12716, Reactivity : human, mouse, rat. Manufacturer validation: Western blot analysis of extracts from MKN-45 cells. Publication Figure : ExtData Fig : 8a, WB analysis of human DIPG cell culture), anti-CAMKII (#4436, Reactivity : human, mouse, rat. Manufacturer validation: Western blot analysis of extracts from 293T cells. Publication Figure : ExtData Fig : 8a, WB analysis of human DIPG cell culture), rabbit anti_phospho-GluA4 (#PA5-36807, Reactivity : human, mouse, rat. Manufacturer validation: Western blot analysis of extracts from HEK-293T cells. Publication Figure : ExtData Fig : 8e, 8g, WB analysis of human DIPG cell culture) and rabbit anti-phospho TrkB (#SAB4503785, Reactivity : human, mouse, rat. Manufacturer validation: Western blot analysis of extracts from NIH-3T3 cells. Publication Figure : ExtData Fig : 2a, 8a, WB analysis of human DIPG cell culture). |

# Eukaryotic cell lines

Policy information about cell lines

| Cell line source(s) | The eukaryotic cell cultures used are patient-derived cultures of high-grade gliomas generated in the Monje lab from biopsy (SU-pcGBM2) or autopsy tissue (SU-DIPG-VI and SU-DIPGXIII-FL/P*), or provided by collaborators from a biopsy (QCTB-R059). 293T cells (ATCC) were used only for virus production. |
| Authentication | Short Tandem Repeat (STR) fingerprinting is performed every 3 months on all cell cultures to ensure authenticity. |
| Mycoplasma contamination | All cell cultures are routinely tested for mycoplasma contamination and all cultures were tested negative. |
| Commonly misidentified lines (See ICLAC register) | No commonly misidentified lines were used. |

## Animals and other organisms

Policy information about studies involving animals; ARRIVE guidelines recommended for reporting animal research

Laboratory animals

All animal experiments were conducted in accordance with protocols approved by the Stanford University Institutional Animal Care and Use Committee (IACUC) and performed in accordance with institutional guidelines. Animals were housed according to standard guidelines with unlimited access to water and food, under a 12 hour light : 12 hour dark cycle, a temperature of 69.7F and 60% humidity. Both male and female NOD-SCID-IL2R gamma chain-deficient (NSG) were used between 1-7months of age. BDNF-TMKI mice were back-bred to NSG mice to facilitate orthotopic xenografting and used at 4-12 weeks of age.

Wild animals

No wild animals were used.

Field-collected samples

No field-collected samples were used.

Ethics oversight

All animal experiments were conducted in accordance with protocols approved by the Stanford University Institutional Animal Care and Use Committee (IACUC) and performed in accordance with institutional guidelines. The IACUC implements regulations from the United States Department of Agriculture (USDA), the Public Health Service (PHS) Policy, California State Regulations and Stanford University Polices and Guidlines to ensure effective and ethical animal research programs.

Note that full information on the approval of the study protocol must also be provided in the manuscript.

