## [Peer Review File · Nature]

Manuscript Title: Glioma synapses recruit mechanisms of adaptive plasticity

Reviewer Comments & Author Rebuttals

Reviewer Reports on the Initial Version:

Referees' comments:

Referee #1 (Remarks to the Author):

In this very well-written manuscript, the authors build on the previous work with the discovery of neuron-glioma synapses (2019) and describe mechanisms contributing to the plasticity of these synapses, involving a BDNF-NTRK2 signaling axis – a mechanism that appears likely to drive malignancy in glioma. The general concept presented here is exciting and is an additional piece in the puzzle of how brain tumor cells use, or hijack, neuronal and neurodevelopmental mechanisms to thrive, and how integration into neuronal circuits is effectively achieved.

The experiments are carefully performed and well presented, and adequate preclinical and clinical models are selected to answer the relevant questions, with really interesting methodological approaches (e.g., live imaging of synaptic plasticity). The complexity of the system – paracrine effects, synaptic effects, tumor cell autonomous network effects, etc.; at least the first two even in parallel for the same molecular pathway investigated here – is certainly a fundamental challenge that is not easy to resolve when it comes to proving the exact role of a distinct mechanism. Along this line, I recommend to consider the following points that will help to further strengthen the main message and impact of this study.

I see three major points that are particularly important:

1.) It needs to be shown how the heterogeneity of glioblastoma cells comes into play and especially which subpopulations are electrophysiologically characterised with regard to glioma cells showing the previously described two types of inward currents.

Previously, the group reported that there are two types of inward currents (EPSCs, and slow inward currents). The currents shown in panel 1e, 3a etc. are kinetically more similar to slow inward currents which the authors concluded to be driven by potassium currents (Venkatesh et al., Nature 2019). The currents that are now seen are driven by glutamate puffing (which has not been performed in Venkatesh et al. 2019) and could be in principle be driven by AMPA receptors, NMDA receptors or glutamate transporters, if they are directly mediated by the puff. First, what is the latency to response after glutamate puffing? In other words, is it a direct response to glutamate or is it possibly an indirect effect? The kinetic of the response needs to be explored pharmacologically with experimental paradigms where AMPAR inhibitors, NMDA inhibitors and glutamate transporter inhibitors such as TFB-TBOA are applied and washed out. In which relation do the glutamate-evoked inward currents stand with the previously electrically evoked slow inward currents and EPSCs? Is there a subpopulation that is modulated by BDNF and one that is not modulated by BDNF? Previously, it has been reported that glioma cells express mechanoreceptors. Puffing with ACSF has to be shown as control to exclude the possibility that a subpopulation of the currents is mechanically provoked. Please also report how the glutamate was exactly applied (was a Picospritzer used? how many PSI were applied?).

2.) It is well understood that it is challenging to decipher the specific pathobiological role of BDNF-NTRK signaling on synaptic plasticity of the neuron-glioma cell synapse vs. paracrine and other effects of this pathway. The authors try hard to do that in Fig. 4, which however requires inter-experiment comparisons (most notably, 4b vs 4c, and in vitro vs in vivo experiments). To further strengthen the concept of a specific biological role of malignant synaptic plasticity for glioma biology, the authors should aim to provide more data using other regulators (stimulation paradigms, stimulators, inhibitors) of general synaptic plasticity. Moreover, they might want to consider to use their elegant mouse model deficient in activity-induced BDNF expression/secretion more, e.g. by growing a control vs NTRK2 KO glioma line. Here, an additive effect of the KO on survival should be missing. This does not exclude paracrine effects but can help to better control for unrelated effects occurring in this system.

3.) Another important question is whether malignant synaptic plasticity is exclusively regulated by the BDNF-NTRK axis, or whether other plausible candidates, most notably NLGN3, have also an effect. This would be an important piece of information and would help to better develop a broader picture of malignant synaptic plasticity in brain tumor pathophysiology.

Other major points:

4.) The top genes associated with NTRK2 expression in pediatric gliomas (Ext Data Fig. 1f) are very interesting. No. 1 is GJA1, which encodes Cx43 – which is THE gap junction protein that is responsible for glioma cell network connectivity/integration, which in turn was found for those glioma cells displaying slow inward currents in the 2019 study of the authors. That might explain the exclusive slow inward current (-like) patch clamp recordings shown in this study; and stresses the importance of understanding the heterogeneous electrical responses better, including its tumor-biological fundamentals. – In addition, and remarkably, No 3 is Ttyh1, which has been described as a key molecular driver for INVASIVE (not interconnecting) tumor microtubes in glioma – which implies the question how synaptic plasticity is associated with tumor cell invasiveness. Looking at this data in synopsis, it is quite likely that both genes play important (co-) roles for NTRK2-high glioma cells, but in distinct subpopulations of glioma cells. 1. Is that indeed the case (should be answered easily with this single cell datasets?). 2. The authors might also want to consider to investigate both genes in more detail with respect to the main findings of this manuscript.

5.) Likewise, the gene expression changes induced by BDNF treatment (Ext Data Fig. 5) show a pattern that is very well fitting to a highly tumor microtube (network) - proficient one (GAP-43; VGF etc.). The authors might want to discuss this aspect (and modify the presentation in the results section a bit, accordingly).

6.) 1k, Please also report the heterogeneity of calcium signals and what kind of subpopulations can be found with glutamate-evoked calcium currents.

7.) Only the AMPAR subunit GluA4 was investigated after BDNF treatment. What about the other subunits? It would be most convincing if all (or at least one more) subunit(s) would also increasingly locate to the cell membrane.

8.) The live imaging system of AMPAR tracking (Fig. 2d) is indeed a very interesting system. In contrast to the biochemical assays used for analysis of the surface proteins this system can be used to explore heterogeneity on multiple levels, which can potentially be improved. Furthermore, the relationship between these signals and electrophysiological response should be correlated. Furthermore, the number of observations needs to be significantly increased to clearly show how this is related to glioma cell heterogeneity. 2f, The time course is very interesting. It would be interesting if this could also be functionally shown with at least calcium imaging, ideally also

electrophysiology if the patches are stable enough.

9.) 3c, The labelling used for electron microscopy is not completely convincing. Gold particles are clustered, often a sign of unspecific labelling and a lot of gold particles can be seen outside the putative glioma cell postsynaptic side. Although a postsynaptic (?) density can be clearly seen in both examples shown, a vesicle cluster of the presynaptic bouton is difficult to distinguish. What criterions were used to be qualified as glioma synapse? Please show more examples at least in an Extended Figure. Control tissue needs to be used to unequivocally show specific binding of immunogold. It is unclear how the quantification was done and whether this method can be used to truly quantify connectivity. Ideally, large whole-cell reconstructions would need to be performed to comment on connectivity which is clearly beyond the scope of the current study. However, a stereological approach should be used to comment on differences between both groups. In addition, please report the exact number of observations made here (how many processes? how many somata were observed? how do these numbers relate?).

Minor points:

Figure 1a Description: Too elaborate - no long explanations needed.

1c: The morphology of the shown cells looks disrupted. HNA signal cannot be clearly detected from the shown image.

1e: Please report how many cells were overall patched, how many cells were responsive to glutamate.

1i, Please show time series as visualization. It would be better to use an example where the exact cell morphology can be deduced from. In this example either the SNR was not good or the cells were out of focus.

3e, Please comment on the exact methods of analysis of these imaging data. How was colocalization quantified? What was counted as puncta? Why can only one synapsin cluster be detected in this field of view? The overall synaptic density seems rather low in this model system. How does this relate to xenograft or even human tissue?

Line 206: should read 4b,c

The finding that NTRK inhibitors are effective in non-NTRK-fusion gliomas (partially or completely by modulating malignant synaptic plasticity) is a very interesting and potentially clinically relevant one. The authors could discuss these implications in the results or discussion section.

Referee #2 (Remarks to the Author):

The manuscript titled "Glioma synapses recruit mechanisms of adaptive plasticity" by Taylor et al. from the laboratory of Dr Michelle Monje builds on the exciting line of investigation from this laboratory over the last few years that has begun to elucidate mechanism relating glioma/neuron interactions. The current study probes BDNF/trkB and neuron-glioma signaling. Trk is highly expressed in many pediatric gliomas. The findings demonstrating that knocking down (they reduce expression by 80% or so) or inhibiting trk signaling increases survival in models are interesting and potentially important. However, the authors don't clearly demonstrate that the adaptive plasticity mechanisms referred to in the title impact tumor function or survival. They hint the importance of non-growth factor trk signaling in tumor regulation but do not determine what this signaling is, only showing that trk signaling can recruit AMPARs in models, but never closing the loop to show that this recruitment is linked to clinical outcome.

The authors use the drug entrectinib to inhibit trk signaling. ntractinib inhibits trk function broadly, targeting A, B and C, but the effects on survival are modest, apparently less effective than NTRK2 knockdown. The authors should test the selectivity of their drug effects by testing whether there is any effect on survival, or other assays of the drug in combination with NTRK2 knockdown.

The authors propose that trk activation has both direct effects on glioma growth and indirect effects via AMPARs. This idea should be tested by examining whether blocking AMPARs together with trk signaling further limits glioma growth or other negative impacts. Without a more direct test of the link between trk and AMPARs the study is interesting but somewhat descriptive.

It is surprising that the authors do not detect GAPDH in their cell surface experiment as numerous reports indicate that this protein is found extracellularly. Indeed, GAPDH localized to the membrane, the nucleus, polysomes, the ER and the Golgi. The data shown in figure 2, where no GAPDH is detected in the putative cell surface fraction raise significant methodological concerns about this experiment. What fraction is this? What is the explanation for this result? These data are not convincing. New experiments and an explanation of how this result was obtained are needed. A better control would be actin.

Figure 3 examines the effects of crispr mediated reduction of Trk expression impacts synapses between neurons and glioma in the authors in vitro model. The statistical power of these experiments is very low with n's under ten cells. The number of processes examined is not reported for c-d. Remarkably, the authors report effects from only six cells in e-g. This is well below standards. Moreover, the authors fail to show examples of both control and knockdown. It is unclear whether these effects would be consistent in large data sets.

Given previous work by these authors linking trk signaling to NLGNs in gliomas is somewhat unexpected that the authors did not examine whether these signaling pathways might intersect here.

Referee #3 (Remarks to the Author):

The work by Taylor et al. is a follow up of their previous study also published in Nature, which described excitatory synaptic formation of glioma cells. In the new paper, the authors tried to address the mechanism that regulates synaptic glutamate receptor and found BDNF-TrkB signaling mediates this. Then they went further to test the effect of BDNF on the tumor cell growth with a final aim to develop effective treatment of this intractable disease. So this work has clinical relevance.

Said this, I found this work has a major disruption of logical flow. In the first half, the authors made effort to establish how glioma malignant synapse is regulated. Then in the rest of study, they are testing if BDNF signaling is involved in the proliferation of glioma cells. Indeed, they found that blocking BDNF signaling slows down the proliferation. However, the authors failed to provide a convincing evidence that the effect is mediated by glutamatergic synapse. TrkB activation can trigger number of different signaling pathways and AMPAR may be one of them. But there is no evidence that the effect of blockade of BDNF signaling is mediated by the blockade of glutamatergic synapse. I therefore, cannot recommend publication of this work in Nature. In practice, the authors should consider splitting the story into two papers.

Minor comments.

It is hard to discern what structures are shown. For example, in Fig. 1i, what part of cells are shown? Need scale bar. Fig. 2D. What is the beads-like structure? Single en passant axon? Fig. 2F as well. Low magnification images might help.

Fig. 2H, I, S4D. The effect of BDNF is so small. Although there is statistical difference, it is hard to imaging such a small activation of a kinase is functionally meaningful.

GluR4 should be used for pHluorin imaging. Also, S862A mutant should be tested.

Referee #4 (Remarks to the Author):

Previous studies by the authors' group and others suggested that synaptic interactions between neurons and glioma cells play a key role in glioma progression (Venkatesh et al, 2019; Venkataramani et al, 2019; etc). In this manuscript, the authors attempted to extend these previous studies and examine whether the neuron-to-glioma malignant synapses are regulated by BDNF-TrkB signaling, a key regulator of synaptic plasticity. The authors provide the data showing that BDNF promotes AMPA receptor trafficking to the glioma cell membrane, resulting in increased amplitude of glutamate-evoked currents in the malignant cells. BDNF-TrkB signaling also regulates the number of neuron-to-glioma synapses. They also showed that blocking TrkB signaling attenuated tumor growth. BDNF regulation of synaptic plasticity including structural (spine growth) and functional (transmitter release, AMPA receptor trafficking) plasticity has been well established. It is therefore not surprising that similar mechanisms are used in neuron-glia synapses. The pan-Trk inhibitors, e.g. entrectinib used in therapeutic targeting of TrkB in pediatric glioma of this study, have also been approved by FDA to treat cancer, including gliomas. Thus, the current study has not reached the level of novelty and significance needed for Nature. Further, the key is to demonstrate that BDNF enhances glioma progression by regulating neuron-glioma synapses specifically, rather than promoting glioma cell growth per se. The experiments using entrectinib in vivo does not prove that the effects are mediated by inhibition of neuron-glia synapses.

Additional major comments:

1. A key experimental setting is the human glioma cells xenografted onto mouse hippocampal slices. This is an artificial system that may not reflect the real situation in the brain of glioma patients. The mouse hippocampal CA1 neurons would presumably sprout their axon terminals to form synapses with the cultured human glioma cells. It is unclear whether these mouse-human synapses in vitro share the same properties, plasticity and regulatory mechanisms as the neuron-glioma synapses in patients' brain in vivo. Single cell gene expression profiling of cells in the xenograft model may help determine whether presynaptic neurons and postsynaptic glioma cells exhibit similar features as those in human glioma in vivo. Ex vivo electrophysiology experiments using surgically derived human glioma tissues containing neurons and glioma cells might also be helpful.
2. It might be incorrect to use the term 'LTP' here. The classic LTP experiments involve a long-term (hours) enhancement of synaptic connections between presynaptic terminals and postsynaptic cells. Here a glutamate puffing instead of stimulation of presynaptic neurons was used to induce currents (non-synaptic) in glioma cells. There was no evidence of its NMDAR and Ca²⁺ influx dependence. AMPA receptor insertion into glioma cell membranes without NMDAR may be irrelevant to LTP. Thus, a transient enhancement of the glutamate-induced currents seen in Figure 1, albeit its involvement of AMPA receptor insertion, is far from the Hebbian type activity-dependent synaptic potentiation.
3. One also needs to distinguish between "BDNF enhancement of basal synaptic transmission" and "BDNF regulation of LTP" (Kang et al, 1995; Figorov, 1996; Patterson, 1996; Korte, 1995; see Ji et al 2009 for in depth analyses). The form of plasticity described here at the best is "BDNF-enhancement of glutamate-puff induced currents".
4. The Figure 4 a-d showed that BDNF alone increased glioma proliferation by 20-30%, but addition of neurons to the glioma culture increased proliferation by 30-60%, and this effect is attenuated in glioma with TrkB gene deletion. The authors interpreted this result as neuron-glioma synapse

playing additional role in glioma proliferation. However, it is well known that neurons are the key source of BDNF, and it is difficult to rule out whether addition of neurons to glioma culture was simply adding more BDNF to the culture. To establish neuron-glioma synapse is important, one needs to demonstrate that blockade of synaptic transmission (e.g. using AMPA receptor antagonists or botulinum toxins) could abrogate the effects of adding neurons to glioma culture.

5. Figure 4h showed that mice xenografted with glioma cells bearing TrkB KO or treatment with Trk inhibitors survived longer than those xenografted with WT glioma cells. This could simply be interpreted as BDNF-TrkB signaling is important for glioma growth or proliferation, and has nothing to do with its regulation of neuron-glioma synapses – a key point of this manuscript.

Minor points:

1. There is no data showing if BDNF treatment can enhance basal GCaMP6 fluorescence.
2. The citation of references contains many errors. For example, Kang et al showed that BDNF enhances basal synaptic transmission (1995). BDNF regulation of hippocampal LTP was demonstrated by Figurov (1996), Korte (1995), Patterson (1996).
3. One also wonders how calcium enters into glioma cells after glutamate puffing. There is a need to demonstrate the expression of NMDAR or calcium channels on the cell surface of glioma cells.
4. In Fig.1, it is unclear whether or not glioma itself has AMPA receptor without contacting the hippocampal tissues.
5. The use of the pan-Trk inhibitor Entrectinib may block NGF, and NT3 signaling, rather than BDNF-TrkB.
6. In Fig. 4e-f, one needs to show that it is truly BDNF but not other factors in the conditional media that stimulated proliferation. Similarly, one needs to show that it was the lack of BDNF but not other factors from the Bdnf^{TMKI} xenografts that prolonged the survival of mice (Fig. 4g).
7. The author claimed that BDNF-TrkB signaling promotes calcium-permeable AMPA receptor trafficking and consequently depolarizes the glioma cell membrane. It is unclear whether or not voltage-gated calcium channels are involved in increased intracellular calcium signaling.

Errors:

1. page 7, line 153 and 154, VGF and GBM should be shown in full name when first time presented. page 8, line 190, it should display full name for DMG. page 16, line 333, full name should be mentioned first shown for SU-DIPGVI.
2. page 19, line 373-374, it should be: "blue denotes nestin staining", and "green denotes synapsin", respectively.
3. page 27, line 460, "X-axis" should be changed to "Y-axis"; page 29, line 485, "the x axis" should be changed to "the y axis".

Author Rebuttals to Initial Comments:

We were delighted to see the Referees' positive comments and enthusiasm for the manuscript and are grateful for the careful review, helpful suggestions and insightful questions. We have worked to address the Referees' comments and suggestions in full, which we feel have improved and strengthened the manuscript. Below we will present in detail the major changes and then will respond to the Referee comments point-by-point.

Major changes and new data:

1. We have changed the flow of the story to make it more cohesive. We appreciate the feedback that the original storytelling seemed like two related stories and believe that we have now addressed that concern.

2. We have now performed *in vivo* optogenetic experiments to test the relative contribution of activity-regulated BDNF to activity-regulated glioma proliferation, stimulating cortical projection (glutamatergic) neuronal activity in WT and BDNF-TMKI mice. We observed the expected increase in glioma proliferation in WT mice following optogenetic stimulation of cortical projection neuronal activity, but the proliferative effect of glutamatergic neuronal activity on glioma proliferation was markedly attenuated in mice lacking activity-regulated BDNF expression and secretion. (Figure 1a-d and Extended Data 1c-d).

Figure 1a-d. **a**, Schematic of Bdnf-TMKI mouse, which lacks activity-regulated *BDNF* expression. **b**, Optogenetic model for optogenetic stimulation of ChR2-expressing neurons (blue) in microenvironment of glioma xenograft (green); light blue rectangle denotes region of analysis. P, postnatal day. **c**, Representative image of glioma cells (SU-DIPG-VI) xenografted into wild-type and Bdnf-TMKI NOD-SCID-gamma (NSG) mice in the presence of optogenetically stimulated neurons quantified in **d**, gray denotes human nuclear antigen (HNA)-positive glioma cells; red denotes Ki67 (proliferative marker). Scale bar = 50µm. **d**, Proliferation index of SU-DIPG-VI glioma xenografted to mice with Thy1+ glutamatergic cortical projection neurons lacking (ChR2-) or expressing Channelrhodopsin (ChR2+) in a wild-type or Bdnf-TMKI genetic background (quantified by Ki67+/HNA, n = 6 wild-type ChR2- mice, 4 Bdnf-TMKI ChR2- mice, 7 wild-type ChR2+ mice, 4 Bdnf-TMKI ChR2+ mice).

Extended Data Fig. 1c-d. **c** Model for optogenetic stimulation of ChR2-expressing neurons (blue) in microenvironment of glioma xenograft (green); light blue rectangle denotes region of analysis. P, postnatal

day. **d**, Proliferation index of SU-DIPGXIIIIFL glioma xenografted to mice with neurons expressing Channelrhodopsin (ChR2+) in a wild-type or Bdnf-TMKI genetic background (Figure 1a) after neuronal optogenetic stimulation (quantified by confocal microscopy of EdU+/HNA cells, $n = 7$ wild-type ChR2+ mice, 8 Bdnf-TMKI ChR2+ mice).

3. As a further control, we stimulated cortical projection neuronal activity in WT and BDNF-TMKI mice but with *NTRK2* KO gliomas. We found a similar proliferation rate in *NTRK2* KO glioma-xenografts with or without activity-regulated BDNF, indicating that loss of activity-regulated BDNF does not exert effects that are independent of glioma TrkB signaling. (Extended Data Figure 1k).

Extended Data Fig 1k. Proliferation index of *NTRK2* KO SU-DIPG-VI glioma xenografted to mice with neurons expressing Channelrhodopsin (ChR2+) in a wild-type or Bdnf-TMKI genetic background after neuronal optogenetic stimulation (quantified by confocal microscopy of EdU+/HNA cells, $n = 5$ wild-type ChR2+ mice, $n = 4$ BDNF-TMKI ChR2+ mice).

4. To explore the hypothesis that the growth-inhibitory effects of activity-regulated BDNF-TrkB signaling in glioma may involve modulation of synaptic biology, we asked whether effects of glioma TrkB signaling are related to or independent of AMPA receptor signaling. We found that pharmacologically blocking AMPA receptors, or genetically blocking TrkB through *NTRK2* KO, each decreased tumor cell proliferation *in vivo* or in neuron-to glioma co-culture (Figure 1 j-m, Extended Data Figure 5a). However, we found no additive effect of blocking AMPA receptors and TrkB in co-culture or *in vivo*, suggesting that the mechanisms may be related.

Figure 1j-m. **j**, Proliferation index of SU-DIPG-VI WT and *NTRK2* KO glioma co-culture with neurons, in the presence and absence of the AMPAR blocker NBQX (10 μ M; $n = 3$ coverslips/group, experiment also repeated in Extended Data Figure 5a). **k**, Experimental model of pontine injected WT and *NTRK2* KO

glioma (SU-DIPG-VI) treated with the AMPAR blocker perampanel or vehicle control. **l**, Representative image of xenografted wild-type glioma cells treated with vehicle, and *NTRK2* KO cells treated with the AMPAR blocker perampanel quantified in **m**, gray denotes HNA positive glioma cells; red denotes Ki67. Scale bar = 50 μ m. **m**, Proliferation rate of wild-type (WT) and *NTRK2* KO glioma xenografts (SU-DIPG-VI) treated with the AMPAR blocker perampanel or vehicle control (quantified by Ki67⁺/HNA⁺; WT + vehicle; *n* = 6 mice, WT + perampanel; *n* = 7 mice, *NTRK2* KO + vehicle; *n* = 5 mice, *NTRK2* KO + perampanel; *n* = 6 mice).

Extended Data Fig 5a. Proliferation index of SU-DIPG-VI WT and *NTRK2* KO glioma monoculture (left), or glioma co-culture with neurons (right), in the presence and absence of the AMPAR blocker NBQX (10 μ M) (quantified as fraction of EdU⁺/HNA⁺ co-positive tumor cells assessed by confocal microscopy, *n* = 3 coverslips/group for glioma monoculture experiments and 6 coverslips/group for neuron-glioma co-culture; experiment also replicated in Figure 1j).

5. To demonstrate that Trk inhibitors are exerting therapeutic benefit through glioma TrkB expression, we treated mice bearing TrkB WT and KO glioma xenografts with the Trk inhibitor entrectinib. While entrectinib decreased the proliferation rate of xenografted *NTRK2* WT DIPG cell *in vivo*, it did not decrease the proliferation rate of *NTRK2* KO glioma xenograft (Extended Data Fig 2e-f), demonstrating that the mechanism of action of entrectinib in DIPG is mediated through glioma cell TrkB.

Extended Data Fig 2e-f. **e**, Experimental model of pontine xenografted WT and *NTRK2* KO glioma (SU-DIPGVI) treated with the Pan-Trk inhibitor, entrectinib, or vehicle control. **f**, Proliferation index of wild-type and *NTRK2* KO SU-DIPGVI glioma xenografted to the pons of NSG mice and treated with vehicle or entrectinib (120mg/kg P.O.). Quantification by confocal microscopy analysis of EdU⁺/HNA⁺ co-positive tumor cells, *n* = 4 wild-type glioma xenografted, vehicle-treated mice, 5 wild-type glioma xenografted,

entrectinib-treated mice, 5 *NTRK2* KO glioma xenografted, vehicle-treated mice, 3 *NTRK2* KO glioma xenografted, entrectinib-treated mice).

6. We have now included the expression levels of the other Trk receptors, TrkA and TrkC, in glioma (Extended Data Figure 1a). We have also tested the proliferative response of glioma to activation of the other neurotrophin receptors, TrkA and TrkC (Extended Data Figure 1f).

Extended Data Fig 1a. Primary human biopsy single cell transcriptomic data illustrating the expression of the neurotrophin family genes in H3K27M⁺ DMG (red; *n* = 2,259 cells, 6 study participants), tumor associated, non-malignant immune cells (blue; *n* = 96 cells, 5 participants) and oligodendrocytes (green; *n* = 232 cells).

7. We expanded our single cell analyses to explore the relationship of TrkB expression with other genes in each molecularly defined subpopulation of tumor cells within DMGs (Extended Data Figure 4). This revealed that TrkB signaling may be playing distinct roles in different cellular subpopulations, potentially contributing not only to synaptic biology in neural precursor cell/oligodendroglial-like cells, but also to TM-related processes in the more astrocyte-like cellular subpopulation (Extended Data Figure 4).

Extended Data Fig. 4

Extended Data Fig. 4

a, Analysis of previously published H3K27M⁺ DMG single-cell RNASeq data³⁹ quantifying the percentage of tumor cells in which *NTRK2* (TrkB) was captured in either the astrocyte-like (AC), oligodendrocyte-like (OC) and oligodendroglial precursor cell-like (OPC) glioma cells. **b**, *NTRK2* expression level in malignant H3K27M⁺ malignant single cells projected on the glial-like cell lineage (x axis) and stemness (stem to differentiated; y axis) scores. *NTRK2* expression level was smoothed (for the purpose of data visualization only) for each cell by assigning each cell with the average *NTRK2* expression of its nearest neighbors in the Lineage vs. Stemness 2-dimensional space. **c**, Difference between the scores of the synaptic (SYN) and tumor microtubule (TM) gene signatures (i.e. SYN – TM) in H3K27M⁺ malignant single cells projected on the lineage (x axis) and stemness (stem to differentiated; y axis) scores. **d**, Heatmap of genes correlating with *NTRK2* expression in distinct cellular subgroups (astrocyte-like, oligodendroglial precursor cell-like, oligodendrocyte-like) of H3K27M⁺ malignant single cells. Genes were ordered according to the AC-OC score difference. **e-g**, Gene Ontology (GO) enrichment analysis for the top genes correlated with *NTRK2* expression in distinct cellular subgroups within H3K27M⁺ diffuse midline glioma tumors (145, 138, 97 genes with Pearson's correlation coefficient greater than 0.25 for the AC-like, OC-like and OPC-like malignant cell states respectively) (**e**, astrocyte-like; **f**, oligodendroglial precursor cell-like; **g**, oligodendrocyte-like). Right, tables depicting the genes associated with the biological processes identified (GO terms) for each cellular subgroup.

8. In order to address the heterogeneity of glioma cell electrophysiological responses and the relationship to *NTRK2* expression, we have performed PatchSeq analysis of glioma

cells xenografted to the hippocampus after electrical stimulation of axonal afferents to assess the gene expression of cells with differing electrophysiological phenotypes. Patch-seq is very technically challenging, and obtaining single cell sequencing and high-quality transcriptomic data from 128 cells took one year. As shown below, we find evidence of TrkB expression in each electrophysiological subtype. However, as evident in the data below, cell numbers are low (128 cells) and variability is high, so many more cells will need to be Patch-sequenced to draw firm conclusions. A robust Patch-seq database should be an aspiration of the Cancer Neuroscience field. We include these data here for the rebuttal only and have made the Patch-seq data available on GEO (GSE222398) to begin such a data-sharing effort.

Patch-Seq analysis of glioma. Patient-derived glioma cells expressing GFP (SU-DIPG-VI and SU-DIPG-XIII) were xenografted to the CA1 region of the hippocampus and allowed to engraft for 8-10 weeks. Acute slices of xenografted hippocampi were prepared and patch-clamp electrophysiology of glioma cells (green) was performed. No pharmacological inhibitors of ion channels or neurotransmitter receptors were used. Baseline electrophysiological properties and response to electrical stimulation of the axonal afferents into the CA1 region were recorded, revealing cells that exhibit “fast responses” (< 15 ms) consistent with EPSCs, slow responses (>1000 ms) consistent with potassium-evoked currents (Venkatesh et al, 2019 *Nature*), and medium-duration responses (15 ms – 1000 ms), some of which could be consistent with GABAergic synaptic currents in DIPG/DMG glioma cells (Barron et al., 2022 *BioRxiv*). The response type of cells exhibiting no response to electrical stimulation are labeled as “none”. After recording, intracellular contents were collected in the recording pipette, and single cell RNA sequencing was performed according to the protocol in Cadwell et al., 2016 *Nature Biotechnology*. Glioma cells for which both electrophysiological response and successful RNA sequencing were obtained are included here (n = 128).

9. To explore the extent to which the BDNF potentiation of the glutamate response is mediated by neuron-to-glioma synapses, we tested the effects of afferent axonal stimulation with and without BDNF and observed a small but consistent and significant increase in the amplitude of synaptic glioma currents (Figure 2i-k).

Figure 2i-k. **i**, Electrophysiological model of GFP⁺ glioma cells (green) xenografted in mouse hippocampal CA1 region with Schaffer collateral afferent stimulation. **j**, Representative averaged voltage-clamp traces of evoked glioma excitatory postsynaptic current (EPSC) in response to axonal stimulation (black arrow) before (grey) and after (blue) application of BDNF protein (100 ng/ml in ACSF, 30 min). **k**, Quantification of data in **j** ($n = 5$ glioma cells exhibiting EPSCs out of 43 glioma cells patched from 4 mice).

10. Regarding electrophysiology controls, we have added the following analyses:

- We examined the effects of NMDAR inhibition, glutamate transporter inhibition and compared to AMPAR inhibition. As expected (Venkatesh et al., 2019; Venkataramani et al., 2019), we did not observe any effect of NMDAR nor glutamate transporter inhibition on glutamate-evoked currents in glioma cells (Figure 2c-d).

Figure 2c-d. **c**, Representative voltage-clamp traces of whole cell patch-clamp electrophysiological recordings in glioma cells. Hippocampal slices were perfused with ACSF containing tetrodotoxin (TTX, 0.5 μ M), and response to a local puff (250msec) application of 1 mM glutamate (black square) was recorded from xenografted glioma cells with sequential application of NMDAR blocker (AP-5, 100 μ M), TBOA (200 μ M), AMPAR blocker (NBQX, 10 μ M). **d**, Quantification of data in **c** ($n = 7$ glioma cells, 4 mice).

- controlling for the effects of other mitogenic paracrine factors, we tested for effects of NLGN3 perfusion on glutamate-evoked currents and found no effect (Figure 2g-h)

Figure 2g-h. **g**, Representative traces of glutamate-evoked inward currents (black square) in patient-derived glioma xenografted cells before (grey) and after 30-minute perfusion with NLGN3 recombinant protein (100 ng/ml) in ACSF (containing TTX, 0.5 μ M) (purple). **h**, Quantification of data in **g** ($n = 5$ glioma cells, 3 mice).

- controlling for a possible nonspecific effect of a fluid puff on glioma currents, we tested the effects of aCSF puff and found no glioma current induced by aCSF puff (Extended Data Fig 5a)

Extended Data Fig 5a. **a**, Proliferation index of SU-DIPG-VI WT and *NTRK2* KO glioma monoculture (left), or glioma co-culture with neurons (right) in the presence and absence of the AMPAR blocker NBQX (10 μ M) (quantified as fraction of EdU⁺/HNA⁺ co-positive tumor cells assessed by confocal microscopy, $n = 3$ coverslips/group for glioma monoculture experiments and 6 coverslips/group for neuron-glioma co-culture; experiment also replicated in Figure 1j).

11. We repeated calcium imaging in response to BDNF in a second patient-derived model and on a two-photon microscope for higher resolution of calcium transients, and we have added additional analyses of the calcium imaging data as requested. Evaluation of the second model (Extended Data Figure 5e-h) replicated the results in the first patient-derived model (Fig 2l-o). For both models, we have also added quantification of duration of response demonstrating that both peak and duration of calcium transients are increased after glutamate puff in the presence of BDNF. No change in basal GCaMP6s fluorescence was observed with addition of BDNF in the absence of glutamate puff (Extended Data 5e). With respect to heterogeneity, violin plots of duration of calcium transients with and without BDNF illustrate some degree of heterogeneity between cells with respect to magnitude of increase (Figure 2o and Ext. Data Fig. 5h). Heterogeneity in calcium transients is also evident between patient-derived models, with more tumor cells exhibiting calcium transients after glutamate puff in SU-DIPG-VI compared to SU-DIPG-XIII (Extended Data Figure 5d), consistent with relative proportion of cells in each model exhibiting synaptic responses in each model (Venkatesh et al., 2019).

Extended Data Fig 5e-h. **e**, Baseline GCaMP6s intensity in SU-DIPG-VI glioma cells before and 30-sec after BDNF exposure, in the absence of glutamate puff ($n = 7$ cells, 3 mice). **f**, GCaMP6s intensity trace of SU-DIPG-VI glioma cells response to glutamate puff before (3 cells, 3 mice : light grey, average: dark grey) and after BDNF perfusion (three cells: light blue, average intensity: dark blue). **g**, SU-DIPG-VI GCaMP6s cell response to glutamate puff at baseline and after BDNF perfusion (100 ng/ml, 30 min, $n = 7$ cells, 4 mice). **h**, Duration of calcium transient response to glutamate puff in SU-DIPG-VI hippocampal xenografted cells, before and after perfusion with BDNF (100 ng/ml, 30 min, $n = 6$ cells, 4 mice).

Figure 2o. **o**, Duration of calcium transient in response to glutamate puff in SU-DIPGXIIIIFL GCaMP6s cells before and after BDNF exposure ($n = 9$ cells, 3 mice).

Extended Data Figure 5d. **d**, Quantification of number of xenografted SU-DIPG-XIIIIFL or SU-DIPG-VI cells glioma cells demonstrating a calcium transient (as depicted in Figure 2i) in response to a glutamate puff (responders, grey, non-responders, white).

12. With respect to AMPAR subunit trafficking to the membrane, we have now completed a time course of GluA4 levels at the membrane, which shows a peak in GluA4 at 15 minutes after BDNF exposure (Figure 3b-c). This is consistent with the time course of GluA2 trafficking to the membrane, which also peaks at 15 minutes (Fig. 3h-i). We also demonstrated that BDNF induces GluA3 subunit levels at the membrane (Fig. 3d-e). Please note that GluA1 is quite lowly expressed in gliomas, whereas GluA2, 3 and 4 are highly expressed (Venkatesh et al., 2019).

Figure 3b-c. **b**, Western blot analysis of cell membrane surface and total cell protein levels of the AMPAR subunit, GluA4, collected from SU-DIPG-VI cell cultures treated with 100 nM BDNF for 5 min, 10 min and 30 min. Surface proteins were labelled by biotinylation and extracted from total protein using avidin conjugation. **c**, Quantification of data in **b** (% of biotinylated cell surface GluA4 from control average, $n = 3$ independent biological replicates).

Figure 3d-e. **d**, Western blot analysis of cell membrane surface and total cell protein levels of the AMPAR subunit, GluA3, collected from SU-DIPG-VI cells treated with 100nM BDNF for 30 min. Surface proteins were labelled by biotinylation and extracted from total protein using avidin conjugation. **e**, Quantification of data in **d**, (% of biotinylated cell surface GluA3 from control average, $n = 3$ independent biological replicates).

13. As a control for other neuron-glioma paracrine factors, we showed that NLGN3 does not affect AMPAR subunit (GluA4) levels at the membrane (Figure 3f-g).

Figure 3 f-g. **f**, Western blot analysis of cell membrane surface and total cell protein levels of the AMPAR subunit, GluA4, collected from SU-DIPGVI cells treated with 100nM neuroigin-3 (NLGN3 or NL3) for 30 min. Surface proteins were labelled by biotinylation and extracted from total protein using avidin

conjugation. **g**, Quantification of data in **f**, (% of biotinylated cell surface GluA4 from control average, $n = 3$ independent biological replicates).

14. We replicated the synaptic structure quantification by electron microscopy in an additional cohort of 7 mice (3 WT, 4 *NTRK2* KO glioma xenografts) and provide improved quality EM images. (Figure 4d-e)

Figure 4d-e. **d**, Immunoelectron microscopy of patient-derived GFP+ DIPG cells (SU-DIPG-VI wild-type (left) and *NTRK2* KO (right)) xenografted into the mouse hippocampus. Arrowheads denote immuno-gold particle labelling of GFP, identifying tumor cells. Neuron-to-glioma synaptic structures defined as postsynaptic density in GFP+ glioma cells (colored green), a synaptic cleft, and clustered synaptic vesicles in opposing presynaptic neuron (colored magenta). Scale bar = 2 μ m. **e**, Quantification of identified synapses in **c** for mice harboring wild-type and *NTRK2* KO tumors ($n = 6$ wild-type mice and 7 *NTRK2* KO mice).

15. We replicated synaptic puncta quantification with neuron-glioma co-culture with *NTRK2* KO or WT PSD95-RFP expressing glioma cells (Figure 4f-g), which confirmed the original results using *NTRK2* KD cells (shown in Extended Data 9b-d).

Figure 4f-g. **f**, Confocal images of neurons co-cultured with PSD95-RFP-labelled wild-type and *NTRK2* KO glioma cells. The images depict glioma processes (blue) and neuronal processes (white) in close approximation in co-culture. White denotes neurofilament (axon); blue denotes nestin staining (glioma cell process); green denotes synapsin (presynaptic puncta), red denotes PSD95-RFP (glioma postsynaptic puncta). Scale bar on left = 4 μ m; Scale bar on right = 1 μ m. **g**, Quantification of the colocalization of postsynaptic glioma-derived PSD95-RFP with neuronal presynaptic synapsin in co-cultures of wild-type ($n = 19$ cells, 12 coverslips from 3 independent experiments), or *NTRK2* KO glioma cells (SU-DIPG-VI, $n = 17$ cells, 12 coverslips from 3 independent experiments).

Additional textual changes, controls, images, and analyses are provided as suggested and detailed in the point-by-point responses below.

I should add that we have also received informal advice from a potential referee who wasn't able to review the full paper (but had seen in on bioarxiv). They said that the study doesn't seem to show synapses, only glutamate-evoked responses, which are notoriously unstable, and the amplitude of responses from glutamate (released from a pipet) typically varies a lot. They suggested that what's needed is to show that the amplitude is stable for a baseline period (many responses over 10-20 min) and then stable responses at the enhance level (many responses over 20-30 minutes), if a comparison to LTP is to be drawn. They mentioned that on hippocampal slices the majority of such a glutamate response is from extra-synaptic receptors, and that the responses are likely only very little due to synaptic receptors.

This is an excellent point. We had initially investigated the overall glutamatergic response of glioma cells, and in revision we have now directly address the synaptic component of malignant plasticity by performing whole cell patch clamp electrophysiological recording of glioma cells (wild-type & *NTRK2* KO, hippocampal xenograft slices) in response to electrical axon afferent stimulation with and without BDNF perfusion. This demonstrates the role that BDNF plays in the synaptic electrophysiological response of glioma cells and its effect on neuron-glioma synaptic strength (Figure 2 i-k):

Figure 2i-k. i, Electrophysiological model of GFP⁺ glioma cells (green) xenografted in mouse hippocampal CA1 region with Schaffer collateral afferent stimulation. j, Representative averaged voltage-clamp traces of evoked glioma excitatory postsynaptic current (EPSC) in response to axonal stimulation (black arrow) before (grey) and after (blue) application of BDNF protein (100 ng/ml in ACSF, 30 min). k, Quantification of data in j ($n = 5$ glioma cells exhibiting EPSCs out of 43 glioma cells patched from 4 mice).

Referees' comments:

Referee #1 (Remarks to the Author):

In this very well-written manuscript, the authors build on the previous work with the discovery of neuron-glioma synapses (2019) and describe mechanisms contributing to the plasticity of these synapses, involving a BDNF-NTRK2 signaling axis – a mechanism that appears likely to drive malignancy in glioma. The general concept presented here is

exciting and is an additional piece in the puzzle of how brain tumor cells use, or hijack, neuronal and neurodevelopmental mechanisms to thrive, and how integration into neuronal circuits is effectively achieved. The experiments are carefully performed and well presented, and adequate preclinical and clinical models are selected to answer the relevant questions, with really interesting methodological approaches (e.g., live imaging of synaptic plasticity). The complexity of the system – paracrine effects, synaptic effects, tumor cell autonomous network effects, etc.; at least the first two even in parallel for the same molecular pathway investigated here – is certainly a fundamental challenge that is not easy to resolve when it comes to proving the exact role of a distinct mechanism. Along this line, I recommend to consider the following points that will help to further strengthen the main message and impact of this study. I see three major points that are particularly important:

We thank the Referee for these positive comments.

1.) It needs to be shown how the heterogeneity of glioblastoma cells comes into play and especially which subpopulations are electrophysiologically characterised with regard to glioma cells showing the previously described two types of inward currents. Previously, the group reported that there are two types of inward currents (EPSCs, and slow inward currents). The currents shown in panel 1e, 3a etc. are kinetically more similar to slow inward currents which the authors concluded to be driven by potassium currents (Venkatesh et al., Nature 2019). The currents that are now seen are driven by glutamate puffing (which has not been performed in Venkatesh et al. 2019) and could be in principle be driven by AMPA receptors, NMDA receptors or glutamate transporters, if they are directly mediated by the puff. First, what is the latency to response after glutamate puffing? In other words, is it a direct response to glutamate or is it possibly an indirect effect? The kinetic of the response needs to be explored pharmacologically with experimental paradigms where AMPAR inhibitors, NMDA inhibitors and glutamate transporter inhibitors such as TFB-TBOA are applied and washed out. In which relation do the glutamate-evoked inward currents stand with the previously electrically evoked slow inward currents and EPSCs? Is there a subpopulation that is modulated by BDNF and one that is not modulated by BDNF? Previously, it has been reported that glioma cells express mechanoreceptors. Puffing with ACSF has to be shown as control to exclude the possibility that a subpopulation of the currents is mechanically provoked. Please also report how the glutamate was exactly applied (was a Picospritzer used? how many PSI were applied?).

In our glutamate puff paradigm, TTX was included in the aCSF perfusion, therefore there is no indirect effect from neuronal activity; we have now made this point clearly in the text.

It is important to note that the kinetics of a glutamate puff will be different to that of a neuronal stim-evoked response, as the puff will activate both synaptic and extra-synaptic receptors.

As described above, we have now evaluated the electrophysiological response to axonal afferent electrical stimulation in the glioma cells, with and without BDNF perfusion (Figure 2i-k), as in the Venkatesh et al., 2019 publication.

Figure 2i-k. **i**, Electrophysiological model of GFP⁺ glioma cells (green) xenografted in mouse hippocampal CA1 region with Schaffer collateral afferent stimulation. **j**, Representative averaged voltage-clamp traces of evoked glioma excitatory postsynaptic current (EPSC) in response to axonal stimulation (black arrow) before (grey) and after (blue) application of BDNF protein (100 ng/ml in ACSF, 30 min). **k**, Quantification of data in **j** ($n = 5$ glioma cells exhibiting EPSCs out of 43 glioma cells patched from 4 mice).

We have now characterized the electrophysiological response to glutamate using AMPAR inhibitors, NMDAR inhibitors and glutamate-transporter inhibitors as recommended (see below). As expected, NBQX inhibits the response to glutamate, while NMDAR inhibition (AP-5) and glutamate-transporter inhibition (TBOA) have no effect (Figure 2c-d).

Figure 2c-d. **c**, Representative voltage-clamp traces of whole cell patch-clamp electrophysiological recordings in glioma cells. Hippocampal slices were perfused with ACSF containing tetrodotoxin (TTX, 0.5 μ M), and response to a local puff (250msec) application of 1 mM glutamate (black square) was recorded from xenografted glioma cells with sequential application of NMDAR blocker (AP-5, 100 μ M), TBOA (200 μ M), AMPAR blocker (NBQX, 10 μ M). **d**, Quantification of data in **c** ($n = 7$ glioma cells, 4 mice).

In addition, we have puffed ACSF to test if there is an effect on mechanoreceptors and no effect was seen.

Extended Data Fig 5b. b, Whole cell patch-clamp electrophysiological recording of glioma cell with ACSF puff, representative voltage clamp trace.

In order to address the heterogeneity of glioma cell electrophysiological responses and the relationship to *NTRK2* expression, we have performed PatchSeq analysis of glioma cells xenografted to the hippocampus after electrical stimulation of axonal afferents to assess the gene expression of cells with differing electrophysiological phenotypes. Patch-seq is very technically challenging, and obtaining single cell sequencing and high-quality transcriptomic data from 128 cells took one year. As shown below, we find evidence of TrkB expression in each electrophysiological subtype. However, as evident in the data below, cell numbers are low (128 cells) and variability is high, so many more cells will need to be Patch-sequenced to draw firm conclusions. A robust Patch-seq database should be an aspiration of the Cancer Neuroscience field. We include these data here for the rebuttal only and have made the Patch-seq data available on GEO (GSE222398) to begin such a data-sharing effort.

Patch-Seq analysis of glioma. Patient-derived glioma cells expressing GFP (SU-DIPG-VI and SU-DIPG-XIII) were xenografted to the CA1 region of the hippocampus and allowed to engraft for 8-10 weeks. Acute slices of xenografted hippocampi were prepared and patch-clamp electrophysiology of glioma cells (green) was performed. No pharmacological inhibitors of ion channels or neurotransmitter receptors were used. Baseline electrophysiological properties and response to electrical stimulation of the axonal afferents into the CA1 region were recorded, revealing cells that exhibit “fast responses” (< 15 ms) consistent with EPSCs, slow responses (>1000 ms) consistent with potassium-evoked currents (Venkatesh et al, 2019 *Nature*), and medium-duration responses (15 ms – 1000 ms), some of which could be consistent with GABAergic synaptic currents in DIPG/DMG glioma cells (Barron et al., 2022 *BioRxiv*). The response type of cells exhibiting no response to electrical stimulation are labeled as “none”. After recording, intracellular contents were collected in the recording pipette, and single cell RNA sequencing was performed according to the protocol in Cadwell et al., 2016 *Nature Biotechnology*. Glioma cells for which both electrophysiological response and successful RNA sequencing were obtained are included here (n = 128).

The details of the picospritzer (~10 psi) are now included in the methods.

2.) It is well understood that it is challenging to decipher the specific pathobiological role of BDNF-NTRK signaling on synaptic plasticity of the neuron-glioma cell synapse vs. paracrine and other effects of this pathway. The authors try hard to do that in Fig. 4, which

however requires inter-experiment comparisons (most notably, 4b vs 4c, and in vitro vs in vivo experiments). To further strengthen the concept of a specific biological role of malignant synaptic plasticity for glioma biology, the authors should aim to provide more data using other regulators (stimulation paradigms, stimulators, inhibitors) of general synaptic plasticity. Moreover, they might want to consider to use their elegant mouse model deficient in activity-induced BDNF expression/secretion more, e.g. by growing a control vs NTRK2 KO glioma line. Here, an additive effect of the KO on survival should be missing. This does not exclude paracrine effects but can help to better control for unrelated effects occurring in this system.

This is an excellent point, and we have now optogenetically stimulated (glutamatergic) cortical projection neurons in awake, behaving BDNF-TMKI mice (that lack activity-regulated BDNF expression and secretion) bearing glioma xenografts to demonstrate the role of BDNF in the effect of neuronal activity on tumor growth. We find that the proliferative effects of cortical projection neuronal activity on two patient-derived glioma models are decreased in the mouse model deficient in activity-induced BDNF expression/secretion (Figure 1a-d and Ext Data 1c-d).

Figure 1a-d. **a**, Schematic of Bdnf-TMKI mouse, which lacks activity-regulated *BDNF* expression. **b**, Optogenetic model for optogenetic stimulation of ChR2-expressing neurons (blue) in microenvironment of glioma xenograft (green); light blue rectangle denotes region of analysis. P, postnatal day. **c**, Representative image of glioma cells (SU-DIPG-VI) xenografted into wild-type and Bdnf-TMKI NOD-SCID-gamma (NSG) mice in the presence of optogenetically stimulated neurons quantified in **d**, gray denotes human nuclear antigen (HNA)-positive glioma cells; red denotes Ki67 (proliferative marker). Scale bar = 50 μ m. **d**, Proliferation index of SU-DIPG-VI glioma xenografted to mice with Thy1+ glutamatergic cortical projection neurons lacking (ChR2-) or expressing Channelrhodopsin (ChR2+) in a wild-type or Bdnf-TMKI genetic background (quantified by Ki67⁺/HNA, $n = 6$ wild-type ChR2- mice, 4 Bdnf-TMKI ChR2- mice, 7 wild-type ChR2+ mice, 4 Bdnf-TMKI ChR2+ mice).

Extended Data Fig 1c-d. c Model for optogenetic stimulation of ChR2-expressing neurons (blue) in microenvironment of glioma xenograft (green); light blue rectangle denotes region of analysis. P, postnatal day. **d**, Proliferation index of SU-DIPGXIIIIFL glioma xenografted to mice with neurons expressing Channelrhodopsin (ChR2+) in a wild-type or Bdnf-TMKI genetic background (Figure 1a) after neuronal optogenetic stimulation (quantified by confocal microscopy of EdU+/HNA cells, $n = 7$ wild-type ChR2+ mice, 8 Bdnf-TMKI ChR2+ mice).

As suggested, we have now measured glioma proliferation of *NTRK2* KO glioma xenografts compared to wild-type cells in the BDNF-TMKI mice (compared to WT mice) to control for unrelated effects (Extended Data Figure 1k).

Extended Data Fig 1k. Proliferation index of *NTRK2* KO SU-DIPG-VI glioma xenografted to mice with neurons expressing Channelrhodopsin (ChR2+) in a wild-type or Bdnf-TMKI genetic background after neuronal optogenetic stimulation (quantified by confocal microscopy of EdU+/HNA cells, $n = 5$ wild-type ChR2+ mice, $n = 4$ BDNF-TMKI ChR2+ mice).

3.) Another important question is whether malignant synaptic plasticity is exclusively regulated by the BDNF-NTRK axis, or whether other plausible candidates, most notably NLGN3, have also an effect. This would be an important piece of information and would help to better develop a broader picture of malignant synaptic plasticity in brain tumor pathophysiology.

This is an important question, and it is likely that BDNF-TrkB signaling is only the first pathway identified that can regulate malignant adaptive plasticity, and there may be others. With respect to NLGN3, we have now tested NLGN3 effects on glutamate-evoked currents and find that it does not acutely modulate the electrophysiological response in glioma cells. To examine the possible role of NLGN3 in glioma cell plasticity, we perfused NLGN3 in our electrophysiological experimental paradigm and compared the effect of NLGN3 on glioma current amplitude to that of BDNF (Figure 2g-h). We also examined the ability of NLGN3 to regulate AMPAR trafficking using cell surface biotinylation protein assays (Figure 3f-g). In both cases, we found no effect of NLGN3.

Figure 2g-h. **g**, Representative traces of glutamate-evoked inward currents (black square) in patient-derived glioma xenografted cells before (grey) and after 30-minute perfusion with NLGN3 recombinant protein (100 ng/ml) in ACSF (containing TTX, 0.5 μ M) (purple). **h**, Quantification of data in **g** ($n = 5$ glioma cells, 3 mice).

Figure 3-f-g. **f**, Western blot analysis of cell membrane surface and total cell protein levels of the AMPAR subunit, GluA4, collected from SU-DIPGVI cells treated with 100nM neuroigin-3 (NLGN3 or NL3) for 30 min. Surface proteins were labelled by biotinylation and extracted from total protein using avidin conjugation. **g**, Quantification of data in **f**, (% of biotinylated cell surface GluA4 from control average, $n = 3$ independent biological replicates).

Connecting NLGN3 and BDNF, however, are our past observations that NLGN3 promotes synaptic connectivity: the loss of NLGN3 reduces the number of neuron-glioma synaptic connections *in vitro* (Venkatesh 2019) and NLGN3 up-regulates TrkB and AMPAR subunit gene expression in glioma (Venkatesh et al., 2017). We have now discussed these important points in the revised manuscript, stating:

The findings here illustrate that neuronal activity-regulated factors not only directly promote glioma growth^{1,3,35,51}, but can also further reinforce neuron-glioma interactions. Two key activity-regulated paracrine factors, NLGN3 and BDNF, each promote neuron-to-glioma synaptic interactions in distinct ways: NLGN3 promotes expression of genes encoding AMPAR subunits (GluA2 and GluA4) as well as NTRK2 (TrkB)², while BDNF-TrkB signaling promotes trafficking of translated AMPAR subunits to the postsynaptic membrane to modulate synaptic strength. Both NLGN3⁴ and BDNF promote neuron-to-glioma synapse formation.

Other major points:

4.) The top genes associated with NTRK2 expression in pediatric gliomas (Ext Data Fig. 1f) are very interesting. No. 1 is GJA1, which encodes Cx43 – which is THE gap junction protein that is responsible for glioma cell network connectivity/integration, which in turn was found for those glioma cells displaying slow inward currents in the 2019 study of the authors. That might explain the exclusive slow inward current (-like) patch clamp recordings shown in this study; and stresses the importance of understanding the heterogeneous electrical responses better, including its tumor-biological fundamentals. – In addition, and remarkably, No 3 is Ttyh1, which has been described as a key molecular driver for INVASIVE (not interconnecting) tumor microtubes in glioma – which implies the question how synaptic plasticity is associated with tumor cell invasiveness. Looking at

this data in synopsis, it is quite likely that both genes play important (co-) roles for NTRK2-high glioma cells, but in distinct subpopulations of glioma cells. 1. Is that indeed the case (should be answered easily with this single cell datasets?).

We agree with the Referee's comments and in response we have now delved deeper into single cell RNA sequencing datasets of glioma, in collaboration with Mario Suva and Itay Tirosh, to better understand the role of *NTRK2* in distinct glioma subpopulations. This revealed that TrkB signaling may be playing distinct roles in different cellular subpopulations, potentially contributing not only to synaptic biology in neural precursor cell/oligodendroglial-like cells, but also to TM-related processes in the more astrocyte-like cellular subpopulation (Extended Data Figure 4). The new text added to the results section now reads:

To explore possible roles for BDNF-TrkB signaling in glioma pathophysiology, we examined gene expression relationships between TrkB and other gene programs at the single cell level using available human H3K27M-mutated diffuse midline glioma primary biopsy tissue single-cell transcriptomic data⁴⁰. NTRK2 is expressed in the majority of glioma cells at varying levels across the defined cellular subpopulations that comprise diffuse midline gliomas, including oligodendrocyte precursor cell-like tumor cells (OPC), astrocyte-like tumor cells (AC) and oligodendrocyte-like tumor cells (OC; Extended Data Fig. 4a, b). As previously demonstrated⁴, synaptic gene expression is enriched in the oligodendroglial compartments of the tumor (oligodendrocyte-like and oligodendrocyte precursor cell-like cellular subpopulations), while tumor microtubule-associated gene expression is enriched in the astrocyte-like compartment (Extended Data Figure 4c). Expression correlation analyses identified different patterns of genes in each cellular compartment that correlate with NTRK2 expression (Extended Data Fig 4c). Examples of genes that are strongly correlated with NTRK2 in the astrocyte-like compartment include: GJA1, TTHY1, GRIK1 and KCNN3; TTHY1 and GJA1 are known to play crucial roles in tumor microtubule formation and connectivity in adult astrocytomas^{41,42}. In the OC-like compartment, NTRK2 expression correlates with: NRXN2, NLGN3, CSPG4, PDGFRA, FGFR1, CNTN1, SLIT2, IGF1R, and CACNG5, and in the OPC-like compartment correlates with: NRXN2, NRXN1, NLGN4X, SYT11, CREB5, SRGAP2C, CSPG4, ASCL1, PI3KR3, CDK6, EGFR, EPHB1 (Extended Data Figure 4d). Gene Ontology analyses of these differentially correlated genes in each cellular sub-compartment revealed correlation of NTRK2 with processes of synaptic communication and neural circuit assembly (Extended Data 4e-g). In the OPC-like compartment, NTRK2 expression correlated with post-synaptic organization, axon guidance, neuronal projection guidance, neuronal migration, ERK signaling cascades and the AKT signaling cascade, consistent with the hypothesized role of TrkB in neuron-to-glioma synapses, consequent effects of AMPAR-mediated synaptic signaling on tumor migration³⁴ and expected signaling consequences of TrkB activation. In the OC-like compartment, the gene sets correlated with NTRK2 expression involve synaptic organization, modulation of synaptic transmission, synaptic plasticity, and learning and memory. In the astrocyte-like compartment, which tends to engage in extensive tumor microtubule connectivity⁴¹, NTRK2 expression correlated with genes involved in axon guidance and neuronal projection morphogenesis. Taken together, these single cell transcriptomic analyses support potential roles for TrkB signaling in neuron-to-glioma synaptic biology as well as

glioma-to-glioma network formation, with *TrkB* correlated with distinct processes in astrocyte-like and oligodendroglial-like cellular subpopulations.

Extended Data Fig. 4

Extended Data Fig. 4

a, Analysis of previously published H3K27M⁺ DMG single-cell RNASeq data³⁹ quantifying the percentage of tumor cells in which *NTRK2* (*TrkB*) was captured in either the astrocyte-like (AC), oligodendrocyte-like (OC) and oligodendroglial precursor cell-like (OPC) glioma cells. **b**, *NTRK2* expression level in malignant H3K27M⁺ malignant single cells projected on the glial-like cell lineage (x axis) and stemness (stem to differentiated; y axis) scores. *NTRK2* expression level was smoothed (for the purpose of data visualization only) for each cell by assigning each cell with the average *NTRK2* expression of its nearest neighbors in the Lineage vs. Stemness 2-dimensional space. **c**, Difference between the scores of the synaptic (SYN) and tumor microtubule (TM) gene signatures (i.e. SYN – TM) in H3K27M⁺ malignant single cells projected on the lineage (x axis) and stemness (stem to differentiated; y axis) scores. **d**, Heatmap of

genes correlating with *NTRK2* expression in distinct cellular subgroups (astrocyte-like, oligodendroglial precursor cell-like, oligodendrocyte-like) of H3K27M⁺ malignant single cells. Genes were ordered according to the AC-OC score difference. **e-g**, Gene Ontology (GO) enrichment analysis for the top genes correlated with *NTRK2* expression in distinct cellular subgroups within H3K27M⁺ diffuse midline glioma tumors (145, 138, 97 genes with Pearson's correlation coefficient greater than 0.25 for the AC-like, OC-like and OPC-like malignant cell states respectively) (**e**, astrocyte-like; **f**, oligodendroglial precursor cell-like; **g**, oligodendrocyte-like). Right, tables depicting the genes associated with the biological processes identified (GO terms) for each cellular subgroup.

2. The authors might also want to consider to investigate both genes in more detail with respect to the main findings of this manuscript.

We have focused on these genes in the single cell analyses discussed above.

5.) Likewise, the gene expression changes induced by BDNF treatment (Ext Data Fig. 5) show a pattern that is very well fitting to a highly tumor microtubule (network) - proficient one (GAP-43; VGF etc.). The authors might want to discuss this aspect (and modify the presentation in the results section a bit, accordingly).

We have now added a much more extensive result section focused on single cell biology and the role of BDNF-TrkB signaling in different cellular subpopulations.

6.) 1k, Please also report the heterogeneity of calcium signals and what kind of subpopulations can be found with glutamate-evoked calcium currents.

We find heterogeneity between patient-derived models and within a given model. For example, we find more responders in the SU-DIPG-VI model than the SU-DIPG-XIII model (Extended Data Figure 5d), which is concordant with our previous findings that SU-DIPG-VI exhibits more neuron-to-glioma synapses than SU-DIPG-XIII (Venkatesh et al, 2019 Nature).

Extended Data Figure 5d. d, Quantification of number of xenografted SU-DIPG-XIII^{FL} or SU-DIPG-VI glioma cells demonstrating a calcium transient (as depicted in Figure 2I) in response to a glutamate puff (responders, grey, non-responders, white).

We have also examined our previous and new data and show the distribution of calcium transient changes in response to BDNF, which we now demonstrate reflects not only increased calcium peak intensity but also calcium transient duration (Figure 2l-o and Extended Data Figure 5 c-f). We also find heterogeneity within patient-derived tumor models as evident in the variable magnitude of increased duration of calcium transients in response to BDNF (Figure 2o and Extended Data Figure 5f-h).

Figure 2 l-o

Figure 2l-o. **l**, Two-photon in-situ imaging illustrating a time series over 8 sec of calcium transients evoked by local glutamate puff (1 mM, 250 msec) in a representative glioma cell before (top) and after (bottom) perfusion with BDNF (100 ng/ml in ACSF, 30 min). Green denotes glioma GCaMP6s fluorescence and red denotes tdTomato nuclear tag. Scale bar = 10 μ m. **m**, GCaMP6s intensity trace of SU-DIPG-XIIIIFL glioma cells response to glutamate puff at baseline (4 cells, 2 mice: light grey, average intensity: dark grey) and after BDNF perfusion (4 cells: light blue, average intensity: dark blue). **n**, SU-DIPG-XIIIIFL GCaMP6s cell response to glutamate puff at baseline and after BDNF exposure ($n = 9$ cells, 3 mice). **o**, Duration of calcium transient in response to glutamate puff in SU-DIPGXIIIIFL GCaMP6s cells before and after BDNF exposure ($n = 9$ cells, 3 mice).

Extended Data Figure 5f-h. **f**, GCaMP6s intensity trace of SU-DIPG-VI glioma cells response to glutamate puff before (3 cells, 3 mice : light grey, average: dark grey) and after BDNF perfusion (three cells: light blue, average intensity: dark blue). **g**, SU-DIPG-VI GCaMP6s cell response to glutamate puff at baseline and after BDNF perfusion (100 ng/ml, 30 min, $n = 7$ cells, 4 mice). **h**, Duration of calcium transient response to glutamate puff in SU-DIPG-VI hippocampal xenografted cells, before and after perfusion with BDNF (100 ng/ml, 30 min, $n = 6$ cells, 4 mice).

7.) Only the AMPAR subunit GluA4 was investigated after BDNF treatment. What about the other subunits? It would be most convincing if all (or at least one more) subunit(s) would also increasingly locate to the cell membrane.

Glioma cells express GluA2, 3 and 4 subunits (Venkatesh et al., 2019 Nature). We have now examined changes in GluA2, GluA3 and GluA4 at the membrane after BDNF exposure (Figure 3). The GluA2 calcium-permeable subunit was examined using the pHluorin live imaging of subunit trafficking. We have now examined GluA3 using cell surface biotinylation and find that similarly to GluA4, BDNF increased GluA3 expression at the cell surface (Figure 3d-e):

Figure 3d-e. **d**, Western blot analysis of cell membrane surface and total cell protein levels of the AMPAR subunit, GluA3, collected from SU-DIPG-VI cells treated with 100nM BDNF for 30 min. Surface proteins were labelled by biotinylation and extracted from total protein using avidin conjugation. **e**, Quantification of data in **d**, (% of biotinylated cell surface GluA3 from control average, $n = 3$ independent biological replicates).

8.) The live imaging system of AMPAR tracking (Fig. 2d) is indeed a very interesting system. In contrast to the biochemical assays used for analysis of the surface proteins this system can be used to explore heterogeneity on multiple levels, which can potentially be improved. Furthermore, the relationship between these signals and electrophysiological response should be correlated. Furthermore, the number of observations needs to be significantly increased to clearly show how this is related to glioma cell heterogeneity 2f. The time course is very interesting. It would be interesting if this could also be functionally shown with at least calcium imaging, ideally also electrophysiology if the patches are stable enough.

This is an important question whether the observed increased AMPAR trafficking correlates with the observed increased glutamate-evoked current. While we can't do electrophysiology or calcium imaging and Phluorin imaging in the same cells, we have now shown that the timing of AMPAR trafficking (by 5 minutes after BDNF exposure,

peaking at 15 min for multiple AMPAR subunits) matches the timing of increased calcium transients observed after BDNF exposure (new Figure 3b-c).

Figure 3b-c. **b**, Western blot analysis of cell membrane surface and total cell protein levels of the AMPAR subunit, GluA4, collected from SU-DIPG-VI cell cultures treated with 100 nM BDNF for 5 min, 10 min and 30 min. Surface proteins were labelled by biotinylation and extracted from total protein using avidin conjugation. **c**, Quantification of data in **b** (% of biotinylated cell surface GluA4 from control average, $n = 3$ independent biological replicates).

9.) 3c, The labelling used for electron microscopy is not completely convincing. Gold particles are clustered, often a sign of unspecific labelling and a lot of gold particles can be seen outside the putative glioma cell postsynaptic side. Although a postsynaptic (?) density can be clearly seen in both examples shown, a vesicle cluster of the presynaptic bouton is difficult to distinguish. What criterions were used to be qualified as glioma synapse? Please show more examples at least in an Extended Figure. Control tissue needs to be used to unequivocally show specific binding of immunogold. It is unclear how the quantification was done and whether this method can be used to truly quantify connectivity. Ideally, large whole-cell reconstructions would need to be performed to comment on connectivity which is clearly beyond the scope of the current study. However, a stereological approach should be used to comment on differences between both groups. In addition, please report the exact number of observations made here (how many processes? how many somata were observed? how do these numbers relate?).

Secondary antibody-only controls were used to establish background levels of non-specific binding in immuno-EM analyses (now shown in Extended Data Figure 9a).

Extended Data Fig 9a. a, Electron microscopy images of glioma xenografted mouse hippocampal tissue sections with immuno-gold particle labeling of GFP. **Left**) secondary only stains to show background levels of non-specific gold particle labeling (black arrows). **Right**) Examples of glioma processes and additional examples of neuron-to-glioma synapses positive for >4 immuno-gold particles (white arrows) in patient-derived SU-DIPG-VI cells xenografted to the hippocampus. Scale bar = 500 nm (left), all other scale bars = 200 nm.

We have repeated the EM analyses with additional samples (WT n=6, KO n=7 mice) and included more clear EM images with clearer presynaptic vesicles (Figure 4d-e and Extended Data Figure 9a).

Figure 4 d. d, Immuno-electron microscopy of patient-derived GFP+ DIPG cells (SU-DIPG-VI wild-type (left) and *NTRK2* KO (right)) xenografted into the mouse hippocampus. Arrowheads denote immuno-gold particle labelling of GFP, identifying tumor cells. Neuron-to-glioma synaptic structures defined as postsynaptic density in GFP+ glioma cells (colored green), a synaptic cleft, and clustered synaptic vesicles in opposing presynaptic neuron (colored magenta). Scale bar = 2 μ m.

We have also included description of the number of processes and synapses observed in each EM micrograph, now included in the methods:

Overall, 280 sections of SU-DIPG-VI WT across 7 mice and 253 sections of NTRK2 KO across 7 mice were analysed...The number of confirmed glioma-neuron synapses identified was divided by the total number of glioma cells identified to provide the percentage of synaptic structures present. Overall, the analyses identified 0-6 glioma processes per section and 0-2 neuron-glioma synaptic structures per section.

Minor points:

Figure 1a Description: Too elaborate - no long explanations needed.

We have edited this text.

1c: The morphology of the shown cells looks disrupted. HNA signal cannot be clearly detected from the shown image.

We have repeated imaging of the electrophysiological experiment acute slices to obtain better images (Figure 2b).

Figure 2b, Representative image of Alexa 568 (red)- filled GFP+ glioma cell post whole-cell patch clamp recording. Co-labelled with GFP (green) and human nuclear antigen (HNA, grey). Scale bars = 10 μ m.

1e: Please report how many cells were overall patched, how many cells were responsive to glutamate.

We have added total cells patched for both glutamate puff and for axonal stimulation. Overall, for glutamate puff there were 25 WT glioma cells patched, with 23 responders and 15 NTRK2 KO glioma cells patched with 8 responders. For axonal stimulation, there were 43 cells patched with 5 exhibiting EPSC, consistent with the ~10% of glioma cells in a given tumor that exhibit neuron-to-glioma synapses (Venaktesh et al, 2019 *Nature*; Venkataramani et al, 2019 *Nature*)

1i, Please show time series as visualization. It would be better to use an example where the exact cell morphology can be deduced from. In this example either the SNR was not good or the cells were out of focus.

We have now included a timeseries of calcium transient images in response to glutamate puff, Figure 2I.

Figure 2I. I, Two-photon in-situ imaging illustrating a time series over 8 sec of calcium transients evoked by local glutamate puff (1 mM, 250 msec) in a representative glioma cell before (top) and after (bottom) perfusion with BDNF (100 ng/ml in ACSF, 30 min). Green denotes glioma GCaMP6s fluorescence and red denotes tdTomato nuclear tag. Scale bar = 10 μ m.

3e, Please comment on the exact methods of analysis of these imaging data. How was colocalization quantified? What was counted as puncta? Why can only one synapsin cluster be detected in this field of view? The overall synaptic density seems rather low in this model system. How does this relate to xenograft or even human tissue?

We have now described more detailed puncta imaging analyses methods.

We have replicated the experiment with *NTRK2* KO glioma cells (Figure 4g, in addition to the previously included *NTRK2* KD glioma cell experiment, Extended Data Figure 9d) and have now included new images of the neuron co-culture staining (Figure 4f). The images are high magnification to ensure reliable puncta identification. The overall synaptic density in co-culture is consistent with previously reported results in our Venkatesh et al., 2019 Nature paper. Synaptogenesis in co-culture is not comparable to the *in vivo* microenvironment. As we now note in the revised manuscript and described previously past work from our lab and from others (Venkatesh et al., 2019 Nature; Venkataramani et al., 2019), approximately 10% of glioma cells within a given tumor exhibit synaptic structures *in vivo* and exhibit EPSCs in response to axonal stimulation in acute slices.

Line 206: should read 4b,c

We will correct this in the text.

The finding that NTRK inhibitors are effective in non-NTRK-fusion gliomas (partially or completely by modulating malignant synaptic plasticity) is a very interesting and potentially clinically relevant one. The authors could discuss these implications in the results or discussion section.

We share the Referee's enthusiasm for the therapeutic implications of these findings, and we have included this very important point in the text:

"These findings provide a rationale to expand the potential therapeutic use of TRK inhibitors, already showing clinical promise in NTRK-fusion malignancies³⁵, to also include non-NTRK-fusion gliomas."

We are grateful for the Referee's helpful comments and suggestions, which have greatly improved the manuscript.

Referee #2 (Remarks to the Author):

The manuscript titled "Glioma synapses recruit mechanisms of adaptive plasticity" by Taylor et al. from the laboratory of Dr Michelle Monje builds on the exciting line of investigation from this laboratory over the last few years that has begun to elucidate mechanism relating glioma/neuron interactions. The current study probes BDNF/trkB and neuron-glioma signaling. Trk is highly expressed in many pediatric gliomas. The findings demonstrating that knocking down (they reduce expression by 80% or so) or inhibiting trk signaling increases survival in models are interesting and potentially important.

However, the authors don't clearly demonstrate that the adaptive plasticity mechanisms referred to in the title impact tumor function or survival. They hint the importance of non-growth factor trk signaling in tumor regulation but do not determine what this signaling is, only showing that trk signaling can recruit AMPARs in models, but never closing the loop to show that this recruitment is linked to clinical outcome.

We thank the Referee for these positive comments. In our previous work, we demonstrated that membrane depolarization itself promotes glioma cell proliferation (Venkatesh et al., 2019 *Nature*), a concept that was recently underscored by further work from the Winkler lab (Hausman et al., 2022 *Nature*). We have now made this point more clear in the revised text.

The authors use the drug entrectinib to inhibit trk signaling. Entrectinib inhibits trk function broadly, targeting A, B and C, but the effects on survival are modest, apparently less

effective than *NTRK2* knockdown. The authors should test the selectivity of their drug effects by testing whether there is any effect on survival, or other assays of the drug in combination with *NTRK2* knockdown.

These are excellent points. The period of TrkB inhibition with Entrectinib is shorter than in the context of the genetic knockout, with Entrectinib administered for only two weeks, and the genetic loss affecting the entirety of the experiment. We have clarified in the figure legend that the drug treatment is only a brief (two week) period in an aggressive tumor model.

As suggested, we have tested for off-target effects by assessing the *in vivo* proliferation rate of WT and *NTRK2* KO tumor cells treated with Entrectinib (Ext Data Fig 2e-f).

Extended Data Fig 2e-f. **e**, Experimental model of pontine xenografted WT and *NTRK2* KO glioma (SU-DIPGVI) treated with the Pan-Trk inhibitor, entrectinib, or vehicle control. **f**, Proliferation index of wild-type and *NTRK2* KO SU-DIPGVI glioma xenografted to the pons of NSG mice and treated with vehicle or entrectinib (120mg/kg P.O.). Quantification by confocal microscopy analysis of EdU+/HNA+ co-positive tumor cells, $n = 4$ wild-type glioma xenografted, vehicle-treated mice, 5 wild-type glioma xenografted, entrectinib-treated mice, 5 *NTRK2* KO glioma xenografted, vehicle-treated mice, 3 *NTRK2* KO glioma xenografted, entrectinib-treated mice).

We have now included the expression levels of the other Trk receptors, TrkA and TrkC, in glioma (Extended Data Figure 1a). We have also tested the proliferative response of glioma to activation of the other neurotrophin receptors, TrkA and TrkC (Extended Data Figure 1f).

Extended Data Fig 1a. Primary human biopsy single cell transcriptomic data illustrating the expression of the neurotrophin family genes in H3K27M⁺ DMG (red; $n = 2,259$ cells, 6 study participants), tumor

associated, non-malignant immune cells (blue; $n = 96$ cells, 5 participants) and oligodendrocytes (green; $n = 232$ cells).

The authors propose that *trk* activation has both direct effects on glioma growth and indirect effects via AMPARs. This idea should be tested by examining whether blocking AMPARs together with *trk* signaling further limits glioma growth or other negative impacts. Without a more direct test of the link between *trk* and AMPARs the study is interesting but somewhat descriptive.

This is an excellent suggestion. In response, we have tested whether effects of glioma *TrkB* signaling are related to or independent of AMPA receptor signaling. We found that pharmacologically blocking AMPA receptors, or genetically blocking *TrkB* through *NTRK2* KO, each decreased tumor cell proliferation *in vivo* or in neuron-to glioma co-culture (Figure 1 j-m and Extended Data Fig. 5a). However, we found no additive effect of blocking AMPA receptors and *TrkB* in co-culture or *in vivo*, suggesting that the mechanisms may be related.

Figure 1j-m. **j**, Proliferation index of SU-DIPG-VI WT and *NTRK2* KO glioma co-culture with neurons in the presence and absence of the AMPAR blocker NBQX ($10\mu\text{M}$; $n = 3$ coverslips/group, experiment also repeated in Extended Data Figure 5a). **k**, Experimental model of pontine injected WT and *NTRK2* KO glioma (SU-DIPG-VI) treated with the AMPAR blocker perampanel or vehicle control. **l**, Representative image of xenografted wild-type glioma cells treated with vehicle, and *NTRK2* KO cells treated with the AMPAR blocker perampanel quantified in **m**, gray denotes HNA positive glioma cells; red denotes Ki67. Scale bar = $50\mu\text{m}$. **m**, Proliferation rate of wild-type (WT) and *NTRK2* KO glioma xenografts (SU-DIPG-VI) treated with the AMPAR blocker perampanel or vehicle control (quantified by Ki67⁺/HNA; WT + vehicle; $n = 6$ mice, WT + perampanel; $n = 7$ mice, *NTRK2* KO + vehicle; $n = 5$ mice, *NTRK2* KO + perampanel; $n = 6$ mice).

Extended Data Fig 5a. a, Proliferation index of SU-DIPG-VI WT and *NTRK2* KO glioma monoculture (left), or glioma co-culture with neurons, in the presence and absence of the AMPAR blocker NBQX (10 μ M) (quantified as fraction of EdU⁺/HNA⁺ co-positive tumor cells assessed by confocal microscopy, $n = 3$ coverslips/group for glioma monoculture experiments and 6 coverslips/group for neuron-glioma co-culture; experiment also replicated in Figure 1j).

It is surprising that the authors do not detect GAPDH in their cell surface experiment as numerous reports indicate that this protein is found extracellularly. Indeed, GAPDH localized to the membrane, the nucleus, polysomes, the ER and the Golgi. The data shown in figure 2, where no GAPDH is detected in the putative cell surface fraction raise significant methodological concerns about this experiment. What fraction is this? What is the explanation for this result? These data are not convincing. New experiments and an explanation of how this result was obtained are needed. A better control would be actin.

GAPDH has indeed been described on the cell surface of some cells (for example it can bind to the HCO⁻³/Cl⁻ anion exchanger in the membrane) and of course in the mitochondrial membrane. In many cells/tissues, GAPDH is chiefly nuclear and/or cytoplasmic. To confirm that GAPDH is the correct control for glioma cells, we examined GAPDH immunostaining in the Human Protein Atlas, which shows that localization of GAPDH in glioma samples is nuclear and/or cytoplasmic. Concordantly, in the glioma plasma membrane fractions examined in this study, we do not expect to find substantial GAPDH and lack of plasma membrane GAPDH is what we consistently found upon Western blot analyses. In contrast, actin does insert into the plasma membrane and we worried that actin would be a less ideal control for this reason. We based our use of GAPDH as a control protein for this experiment on a published *Cell* STAR protocol for use of cell surface biotinylation assays to study AMPA receptor subunits in neurons (<https://star-protocols.cell.com/protocols/1216>).

Further supporting the conclusion that AMPAR subunits increased membrane levels after BDNF exposure is the orthogonal approach of the GluA2-Phlourin experiments, which do not rely on Western blotting at all and demonstrate the same increase in membrane trafficking following BDNF exposure.

Figure 3 examines the effects of crispr mediated reduction of Trk expression impacts synapses between neurons and glioma in the authors' in vitro model. The statistical power of these experiments is very low with n's under ten cells. The number of processes examined is not reported for c-d. Remarkably, the authors report effects from only six cells in e-g. This is well below standards. Moreover, the authors fail to show examples of both control and knockdown. It is unclear whether these effects would be consistent in large data sets.

We have now repeated the neuron-glioma co-culture synaptic puncta imaging analyses, quantifying the percent of PSD-95 puncta co-localized with synapsin on each glioma cell in wild-type and *NTRK2* KO glioma, with a higher number of n's and new representative images (Figure 4f-g).

Figure 4f-g. f, Confocal images of neurons co-cultured with PSD95-RFP-labelled wild-type and *NTRK2* KO glioma cells. The images depict glioma processes (blue) and neuronal processes (white) in close approximation in co-culture. White denotes neurofilament (axon); blue denotes nestin staining (glioma cell process); green denotes synapsin (presynaptic puncta), red denotes PSD95-RFP (glioma postsynaptic puncta). Scale bar on left = 4 μ m; Scale bar on right = 1 μ m. g, Quantification of the colocalization of postsynaptic glioma-derived PSD95-RFP with neuronal presynaptic synapsin in co-cultures of wild-type ($n = 19$ cells, 12 coverslips from 3 independent experiments), or *NTRK2* KO glioma cells (SU-DIPG-VI, $n = 17$ cells, 12 coverslips from 3 independent experiments).

Given previous work by these authors linking trk signaling to NLGNs in gliomas is somewhat unexpected that the authors did not examine whether these signaling pathways might intersect here.

This is a great point. We have previously observed that exposure to NLGN3 increases the expression of *NTRK2* in glioma cells (Venkatesh et al., 2017 Nature). To examine the potential role of soluble NLGN3 on the amplitude of the glutamatergic current in glioma cells, we perfused NLGN3 in our xenografted hippocampal slice electrophysiological experimental paradigm and found no effect of NLGN3 on glioma glutamatergic current

amplitude, in contrast to BDNF (Figure 2g,h). This is a nice control, and we thank the Referee for suggesting it.

We also tested for any effects of NLGN3 on GluA4 trafficking and find no effects (Extended Data Figure 3f-g).

Figure 2g-h. **g**, Representative traces of glutamate-evoked inward currents (black square) in patient-derived glioma xenografted cells before (grey) and after 30-minute perfusion with NLGN3 recombinant protein (100 ng/ml) in ACSF (containing TTX, 0.5 μM) (purple). **h**, Quantification of data in **g** ($n = 5$ glioma cells, 3 mice).

Figure 3f-g. **f**, Western blot analysis of cell membrane surface and total cell protein levels of the AMPAR subunit, GluA4, collected from SU-DIPGVI cells treated with 100nM neuroigin-3 (NLGN3 or NL3) for 30 min. Surface proteins were labelled by biotinylation and extracted from total protein using avidin conjugation. **g**, Quantification of data in **f**, (% of biotinylated cell surface GluA4 from control average, $n = 3$ independent biological replicates).

We are grateful for the Referee's helpful comments and suggestions, which have greatly improved the manuscript.

Referee #3 (Remarks to the Author):

The work by Taylor et al. is a follow up of their previous study also published in Nature, which described excitatory synaptic formation of glioma cells. In the new paper, the

authors tried to address the mechanism that regulates synaptic glutamate receptor and found BDNF-TrkB signaling mediates this. Then they went further to test the effect of BDNF on the tumor cell growth with a final aim to develop effective treatment of this intractable disease. So this work has clinical relevance. We appreciate these positive comments.

Said this, I found this work has a major disruption of logical flow. In the first half, the authors made effort to establish how glioma malignant synapse is regulated. Then in the rest of study, they are testing if BDNF signaling is involved in the proliferation of glioma cells. Indeed, they found that blocking BDNF signaling slows down the proliferation. However, the authors failed to provide a convincing evidence that the effect is mediated by glutamatergic synapse. TrkB activation can trigger number of different signaling pathways and AMPAR may be one of them. But there is no evidence that the effect of blockade of BDNF signaling is mediated by the blockade of glutamatergic synapse. I therefore, cannot recommend publication of this work in Nature. In practice, the authors should consider splitting the story into two papers.

We appreciate these comments and have now re-organized the flow of the story. We agree that the connection should be bolstered between the effects of BDNF on glioma synaptic biology and on glioma growth/mouse survival. To provide evidence that there is an important link between TrkB signaling and the AMPAR-mediated effect on glioma proliferation, we have performed several new experiments.

To explore the hypothesis that the growth-inhibitory effects of activity-regulated BDNF-TrkB signaling in glioma may involve modulation of synaptic biology, we asked whether effects of glioma TrkB signaling are related to or independent of AMPA receptor signaling. We found that pharmacologically blocking AMPA receptors, or genetically blocking TrkB through *NTRK2* KO, each decreased tumor cell proliferation *in vivo* or in neuron-to glioma co-culture (Figure 1 j-m and Extended Data Fig 5a). However, we found no additive effect of blocking AMPA receptors and TrkB in co-culture or *in vivo*, suggesting that the mechanisms may be related.

Figure 1j-m. **j**, Proliferation index of SU-DIPG-VI WT and *NTRK2* KO glioma co-culture with neurons, in the presence and absence of the AMPAR blocker NBQX (10 μ M; $n = 3$ coverslips/group, experiment also repeated in Extended Data Figure 5a). **k**, Experimental model of pontine injected WT and *NTRK2* KO glioma (SU-DIPG-VI) treated with the AMPAR blocker perampanel or vehicle control. **l**, Representative image of xenografted wild-type glioma cells treated with vehicle, and *NTRK2* KO cells treated with the

AMPA blocker perampanel quantified in **m**, gray denotes HNA positive glioma cells; red denotes Ki67. Scale bar = 50 μ m. **m**, Proliferation rate of wild-type (WT) and *NTRK2* KO glioma xenografts (SU-DIPG-VI) treated with the AMPAR blocker perampanel or vehicle control (quantified by Ki67⁺/HNA⁺; WT + vehicle; *n* = 6 mice, WT + perampanel; *n* = 7 mice, *NTRK2* KO + vehicle; *n* = 5 mice, *NTRK2* KO + perampanel; *n* = 6 mice).

Extended Data Fig 5a. a, Proliferation index of SU-DIPG-VI WT and *NTRK2* KO glioma monoculture (left), or glioma co-culture with neurons (right), in the presence and absence of the AMPAR blocker NBQX (10 μ M) (quantified as fraction of EdU⁺/HNA⁺ co-positive tumor cells assessed by confocal microscopy, *n* = 3 coverslips/group for glioma monoculture experiments and 6 coverslips/group for neuron-glioma co-culture; experiment also replicated in Figure 1j).

Minor comments.

It is hard to discern what structures are shown. For example, in Fig. 1i, what part of cells are shown? Need scale bar. Fig. 2D. What is the beads-like structure? Single en passant axon? Fig. 2F as well. Low magnification images might help.

We have included scale bars for all images and improved our description of the cellular structures shown in the legends.

Fig. 2H, I, S4D. The effect of BDNF is so small. Although there is statistical difference, it is hard to imaging such a small activation of a kinase is functionally meaningful.

It has been reported in the literature that an increase in phosphorylation of these glutamatergic channels during LTP is often at around 10-30% (Lee, Nature 2000; Caldeira, JBC 2007). We found BDNF increased phosphorylation by approximately 20% in our glioma cultures, which is concordant with the findings in the synaptic literature.

GluR4 should be used for pHluorin imaging. Also, S862A mutant should be tested.

We attempted to generate GluA4 pHluorin-expressing glioma cultures, and to generate a GluA4 S862A mutant but encountered technical issues with both constructs. In detail, we attempted to generate the two GluA4-pHluorin tagged plasmids (wild-type and S862A

mutant) using two different strategies over a 10 month period. The first by cutting the GluA4 transcript from our glioma cultures to ligate into the SEP(pHluorin)-TagBFP2 lentiviral backbone and performing site-directed mutagenesis. The wild-type construct generated after Gibson assembly of the transcript and backbone produced a fragmented construct of 6,332bp (expected 10,269bp) and the site-directed mutagenesis failed, producing only a circularized backbone. For the second strategy, we purchased a synthesized transcript of GluA4-SEP and ligated into a TagBFP2 lentiviral backbone. The Gibson assembly failed to generate colonies 3 times, after troubleshooting, we obtained colonies and isolated 2 plasmids for the wild-type and 2 plasmids for the S862A mutant, however sequencing confirmed both had failed, producing only the original construct (5,735bp). We performed the Gibson assembly and failed to produce bacterial colonies. We repurchased all reagents, including the Gibson Assembly and Stbl3 *E.coli*, and regenerated the plasmid backbone (sequenced confirmed) and successfully generated bacterial colonies; after isolating 5 plasmids for the wild-type and 5 plasmids for the S862A mutant, we found that all had the incorrect sequence, producing either fragmented constructs (6,816bp for wild-type and 7,311bp for S862A mutant), or producing circularized backbone (3,294bp). We apologize that we could not generate these constructs.

While we could not complete these suggested experiments, the additional cell surface biotinylation studies of GluA4 and GluA3 performed during the revision period further supports the conclusion that BDNF-TrkB signaling increases AMPAR trafficking to the membrane.

We are grateful for the Referee's helpful comments and suggestions, which have greatly improved the manuscript.

Referee #4 (Remarks to the Author):

Previous studies by the authors' group and others suggested that synaptic interactions between neurons and glioma cells play a key role in glioma progression (Venkatesh et al, 2019; Venkataramani et al, 2019; etc). In this manuscript, the authors attempted to extend these previous studies and examine whether the neuron-to-glioma malignant synapses are regulated by BDNF-TrkB signaling, a key regulator of synaptic plasticity. The authors provide the data showing that BDNF promotes AMPA receptor trafficking to the glioma cell membrane, resulting in increased amplitude of glutamate-evoked currents in the malignant cells. BDNF-TrkB signaling also regulates the number of neuron-to-glioma synapses. They also showed that blocking TrkB signaling attenuated tumor growth. BDNF regulation of synaptic plasticity including structural (spine growth) and functional (transmitter release, AMPA receptor trafficking) plasticity has been well established. It is therefore not surprising that similar mechanisms are used in neuron-glia synapses. The pan-Trk inhibitors, e.g. entrectinib used in therapeutic targeting of TrkB in pediatric glioma of this study, have also been approved by FDA to treat cancer, including gliomas. Thus, the current study has not reached the level of novelty and

significance needed for Nature. Further, the key is to demonstrate that BDNF enhances glioma progression by regulating neuron-glioma synapses specifically, rather than promoting glioma cell growth per se. The experiments using entrectinib in vivo does not prove that the effects are mediated by inhibition of neuron-glia synapses.

Additional major comments:

1. A key experimental setting is the human glioma cells xenografted onto mouse hippocampal slices. This is an artificial system that may not reflect the real situation in the brain of glioma patients. The mouse hippocampal CA1 neurons would presumably sprout their axon terminals to form synapses with the cultured human glioma cells. It is unclear whether these mouse-human synapses in vitro share the same properties, plasticity and regulatory mechanisms as the neuron-glioma synapses in patients' brain in vivo. Single cell gene expression profiling of cells in the xenograft model may help determine whether presynaptic neurons and postsynaptic glioma cells exhibit similar features as those in human glioma in vivo. Ex vivo electrophysiology experiments using surgically derived human glioma tissues containing neurons and glioma cells might also be helpful.

We thank the Referee for these helpful comments that underscore our need to better explain the background literature in the manuscript. We have now added more clarification to the text.

It is important to clarify that our experimental model system does not place glioma cells onto hippocampal slices, but rather that glioma cells are xenografted into the hippocampus and allowed to engraft/invade/interact with the in vivo microenvironment for 8 weeks. Then acute slices of the these xenografted hippocampi are used for electrophysiology.

In our previous publication (Venkatesh et al., 2019 Nature), we extensively examined single cell RNA sequencing data from both primary human glioma biopsy samples and patient-derived xenograft samples to confirm that our models reflect the synaptic biology that we observe in our experimental set ups. In the back-to-back publication by Venkataramani et al. (2019 Nature) electrophysiological recordings of primary human GBM biopsy material demonstrated the same neuron-to-glioma AMPAR-mediated synapses that we found in our human xenograft models (Venkataramani et al., Nature 2019).

2. It might be incorrect to use the term 'LTP' here. The classic LTP experiments involve a long-term (hours) enhancement of synaptic connections between presynaptic terminals and postsynaptic cells. Here a glutamate puffing instead of stimulation of presynaptic neurons was used to induce currents (non-synaptic) in glioma cells. There was no evidence of its NMDAR and Ca²⁺ influx dependence. AMPA receptor insertion into glioma cell membranes without NMDAR may be irrelevant to LTP. Thus, a transient enhancement of the glutamate-induced currents seen in Figure 1, albeit its involvement of AMPA receptor insertion, is far from the Hebbian type activity-dependent synaptic potentiation.

In our original introduction, we were aiming to describe the concept of plasticity to a broader audience and highlight the classical mechanism of LTP. We have now carefully reworded our text and removed mention of LTP in the manuscript. As glioma cells do not express NMDA receptors, the mechanism is certainly not that of classical LTP, and we have clarified this in the text.

3. One also needs to distinguish between “BDNF enhancement of basal synaptic transmission” and “BDNF regulation of LTP” (Kang et al, 1995; Figorov, 1996; Patterson, 1996; Korte, 1995; see Ji et al 2009 for in depth analyses). The form of plasticity described here at the best is “BDNF-enhancement of glutamate-puff induced currents”.

As suggested, we have now evaluated the glioma cell electrophysiological response to axonal afferent electrical stimulation, with and without BDNF perfusion (Figure 2i-k).

Figure 2i-k. i, Electrophysiological model of GFP⁺ glioma cells (green) xenografted in mouse hippocampal CA1 region with Schaffer collateral afferent stimulation. j, Representative averaged voltage-clamp traces of evoked glioma excitatory postsynaptic current (EPSC) in response to axonal stimulation (black arrow) before (grey) and after (blue) application of BDNF protein (100 ng/ml in ACSF, 30 min). k, Quantification of data in j ($n = 5$ glioma cells exhibiting EPSCs out of 43 glioma cells patched from 4 mice).

We thank the referee for these important points, and have also now cited these important references.

4. The Figure 4 a-d showed that BDNF alone increased glioma proliferation by 20-30%, but addition of neurons to the glioma culture increased proliferation by 30-60%, and this effect is attenuated in glioma with TrkB gene deletion. The authors interpreted this result as neuron-glioma synapse playing additional role in glioma proliferation. However, it is well known that neurons are the key source of BDNF, and it is difficult to rule out whether addition of neurons to glioma culture was simply adding more BDNF to the culture. To establish neuron-glioma synapse is important, one needs to demonstrate that blockade of synaptic transmission (e.g. using AMPA receptor antagonists or botulinum toxins) could abrogate the effects of adding neurons to glioma culture.

This is an excellent suggestion. We previously demonstrated that the addition of the AMPAR inhibitor, NBQX, to neuron-glioma co-cultures reduced glioma proliferation (Venkatesh 2019 Nature) and we have now re-demonstrated this here (Figure 1j and

Extended Data Figure 5a). To examine the extent to which BDNF exerts its effects on glioma proliferation through modulation of AMPARs, we have now performed neuron co-cultures with WT and *NTRK2* KO glioma cells and treated with NBQX. If it is correct that BDNF-TrkB signaling in glioma promotes proliferation through the effects on AMPAR signaling, then loss of TrkB should have a minimal effect in the absence of AMPAR signaling, which is what we observed (Figure 1j and Extended Data Figure 5a).

Figure 1j. j, Proliferation index of SU-DIPG-VI WT and *NTRK2* KO glioma co-culture with neurons, in the presence and absence of the AMPAR blocker NBQX (10 μ M; $n = 3$ coverslips/group, experiment also repeated in Extended Data Figure 5a).

Extended Data Fig 5a. a, Proliferation index of SU-DIPG-VI WT and *NTRK2* KO glioma monoculture (left), or glioma co-culture with neurons (right), in the presence and absence of the AMPAR blocker NBQX (10 μ M) (quantified as fraction of EdU⁺/HNA⁺ co-positive tumor cells assessed by confocal microscopy, $n = 3$ coverslips/group for glioma monoculture experiments and 6 coverslips/group for neuron-glioma co-culture; experiment also replicated in Figure 1j).

5. Figure 4h showed that mice xenografted with glioma cells bearing TrkB KO or treatment with Trk inhibitors survived longer than those xenografted with WT glioma cells. This could simply be interpreted as BDNF-TrkB signaling is important for glioma growth or proliferation, and has nothing to do with its regulation of neuron-glioma synapses – a key point of this manuscript.

We thank the Referee for this helpful suggestion. We have now also examined the intersection of BDNF-TrkB signaling and glioma AMPAR biology *in vivo*. We found that pharmacologically blocking AMPA receptors, or genetically blocking TrkB through

NTRK2 KO, each decreased tumor cell proliferation *in vivo* (Figure 1 k-m). However, we found no additive effect of blocking AMPA receptors and TrkB in co-culture or *in vivo*, suggesting that the mechanisms may be related.

Figure 1k-m. **k**, Experimental model of pontine injected WT and *NTRK2* KO glioma (SU-DIPG-VI) treated with the AMPAR blocker perampanel or vehicle control. **l**, Representative image of xenografted wild-type glioma cells treated with vehicle, and *NTRK2* KO cells treated with the AMPAR blocker perampanel quantified in **m**, gray denotes HNA positive glioma cells; red denotes Ki67. Scale bar = 50µm. **m**, Proliferation rate of wild-type (WT) and *NTRK2* KO glioma xenografts (SU-DIPG-VI) treated with the AMPAR blocker perampanel or vehicle control (quantified by Ki67⁺/HNA; WT + vehicle; *n* = 6 mice, WT + perampanel; *n* = 7 mice, *NTRK2* KO + vehicle; *n* = 5 mice, *NTRK2* KO + perampanel; *n* = 6 mice).

Minor points:

1. There is no data showing if BDNF treatment can enhance basal GCaMP6 fluorescence.

We have now included the analysis of basal GCaMP6 levels upon BDNF treatment, in the absence of a glutamate puff. There is no change in basal GCaMP6s intensity in the absence of glutamate puff (Extended Data Figure 5e).

Extended Data Fig 5e. e, Baseline GCaMP6s intensity in SU-DIPG-VI glioma cells before and 30-sec after BDNF exposure, in the absence of glutamate puff (*n* = 7 cells, 3 mice). **f**, GCaMP6s intensity trace of SU-

DIPG-VI glioma cells response to glutamate puff before (3 cells, 3 mice : light grey, average: dark grey) and after BDNF perfusion (three cells: light blue, average intensity: dark blue).

2. The citation of references contains many errors. For example, Kang et al showed that BDNF enhances basal synaptic transmission (1995). BDNF regulation of hippocampal LTP was demonstrated by Figurov (1996), Korte (1995), Patterson (1996).

We have added these citations in the revised manuscript, and corrected reference to the Kang et al. paper. Thank you for pointing this out.

3. One also wonders how calcium enters into glioma cells after glutamate puffing. There is a need to demonstrate the expression of NMDAR or calcium channels on the cell surface of glioma cells.

We previously reported that pediatric gliomas do not express NMDA receptors (Venkatesh et al., *Nature* 2019). We have now performed electrophysiological recordings of glioma cells with glutamate puff in the presence of an NMDAR inhibitor, AP-5 and find no NMDAR-mediated currents.

Figure 2c-d. **c**, Representative voltage-clamp traces of whole cell patch-clamp electrophysiological recordings in glioma cells. Hippocampal slices were perfused with ACSF containing tetrodotoxin (TTX, 0.5 μ M), and response to a local puff (250 msec) application of 1 mM glutamate (black square) was recorded from xenografted glioma cells with sequential application of NMDAR blocker (AP-5, 100 μ M), TBOA (200 μ M), AMPAR blocker (NBQX, 10 μ M). **d**, Quantification of data in **c** ($n = 7$ glioma cells, 4 mice).

We and others have also previously reported that the AMPA receptors in glioma are under-edited and calcium-permeable (Venkatesh et al., 2019 *Nature*; Venkataramani et al., 2019 *Nature*). In addition, gliomas single cell RNAseq datasets reveal calcium channel gene expression (Venkatesh et al., 2019 *Nature*), also shown below.

AMPA

NMDAR

Calcium channels

Expression of AMPAR, NMDAR and calcium channel genes in primary biopsy samples of high-grade gliomas. Single cell transcriptomic data from malignant cells from primary human biopsy samples of the major classes of high-grade glioma (H3K27M-mutant diffuse midline gliomas (grey), IDH WT hemispheric high-grade gliomas (red) and IDH-mutant hemispheric high-grade gliomas (purple), as well as normal cells in the tumor microenvironment (immune cells, green, and oligodendrocytes, yellow). Data from Venkatesh et al., 2019 *Nature*.

4. In Fig.1, it is unclear whether or not glioma itself has AMPA receptor without contacting the hippocampal tissues.

Single cell transcriptomic data from primary human biopsy samples demonstrate robust AMPAR subunit gene expression in the malignant cells, as shown above (Venkatesh et al., 2019 *Nature*; Venkataramani et al. 2019 *Nature*). Furthermore, our protein western blot data of patient-derived glioma cells in culture demonstrate expression AMPA receptors at baseline.

5. The use of the pan-Trk inhibitor Entrectinib may block NGF, and NT3 signaling, rather than BDNF-TrkB.

This is an important comment, and have addressed this possibility in the following ways:

1. To demonstrate that Trk inhibitors are exerting therapeutic benefit through glioma TrkB expression, we treated mice bearing TrkB WT and KO glioma xenografts with the Trk inhibitor entrectinib. While entrectinib decreased the proliferation rate of xenografted *NTRK2* WT DIPG cell in vivo, it did not decrease the proliferation rate of *NTRK2* KO glioma xenograft (Extended Data Figure 2e-f), demonstrating that the mechanism of action of entrectinib in DIPG is mediated through TrkB.

Extended Data Fig 2e-f. **e**, Experimental model of pontine xenografted WT and *NTRK2* KO glioma (SU-DIPGVI) treated with the Pan-Trk inhibitor, entrectinib, or vehicle control. **f**, Proliferation index of wild-type and *NTRK2* KO SU-DIPGVI glioma xenografted to the pons of NSG mice and treated with vehicle or entrectinib (120mg/kg P.O.). Quantification by confocal microscopy analysis of EdU+/HNA+ co-positive tumor cells, $n = 4$ wild-type glioma xenografted, vehicle-treated mice, 5 wild-type glioma xenografted, entrectinib-treated mice, 5 *NTRK2* KO glioma xenografted, vehicle-treated mice, 3 *NTRK2* KO glioma xenografted, entrectinib-treated mice).

2. To examine the possible role of NGF and NT3 signaling, we tested glioma proliferation in response to NGF and NT3 and found no effect (Extended Data Figure 1f).

6. In Fig. 4e-f, one needs to show that it is truly BDNF but not other factors in the conditional media that stimulated proliferation. Similarly, one needs to show that it was the lack of BDNF but not other factors from the Bdnf-TMKI xenografts that prolonged the survival of mice (Fig. 4g)

We used the same experimental paradigm for the conditioned media experiments as in our Venkatesh et al., 2015 *Cell* paper, in which we extensively characterized the factors secreted in response to neuronal activity in cortical slices and performed sufficiency and necessity testing to determine which activity-regulated paracrine factors influence glioma proliferation. This showed that the proliferation-inducing effects of cortical slice conditioned medium is chiefly accounted for by a shed form of NLGN3 and BDNF (Venkatesh et al., 2015 *Cell*).

In response, we have now performed *in vivo* optogenetic experiments, stimulating cortical projection (glutamatergic) neuronal activity in WT and BDNF-TMKI mice. We observed the expected increase in glioma proliferation in WT mice, but the proliferative effect of glutamatergic neuronal activity on glioma proliferation was markedly attenuated in mice lacking activity-regulated BDNF expression and secretion. (Figure 1a-d and Extended Data 1c-d).

Figure 1a-d. **a**, Schematic of Bdnf-TMKI mouse, which lacks activity-regulated *BDNF* expression. **b**, Optogenetic model for optogenetic stimulation of ChR2-expressing neurons (blue) in microenvironment of glioma xenograft (green); light blue rectangle denotes region of analysis. P, postnatal day. **c**, Representative image of glioma cells (SU-DIPG-VI) xenografted into wild-type and Bdnf-TMKI NOD-SCID-gamma (NSG) mice in the presence of optogenetically stimulated neurons quantified in **d**, gray denotes human nuclear antigen (HNA)-positive glioma cells; red denotes Ki67 (proliferative marker). Scale bar = 50 μ m. **d**, Proliferation index of SU-DIPG-VI glioma xenografted to mice with Thy1+ glutamatergic cortical projection neurons lacking (ChR2-) or expressing Channelrhodopsin (ChR2+) in a wild-type or Bdnf-TMKI genetic background (quantified by Ki67⁺/HNA, $n = 6$ wild-type ChR2- mice, 4 Bdnf-TMKI ChR2- mice, 7 wild-type ChR2+ mice, 4 Bdnf-TMKI ChR2+ mice).

Extended Data Fig 1c-d. **c** Model for optogenetic stimulation of ChR2-expressing neurons (blue) in microenvironment of glioma xenograft (green); light blue rectangle denotes region of analysis. P, postnatal day. **d**, Proliferation index of SU-DIPGXIIIIFL glioma xenografted to mice with neurons expressing Channelrhodopsin (ChR2+) in a wild-type or *Bdnf-TMKI* genetic background (Figure 1a) after neuronal optogenetic stimulation (quantified by confocal microscopy of EdU+/HNA cells, $n = 7$ wild-type ChR2+ mice, 8 *Bdnf-TMKI* ChR2+ mice).

As a further control, we stimulated cortical projection neuronal activity in WT and *Bdnf-TMKI* mice but with *NTRK2* KO glioma cells, and found a similar proliferation rate in *NTRK2* KO glioma-xenografts with or without activity-regulated BDNF, indicating that loss of activity-regulated BDNF does not exert effects that are independent of glioma TrkB signaling. (Extended data Fig. 1k).

Extended Data Fig1k. Proliferation index of *NTRK2* KO SU-DIPG-VI glioma xenografted to mice with neurons expressing Channelrhodopsin (ChR2+) in a wild-type or *Bdnf-TMKI* genetic background after neuronal optogenetic stimulation (quantified by confocal microscopy of EdU+/HNA cells, $n = 5$ wild-type ChR2+ mice, $n = 4$ BDNF-TMKI ChR2+ mice).

7. The author claimed that BDNF-TrkB signaling promotes calcium-permeable AMPA receptor trafficking and consequently depolarizes the glioma cell membrane. It is unclear whether or not voltage-gated calcium channels are involved in increased intracellular calcium signaling.

While we and others have shown that the AMPA receptors expressed in glioma are calcium permeable (Venkatesh et al, 2019 Nature; Venkataramani et al., 2019), we do

not mean to suggest that is the only source of calcium in the calcium transients and we will clarify this in the text. Several voltage-gated calcium channels are expressed in glioma (as shown above in response to point #3), and voltage-gated calcium channel biology is a broad and important topic that we and others will be studying for some time.

Errors:

1. page 7, line 153 and 154, VGF and GBM should be shown in full name when first time presented. page 8, line 190, it should display full name for DMG. page 16, line 333, full name should be mentioned first shown for SU-DIPGVI.
2. page 19, line 373-374, it should be: "blue denotes nestin staining", and "green denotes synapsin", respectively.
3. page 27, line 460, "X-axis" should be changed to "Y-axis"; page 29, line 485, "the x axis" should be changed to "the y axis".

Thank you for these helpful corrections. We have corrected these points in the revised manuscript.

We are grateful for the Referee's helpful comments and suggestions, which have greatly improved the manuscript.

Reviewer Reports on the First Revision:

Referees' comments:

Referee #1 (Remarks to the Author):

The authors have responded exceptionally well to all of my points, and have now included extensive new experimental data that collectively addresses all key points in an excellent way, which strengthens the methodology and further improved the story.

I have no remaining issues (only one minor: in Fig. 4, "syapsin" needs to be corrected in a panel).

Referee #2 (Remarks to the Author):

The authors have responded to the initial review with numerous new experiments and reorganized the manuscript to address other concerns. These changes have effectively addressed nearly all the issues raised in the first round of review. The one question that the authors did not address experimentally is whether adaptive plasticity mediates the effects they see. One option would be for the authors to test known mechanisms that mediate different types of adaptive plasticity; for instance blocking signaling that mediates different types of adaptive plasticity such as RA signaling (homostatic), CamKII pathway (LTP), or calcimerin pathway (LTD).

The the new experiments generated a few unexpected findings that in some cases complicate the results. For instance, the authors show that there is not an additive effect of blocking the AMPAR and knocking out trks in the glioma and suggest that this might indicate a shared mechanism. This is an interesting and important idea that would strengthen their hypothesis. However, the mechanism is not explored. It is also unexpected given the previous results that the NLGN3 plays no role in the change in glutamtergic responses observed. These two unexpected findings suggest that the authors have not yet found the key mechanism that mediates recruitment of AMPARs. Results from these experiments might have provided additional suport for the adaptive plastiicty model, but do not in their current form.

Referee #3 (Remarks to the Author):

The authors have made a significant effort to improve the logical flow of the manuscript by adding new data and rewriting, which I saw was the major problem with the earlier version. The paper is now more coherent and easier to follow. I appreciate the authors' efforts. However, I still have mixed feelings about its suitability for Nature. The first paper by this group is truly appropriate for Nature, as it shows that excitatory synapses are formed on glioma cells. I have no doubt about that. This paper, on the other hand, the AMPAR part, which is still the major part of the study, is incremental. The more interesting part is the role of BDNF in the regulation of AMPAR. However, at

the same time, Google search of "BDNF" and "glioma" shows a few studies already pointing to the regulation of glioma growth by BDNF. None of these studies performed the analyses at a high standard as seen in this paper, but still partially diminishes the excitement.

In conclusion, from my point of view, this paper is well written and I have no objection to its publication. The experiments are generally performed at a high standard using human preparations that are not readily available. For the suitability to Nature, I am a bit hesitant to fully support that due to overlap with the previous publication.

Minor comments.

I understand that the authors tried to use GluR4 but somehow failed. It is becoming increasingly clear that four different AMPAR subtypes are quite different, and especially the GluR2(Q) mutant has a strange structure not seen in other subunits (see for example studies by Greger et al). This should be noted.

The glioma cells in xenografted sections do not look like they are forming a tumor, as would be expected for a glioma. Rather, they are isolated. Low magnification images should be shown.

Expanded data 1f. Need BDNF here as a positive control, similar to Fig. 1h. Why is NT4 not effective? It also works through TrkB.

Referee #4 (Remarks to the Author):

The authors have addressed most of my concerns and questions. It would be nice if they could attempt to perform electrophysiological experiments using surgically derived human glioma tissues which contain both neurons and glioma cells. It would be truly clinically significant if blockade of neuron-glioma synapse or inhibition of BDNF-TrkB signaling could truly attenuate glioma proliferation.

Author Rebuttals to First Revision:

We appreciate the thoughtful Referee and Editor comments. A key question is to link plasticity mechanisms to functional/clinical outcome. This is an important question that we have now directly addressed with new data: We had previously shown that membrane depolarization promotes glioma proliferation (Venkatesh et al., 2019). Now, in response to this excellent question, we have demonstrated that the *magnitude* of depolarization matters: optogenetically modulating the amplitude of glioma cell depolarization to model the effects of plasticity of malignant synaptic strength demonstrates that the magnitude of the glioma current differentially promotes glioma proliferation. Greater glioma membrane depolarization results in a greater increase in tumor cell proliferation (Now shown in new Figure 4h-k and Extended Data Figure 10e-g.)

Figure 4h-k: **h**, Electrophysiological trace of channelrhodopsin-expressing (ChR2+) glioma (SU-DIPG-XIII-FL) in response to optogenetic stimulation (470 nm, 1.0 mW/mm²) with either 5 ms (light blue) or 25 ms (dark blue) light pulse width. **i**, Quantification of total accumulated charge upon 2 seconds of optogenetic stimulation at a frequency of 20 Hz with 5 ms or 10 ms light pulse width as shown in **h**, in comparison to no blue light stimulation ($n = 5$ glioma cells per group). **j**, Optogenetic model for optogenetic stimulation of xenografted ChR2+ glioma (blue); light blue rectangle denotes region of analysis. P, postnatal day. **k**, Proliferation index of xenografted ChR2+ SU-DIPG-XIII-FL glioma in a wild-type genetic background after mock stimulation or blue light stimulation at 5 ms or 25 ms light pulse width (quantified by Ki67⁺/HNA, $n = 3$ mice in each group). Data are mean \pm s.e.m. * $P < 0.05$, ** $P < 0.01$, Two-tailed unpaired Student's t -test for **k**. One-sample t -test for **i**.

Extended Data Figure 10e-g: **e**, The cation channel, Channelrhodopsin-2, is gated by blue light, inducing membrane depolarization of the cell. **f**, Electrophysiological traces of patch-clamped glioma cells stimulated with 470 nm light at 20 Hz, 1.0 mW/mm² for 2 seconds (blue lines) at either 5 ms (light blue) or 25 ms (dark blue) light pulse width. Note the difference in current amplitude elicited by 5 ms vs 25 ms light pulse widths. **g**, Representative images of xenografted ChR2+ glioma cells quantified in 4k after mock stimulation, or optogenetic stimulation at 5 ms and 25 ms light pulse width, gray denotes HNA positive glioma cells; red denotes Ki67. Scale bar = 50 μ m.

In addition, supporting the clinical relevance of the increased synaptic connectivity findings, work in the glioma literature reveals increased synaptic gene expression at tumor recurrence compared to initial biopsy (Varn et al., 2022), concordant with this manuscript's demonstration of activity-dependent (and therefore likely to increase with time/circuit activity) BDNF-TrkB signaling elaboration of synapse numbers.

Response to Referees' Comments

Referee #1 (Remarks to the Author):

The authors have responded exceptionally well to all of my points, and have now included extensive new experimental data that collectively addresses all key points in an excellent way, which strengthens the methodology and further improved the story.

We are grateful for these positive and supportive comments.

I have no remaining issues (only one minor: in Fig. 4, "syapsin" needs to be corrected in a panel).

Thank you: we have fixed the typo in Fig. 4.

Referee #2 (Remarks to the Author):

The authors have responded to the initial review with numerous new experiments and reorganized the manuscript to address other concerns. These changes have effectively addressed nearly all the issues raised in the first round of review. The one question that the authors did not address experientially is whether adaptive plasticity mediates the effects they see. One option would be for the authors to test known mechanisms that mediate different types of adaptive plasticity; for instance blocking signaling that mediates different types of adaptive plasticity such as RA signaling (homostatic), CamKII pathway (LTP), or calciumerin pathway (LTD).

The new experiments generated a few unexpected findings that in some cases complicate the results. For instance, the authors show that there is not an additive effect of blocking the AMPAR and knocking out *trks* in the glioma and suggest that this might indicate a shared mechanism. This is an interesting and important idea that would strengthen their hypothesis. However, the mechanism is not explored.

We appreciate these comments and have now explored mechanism more deeply. We now show that BDNF signaling promotes CAMKII phosphorylation (Extended Data Figure 8a, d) and CAMKII inhibition reduces glioma cell proliferation (Extended Data Figure 5b) and abrogates the electrophysiological effect of BDNF on glioma current amplitude (Figure 2d-e and shown below).

d, Representative traces of xenografted SU-DIPG-VI cells in response to glutamate puff (black square), before and after perfusion with 100ng/ml BDNF recombinant protein in ACSF (containing TTX, 0.5 μ M) for 30 minutes (blue). Left trace, control condition; middle trace, with the addition of a CAMKII inhibitor, KN-93 (10 μ M); right trace, with the addition of the KN-93 inactive analog, KN-92 (10 μ M). **e**, Quantification of data in **d** ($n = 6$ cells per group, from 5 mice for control, 3 mice for KN-93 treated and 3 mice for KN-92 treated).

It is also unexpected given the previous results that the NLGN3 plays no role in the change in glutamatergic responses observed. These two unexpected findings suggest that the authors have not yet

found the key mechanism that mediates recruitment of AMPARs. Results from these experiments might have provided additional support for the adaptive plasticity model, but do not in their current form.

In our previous extensive work on NLGN3, we found that NLGN3 is shed in a neuronal activity-dependent manner (cleaved at the membrane from neurons and oligodendrocyte precursor cells by ADAM10) and that shed NLGN3 functions to **1. Promote glioma cell proliferation by stimulating oncogenic signaling pathways, including PI3K-mTOR, RAS and SRC pathways** (Venkatesh et al., 2015 Cell; Venkatesh et al., 2017 Nature), **2. Upregulate glioma cell expression of synaptic genes** (Venkatesh et al., 2017 Nature) and **3. Promote neuron-to-glioma synapse formation** (Venkatesh et al., 2019 Nature). It is not surprising that soluble NLGN3 does not acutely regulate the strength of glutamatergic currents as BDNF does: While NLGN3 plays different, important roles in glioma pathophysiology, we did not expect NLGN3 binding to the glioma cell to increase glutamatergic current amplitude on this timescale. We tested NLGN3 as a control to show that not all paracrine factors involved in neuron-to-glioma interactions exert the same effect on glutamatergic current amplitude.

Referee #3 (Remarks to the Author):

The authors have made a significant effort to improve the logical flow of the manuscript by adding new data and rewriting, which I saw was the major problem with the earlier version. The paper is now more coherent and easier to follow. I appreciate the authors' efforts. However, I still have mixed feelings about its suitability for Nature. The first paper by this group is truly appropriate for Nature, as it shows that excitatory synapses are formed on glioma cells. I have no doubt about that. This paper, on the other hand, the AMPAR part, which is still the major part of the study, is incremental. The more interesting part is the role of BDNF in the regulation of AMPAR. However, at the same time, Google search of "BDNF" and "glioma" shows a few studies already pointing to the regulation of glioma growth by BDNF. None of these studies performed the analyses at a high standard as seen in this paper, but still partially diminishes the excitement.

In conclusion, from my point of view, this paper is well written and I have no objection to its publication. The experiments are generally performed at a high standard using human preparations that are not readily available. For the suitability to Nature, I am a bit hesitant to fully support that due to overlap with the previous publication.

We appreciate these comments, and have now clarified the distinctions between previous work and this paper. Previous work on BDNF in glioma by other groups focused on paracrine signaling of BDNF between tumor cells. In our past work, we had shown a role for neuronal activity-dependent paracrine BDNF signaling between neurons and glioma cells (Venkatesh et al., 2015). This manuscript demonstrates a much greater therapeutic benefit of targeting BDNF-TrkB signaling than would be expected based on that relatively weak effect of BDNF as a paracrine mitogen, and instead uncovers a role in plasticity of neuron-to-glioma synaptic strength and number. The novelty here is not BDNF per se, but rather the concept that neuron-to-cancer cell synapses can hijack mechanisms of adaptive plasticity, a finding that would predict reinforcement and elaboration of neuron-to-glioma synapses over time and with experience/nervous system activity. Such a prediction fits well with clinical and molecular observations (e.g. Varn et al., 2022) and could explain several aspects of glioma pathophysiology that would have remained enigmatic without the new and clinically important insights presented here.

Minor comments.

I understand that the authors tried to use GluR4 but somehow failed. It is becoming increasingly clear that

four different AMPAR subtypes are quite different, and especially the GluR2(Q) mutant has a strange structure not seen in other subunits (see for example studies by Greger et al). This should be noted. We thank the referee for this helpful suggestion and have now added this important point and Greger references to the discussion, stating *“In neurons, AMPAR subunit composition influences receptor structure and electrophysiological properties^{59,60}, and it remains to be determined if varying subunit composition may contribute to variation in glioma AMPAR-mediated currents.”*

The glioma cells in xenografted sections do not look like they are forming a tumor, as would be expected for a glioma. Rather, they are isolated. Low magnification images should be shown.

Many gliomas, including diffuse midline gliomas and diffuse hemispheric gliomas, do not form nodular tumors, but rather diffusely infiltrate – and synaptically integrate – into the brain. The growth pattern of glioma cells in these patient-derived xenografted mice mirrors what is seen in patients. Low magnification images are shown in Extended Data Figure 1e, i and j.

Expanded data 1f. Need BDNF here as a positive control, similar to Fig. 1h. Why is NT4 not effective? It also works through TrkB.

We appreciate this comment and have now repeated this experiment with BDNF as a positive control. In the initial testing of NGF, NT3 and NT4 (performed 3 independent times, in duplicate each time), there was a trend towards a proliferative effect with NT4, but no significant increase in proliferation. We have now repeated this experiment with newly purchased recombinant neurotrophins, and we found a clear proliferative effect of NT4 (and BDNF), but not NT3 nor NGF. Repeating the experiment again replicated these results. We are grateful for this question and the opportunity to correct this. Shown here are all data points from all experiments, color coded by experiment (black – original data with older recombinant proteins, red and blue – new data with newly purchased recombinant proteins replicated in two independent experiments), as well as a representative experiment with the newly purchased recombinant proteins and the BDNF positive control (Extended Data Figure 1f).

Proliferation rate of SU-DIPG-XIII-FL cultures treated with recombinant proteins NGF, NT3, NT4 (100 μM each), compared to vehicle control (quantified by confocal microscopy of EdU+/DAPI cells. Left panel: $n = 8-11$ coverslips/group (black = the original data presented in Revision #1, each dot represents the average of two coverslips from 3 independent experiments, red and blue = new experiments, each dot represents one coverslip). Right panel: 4 coverslips/group ('red' experiment). Data are mean \pm s.e.m. **P < 0.01, ns = not significant. Kruskal-Wallis test with Dunn's post hoc analysis for left panel and one-way analysis of variance (ANOVA) with Tukey's post hoc analysis for f (right panel).

Referee #4 (Remarks to the Author):

The authors have addressed most of my concerns and questions. It would be nice if they could attempt to perform electrophysiological experiments using surgically derived human glioma tissues which contain both neurons and glioma cells. It would be truly clinically significant if blockade of neuron-glioma synapse or inhibition of BDNF-TrkB signaling could truly attenuate glioma proliferation.

We appreciate these supportive comments and agree that it would be nice to be able to study this biology directly in patient tissue. However, electrophysiological experiments are not possible in primary patient tissue in this context for the following reasons: 1. This work was performed chiefly in diffuse midline gliomas given their broad expression of TrkB and lack of BDNF expression. Diffuse midline gliomas are never surgically resected because they grow diffusely through structures like the pons, so such tissue is not available. 2. For hemispheric high-grade gliomas like glioblastoma, which are surgically resected, we are unfortunately unable to collect the tissue into ice-cold, oxygenated media in the operating room due to strict institutional rules that all tissue must be examined by neuropathology prior to release for research. (We in fact attempted electrophysiology from surgically resected tissue in our 2019 paper, and found we could not get it in time to preserve neuronal integrity in our hospital system.) In terms of clinical significance, clinical studies of perampanel to block neuron-to-glioma synapses are ongoing at multiple sites, and we are planning a national clinical trial of a Trk inhibitor for (*NTRK* non-fusion) pediatric gliomas based on the results presented in this manuscript.

Author Rebuttals to First Revision:

Referees' comments:

Referee #2 (Remarks to the Author):

The authors have responded effectively to my comments and addressed them with new experiments analysis. The new data more directly linking neuronal activity to outcome is quite interesting and does extend the work into newer areas. My only suggestion is that the authors clarify their use of NLGN3 as a negative control.

Reviewer Reports on the Second Revision:

Response to Referees

The authors have responded effectively to my comments and addressed them with new experiments analysis. The new data more directly linking neuronal activity to outcome is quite interesting and does extend the work into newer areas. My only suggestion is that the authors clarify their use of NLGN3 as a negative control.

We appreciate the positive comments and have added the suggested clarification, now stating:

“In comparison, and as a control to assess specificity of BDNF amongst other known neuron-to-glioma paracrine factors^{1,2,4}, soluble NLGN3 was similarly tested and exerted no acute effect on glutamatergic current amplitude in glioma cells (Extended Data Figure 6e, f).”